# Single-cell division tracing and transcriptomics reveal cell types and differentiation paths in the regenerating lung

Leila R. Martins [1,2] ✉, Lina Sieverling[2,3], Michelle Michelhans[1,2,3,4], Chiara Schiller [1,2,5], Cihan Erkut [1,2], Thomas G. P. Grünewald [2,6,7,8,9], Sergio Triana [10,11,12], Stefan Fröhling [2,3,9,13], Lars Velten [14,15], Hanno Glimm [16,17,18,19] & Claudia Scholl [1,2] ✉

Understanding the molecular and cellular processes involved in lung epithelial regeneration may fuel the development of therapeutic approaches for lung diseases. We combine mouse models allowing diphtheria toxin-mediated damage of specific epithelial cell types and parallel GFP-labeling of functionally dividing cells with single-cell transcriptomics to characterize the regeneration of the distal lung. We uncover cell types, including *Krt13*+ basal and *Krt15*+ club cells, detect an intermediate cell state between basal and goblet cells, reveal goblet cells as actively dividing progenitor cells, and provide evidence that adventitial fibroblasts act as supporting cells in epithelial regeneration. We also show that diphtheria toxin-expressing cells can persist in the lung, express specific inflammatory factors, and transcriptionally resemble a previously undescribed population in the lungs of COVID-19 patients. Our study provides a comprehensive single-cell atlas of the distal lung that characterizes early transcriptional and cellular responses to concise epithelial injury, encompassing proliferation, differentiation, and cell-to-cell interactions.

Respiratory diseases are among the leading causes of death worldwide[1]. In the adult lung, epithelial cells possess a low steady-state turnover but can respond to lung injury with proliferation and differentiation to replace damaged cells[2]. Understanding the process of lung regeneration at the cellular and molecular level, including the identification of progenitor cells, is a prerequisite for the development of novel therapeutic strategies. Recent advances in single-cell transcriptomics of the human lung have enabled the identification of new cell types, increased the understanding of differentiation trajectories during lung development and regeneration, and allowed the comprehension of pathological processes from a cell type-specific point of view[3–7].

Lung epithelial regeneration is commonly studied in mice, in which epithelial cells must first be injured to trigger repair. This is achieved by various means, including exposure to toxic gases[8–10], detergents[11], infectious agents[12], chemicals[13,14], and genetic approaches[13,15–17]. The advantage of genetic approaches is that they allow the depletion of specific cell types without interfering with the rest of the tissue.

The mouse trachea is lined by a pseudostratified epithelium containing ciliated cells, secretory goblet and club cells, basal cells, and rarer cell types such as tuft and neuroendocrine cells[2]. The morphology and cellular composition changes gradually from the trachea to the bronchioles, where the cuboid epithelium is composed mainly

of ciliated and club cells. Rare bronchioalveolar stem cells (BASCs), co-expressing *Scgb1a1* and *Sftpc*, can be found in the bronchioalveolar duct junction. The alveoli are lined by cuboid alveolar type 2 cells (AT2) specialized in surfactant secretion, and flat alveolar type 1 cells (AT1) responsible for gas exchange[2].

In the damaged lung, different epithelial cells can self-renew and differentiate, depending on the region and type of injury[18]. In the mouse trachea and main bronchi, basal cells are considered the major stem cell population[19]. Lineage tracing of *Scgb1a1*+ cells demonstrated that club cells act as progenitors for goblet[20] and ciliated[9] cells, and can regenerate the alveoli after injury with bleomycin or influenza[21]. BASCs have been reported to contribute to bronchioalveolar regeneration by their ability to differentiate into AT2, club, and ciliated cells[13]. Lastly, AT2 cells maintain the alveolar epithelium by self-renewing and differentiating into AT1 cells[15]. Due to their progenitor role in distinct regions of the lung, club and AT2 cells are frequently targeted in lung regeneration studies using naphthalene[22–24] and bleomycin[14,21,25], respectively.

Lung epithelial regeneration is also dependent on cues from the microenvironment[18]. For example, it has been shown that airway smooth muscle cells promote club cell repair by secreting FGF10[24], and a population of *Pdgfrα*+ fibroblasts in the mouse trachea induced differentiation of basal cells through the production of IL6[26]. In the distal lung, *Axin2*+ *Pdgfrα*+ fibroblasts support AT2 cells through Il6, Bmp, and Fgf signaling modulation[27]. Using single-cell RNA sequencing (scRNA-seq), Tsukui et al. recently classified collagen-producing cells in the mouse and human lung as peribronchial fibroblasts, adventitial fibroblasts, alveolar fibroblasts, pericytes, and smooth muscle cells, but their role in epithelial regeneration is unclear[28].

In spite of recent progress, there are still knowledge gaps on how the different lung cell types function during regeneration and whether there are any unknown cell types. In this study, we used scRNA-seq to analyze the consequences of epithelial cell type-specific depletion through inducible expression of diphtheria toxin subunit A (DTA) in the mouse distal lung, which has the potential to identify cell types or transition states that may be missed due to toxic effects after applying chemical or infectious agents. We combined this with a GFP-labeling approach to trace dividing cells in vivo, and applied a sorting strategy to enrich for rare cell types. This approach allowed us to discover epithelial populations with potential progenitor cell properties, alternative differentiation trajectories, and adventitial fibroblasts as supporting cells during epithelial regeneration. We propose that club, AT2, and ciliated cells participate in the damage-induced inflammatory response, and we show that viable DTA-expressing cells can persist in the lung and express a specific transcriptional profile that reveals a rare epithelial population in the lungs of COVID-19 patients.

## Results

### *Scgb1a1*+ cell loss activates several cell populations in the distal lung

To better understand which cells play an active role in lung epithelial regeneration, we generated a mouse model that enables precise depletion of specific epithelial cell types and the isolation and detection of dividing cells. Specifically, we crossed Scgb1a1-CreER x Rosa26R-DTA mice[9], in which tamoxifen administration induces DTA expression and subsequent apoptosis in *Scgb1a1*-expressing cells, with CycB1-GFP transgenic mice, which allows the detection and isolation of actively dividing cells throughout the body (including lung) in a functional manner[29] (Fig. 1a). Specifically, these mice constitutively express CycB1-GFP, a fusion protein of the N-terminal portion of cyclin B1 and eGFP that behaves like full-length cyclin B1 in terms of expression during the cell cycle, i.e. it is degraded in G0/G1 phases so that cells are GFP negative (GFP-), and it is stable in S/G2/M stages rendering cells GFP positive (GFP+). *Scgb1a1* (also known as *CC10*) is physiologically expressed by club cells and BASCs[9]. Scgb1a1-CreER x

Rosa26R-DTA x CycB1-GFP (SRC) mice were injected with tamoxifen, and lungs were harvested before (day 0) and 2 and 3 days afterwards. Immunofluorescence (IF) analysis confirmed the progressive loss of club cells and revealed an increase in the number of GFP+ dividing cells over time (Fig. 1b). On day 2, the majority of GFP+ cells were located in the airway epithelium, consisting of club cells that were not yet damaged but replicated to repair the epithelium. On day 3, dividing GFP+ cells appeared in the airways, underlying connective tissue, and alveoli (Fig. 1b).

To further characterize the heterogeneous population of proliferating cells that may contain stem or progenitor cells, we performed scRNA-seq of viable (DAPI-) GFP+ cells isolated by fluorescent-activated cell sorting (FACS) from the distal lung of SRC mice at day 2 and 3 after tamoxifen injection (Fig. 1c). After excluding endothelial and hematopoietic cells, GFP+ cells represented 0.034% and 0.16% of total cells on day 2 and 3, respectively (Supplementary Fig. 1a). After removing low-quality cells (see Supplementary Data 1 for filtering parameters), in silico cell cycle analysis confirmed that the majority of sorted GFP+ cells were dividing, with 42% in G2/M and 44% in S phase (Fig. 1d), including both epithelial (*Epcam*+) and mesenchymal (*Col1a1*+) cells (Fig. 1e). Proliferating mesenchymal cells were identified as adventitial (*Dcn*+) and alveolar (*Inmt*+) fibroblasts (Fig. 1f, g). Five of the seven epithelial cell clusters could be allocated to distinct cell types based on the expression of known marker genes: AT2, basal, club, ciliated, and goblet cells (Fig. 1f, g and Supplementary Data 2). IF confirmed the transcriptomic results identifying proliferating club, goblet, basal, and AT2 cells, as well as adventitial and alveolar fibroblasts (Supplementary Fig. 1b). The small population of non-dividing ciliated cells (Fig. 1f) was isolated during FACS due to high auto-fluorescence. Basal cells represented 39% of dividing cells on day 2, while AT2 cells accounted for the majority (73 %) of dividing cells on day 3 (Fig. 1h). This was unexpected since AT2 cells are only known to function as progenitors for AT2 and AT1 cells upon alveoli injury[2,19].

Among the differentially expressed genes (DEGs) in dividing club (club_div) cells compared to all other dividing cell types was the club cell marker *Krt7* (Fig. 1g)[30], but other canonical club cell markers such as *Scgb3a2* or *Cyp2f2* were not consistently expressed (Supplementary Fig. 1c). In fact, lower expression of these genes was associated with higher expression of AT2 marker genes including *Sftpc*, *Napsa*, and *Lamp3*, suggesting priming of club_div cells for differentiation into AT2 cells (Fig. 1g and Supplementary Fig. 1c). Club_div cells showed no similarity to reported club cell progenitor populations including lineage-negative epithelial progenitors (*H2-K1*+, *Sox2*+ *Itgb4*+), variant club cells (*Scgb3a2*high, *Cyp2f2*-, *Upk3a*+), or hillock club cells (*Krt13*+, *Krt4*+) (Supplementary Fig. 1c)[23,31,32]. Moreover, although 37% of club_div cells at day 3 co-expressed *Sftpc* and *Scgb1a1*, a characteristic feature of BASCs, they did not consistently express additional BASC marker genes described previously (Supplementary Fig. 1d)[13,33]. Additionally, *Sftpc*+ *Scgb1a1*+ cells were transcriptionally similar to the other club_div cells (p_adj <0.05, min.pct=0.2, test.use= "MAST"), suggesting that either most BASCs were destroyed by DTA, or that they were not proliferating at the timepoints analyzed.

We found that goblet cells (*Agr2*+, *Ltf*+, *Scgb3a1*+, *Lypd2*+), which also expressed *Scgb1a1* (Supplementary Fig. 1e, Supplementary Data 2), were dividing in the injured lung (Fig. 1f–h and Supplementary Fig. 1b). This was surprising, since goblet cells have never been shown to be capable of self-renewal[19,34]. Additionally, a cell type expressing both basal (*Krt5*, *Krt15*) and goblet (*Agr2, Lypd2, Ltf*) cell markers was among the dividing cells (Fig. 1f, g, *n* = 56 cells). We named this population basal-goblet_div, which might represent an intermediate cell state between goblet and basal cells, and we confirmed that it was not an artifact caused by doublets (Supplementary Fig. 1f).

Finally, the remaining seventh cluster of dividing epithelial cells expressed DTA, indicating the presence of lung cells that survived the expression of this highly potent toxin (Supplementary Fig. 1g). These

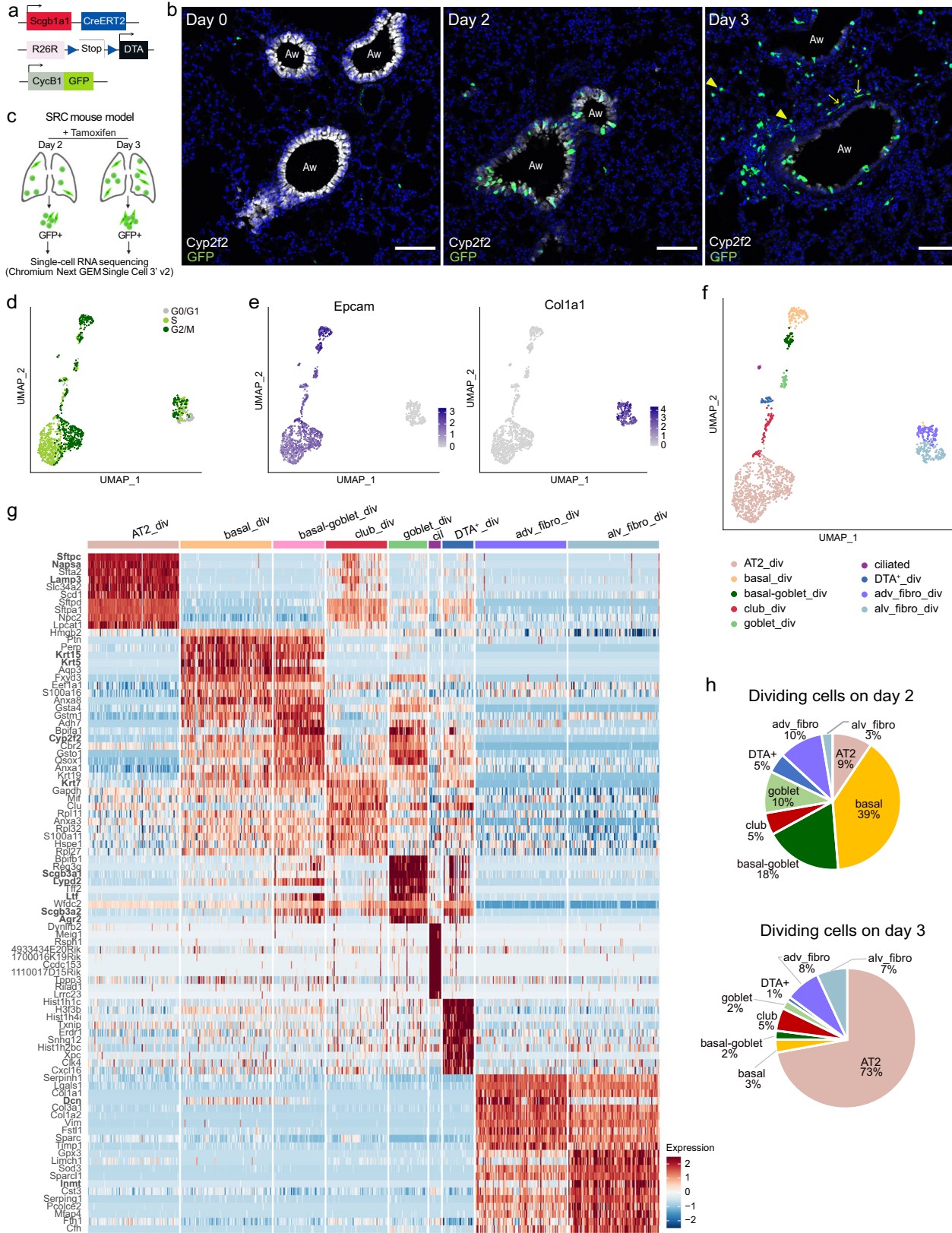

DTA⁺ cells clustered separately from club_div and goblet_div cells and had a distinct gene signature, although they expressed secretory cell genes such as *Scgb3a2* (Fig. 1f, g and Supplementary Data 2). Importantly, DTA⁺ cells did not have a higher percentage of mitochondrial genes or a lower number of counts or genes per cell, supporting the notion that these are viable cells (Supplementary Fig. 1h).

In summary, *Scgb1a1*⁺ cell loss induces widespread cellular activation with immediate proliferation of several populations in the lung rather than triggering a specific cell type. Among the dividing cells were goblet, basal-goblet, and AT2-primed club cell populations, which might represent progenitors or intermediate cell states in the distal lung.

**Fig. 1 | Characterization of dividing cells after targeted depletion of _Scgb1a1_⁺ cells. a** Schematic of Scgb1a1-CreER, Rosa26R-DTA, and CycB1-GFP transgenes in SRC mice. **b** IF staining of SRC mouse lungs with Cyp2f2 (white, club cells), GFP (green, dividing cells), and DAPI (blue, nuclei), before (day 0) and after 2 and 3 days of a single tamoxifen injection. GFP⁺ alveolar cells (arrow heads) and GFP⁺ spindle-like cells (arrows) can be observed on day 3. Aw: airway. Scale bar: 100 μm. Images are representative of five independent experiments with _n_ = 3 animals analyzed per timepoint. **c** Scheme of cell sorting and scRNA-seq strategy.

Created with BioRender.com. **d**–**f** UMAP embedding of scRNA-seq data from GFP⁺ cells sorted from lungs of SRC mice two days (_n_ = 3 mice) and three days (_n_ = 2 mice) after tamoxifen injection. **d** Cell cycle phase distribution. **e** Normalized expression of _Epcam_ (epithelial cells) and _Col1a1_ (mesenchymal cells). **f** Cell type assignment. **g** Heatmap of the top ten upregulated genes across cell populations ranked by power (roc test). Scaling of expression was done after downsampling to 100 cells per cell type. DEGs mentioned in the text are in bold. **h** Percentage of dividing cell types in the distal lung at day 2 and 3 after tamoxifen injection.

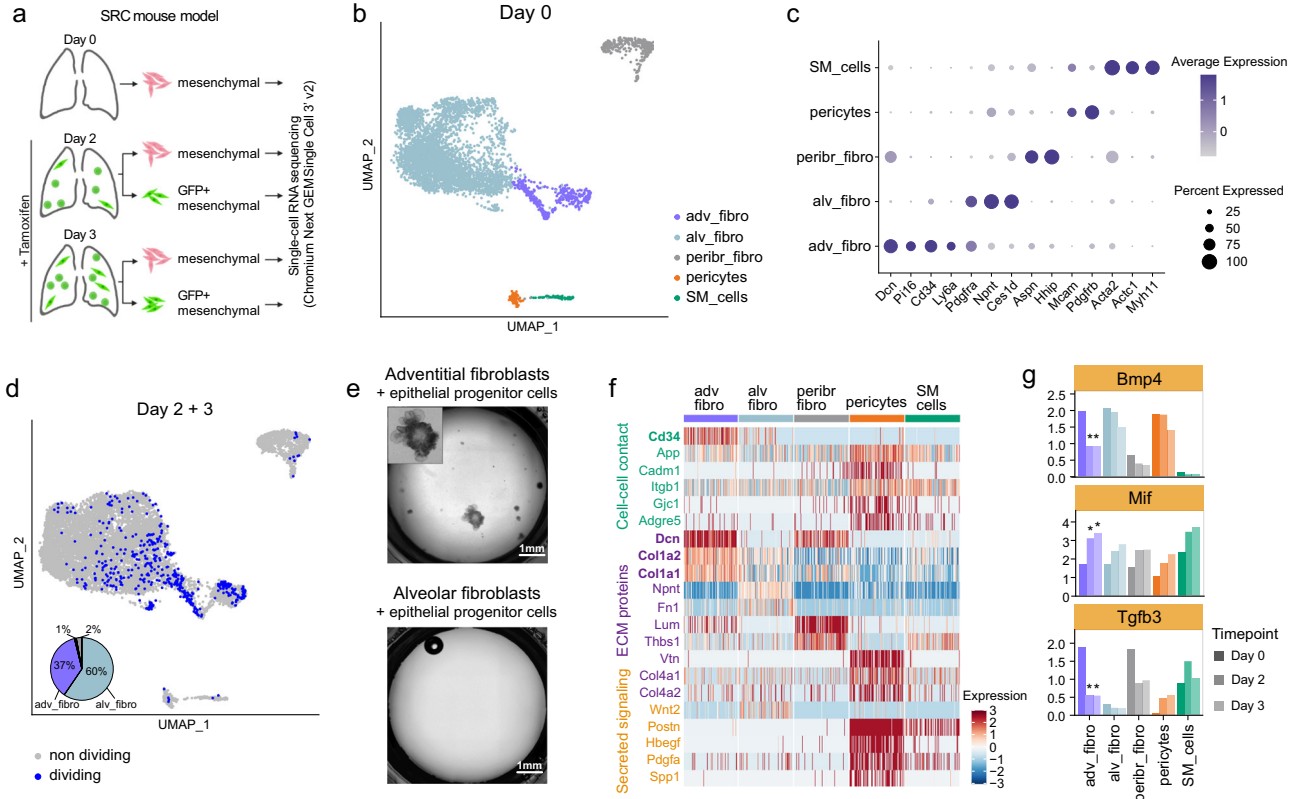

**Fig. 2 | Characterization of mesenchymal cell activation after epithelial injury in SRC mice. a** Scheme of cell sorting and scRNA-seq strategy. Created with BioRender.com. **b** UMAP embedding showing mesenchymal cell types sorted from uninjured SRC mouse lungs (_n_ = 2 mice). **c** Dot plot showing the average scaled expression and percentage of cells expressing cell-type specific marker genes across the mesenchymal cell populations in uninjured lungs. **d** UMAP embedding showing the distribution of GFP⁺-sorted dividing mesenchymal cells on day 2 (_n_ = 12 mice) and day 3 (_n_ = 4 mice), and non-dividing total cells on day 2 (_n_ = 3 mice) and day 3 (_n_ = 2 mice). The pie chart depicts the percentage of the mesenchymal cell types among the diving cells (light blue: alveolar fibroblasts, purple: adventitial fibroblasts, gray: peribronchial fibroblasts, orange: pericytes, green: SM cells).

**e** Lung epithelial organoid cultures with adventitial fibroblasts or alveolar fibroblasts as supporting cells, taken after three weeks of culture. The inset in the upper image shows the morphology of one organoid magnified twice as much as in the main figure. Images are representative of two independent experiments. **f** Heatmap of differentially expressed ligands among the different mesenchymal cell types on day 0. Upregulated genes in adventitial fibroblasts are bold. **g** Average normalized expression of _Bmp4_, _Tgfb3_, and _Mif_ on day 0, 2, and 3 in all mesenchymal populations. Asterisks denote significant differential expression on day 2 or 3 compared to day 0, when considering FC ≥ 1.5 and _p_\_adj ≤ 0.05 (MAST test, all DEG had a _p_\_adj < 0.001). adv_fibro adventitial fibroblasts, alv_fibro alveolar fibroblasts, peribr_fibro peribronchial fibroblasts, SM_cells smooth muscle cells.

## Adventitial fibroblasts support lung epithelial regeneration

Several mesenchymal cell types are known to support epithelial regeneration upon lung injury[24,26,27]. Conversely, mesenchymal cells can also give rise to aberrant populations following lung injury, such as _Axin2_⁺ _Pdgfrα_⁻ myogenic progenitors[27] and _Cthrc1_⁺ fibroblasts[28], which contribute to fibrosis. To analyze the behavior of mesenchymal cells during loss of _Scgb1a1_⁺ cells, we performed scRNA-seq of actively dividing (GFP⁺) and non-dividing mesenchymal cells from the distal lung of SRC mice (Fig. 2a).

In the uninjured lung, we identified adventitial fibroblasts, alveolar fibroblasts, peribronchial fibroblasts, pericytes, and smooth muscle cells (Fig. 2b, c and Supplementary Data 3), annotated according to Tsukui et al. [28]. The types of mesenchymal cells remained unchanged upon injury (Supplementary Fig. 2a), and none of the post-injury

mesenchymal cell populations showed increased fibrosis-associated genes, suggesting that this mouse model does not induce lung fibrosis (Supplementary Fig. 2b)[28,35]. Analysis of the GFP⁺ cells confirmed our previous observation (Fig. 1e–h) that alveolar and adventitial fibroblasts are the major proliferating mesenchymal cell types after _Scgb1a1_⁺ cell depletion, accounting for 60% and 37% of the total mesenchymal dividing cells, respectively (Fig. 2d). Since alveolar fibroblasts were approximately eight times more abundant than adventitial fibroblasts, the high proportion of adventitial fibroblasts in the total dividing cells suggests a higher activation and a possible functional role of these cells. To test this, we performed organoid cultures with epithelial progenitor cells (EPCAM^high CD24^dim)[36] in co-culture with adventitial fibroblasts (PDGFRA⁺ CD34⁺ SCA-1⁺) or alveolar fibroblasts (CD34⁻ SCA-1⁻ PDGFRA⁺ NPNT⁺) (Supplementary Fig. 2c).

After three to four weeks of culture, alveolar and bronchoalveolar organoids, distinguished by their morphology, were formed with adventitial fibroblasts as supporting cells (mean, 29 organoids per well (range: 24–35); organoid size, 0.125–1.075 mm in diameter), but not with alveolar fibroblasts (Fig. 2e), reinforcing our hypothesis.

To learn more about the supporting role of adventitial fibroblasts, we examined the expression of genes encoding cell-cell contact proteins, extracellular matrix proteins, and secreted ligands from the CellChat ligand-receptor database[37]. Adventitial fibroblasts differentially expressed *Cd34*, *Dcn*, *Col1a2*, and *Col1a1* in uninjured lungs (Fig. 2f), which was confirmed on protein level by mass spectrometry for CD34 and DCN, while COL1A1 was also high in alveolar fibroblasts (Supplementary Fig. 2d). After injury, adventitial fibroblasts significantly downregulated mRNAs of the secreted ligands *Bmp4* and *Tgfb3* (Fig. 2g). These signaling proteins are known regulators of the lung epithelium. For example, inhibition of BMP signaling induces proliferation of lung epithelial cells in vitro and in vivo[38] and TGF-β has a cytostatic effect on lung epithelial cells[39]. Adventitial fibroblasts also significantly upregulated several pro-inflammatory factors after *Scgb1a1*[+] cell depletion (Fig. 2g and Supplementary Fig. 2e), corroborating the communication between mesenchymal and immune cells in the lung[40]. Among those was *Mif* (Fig. 2g), a pro-inflammatory factor that was reported to stimulate AT2 cell proliferation in vitro[41,42].

In summary, our data indicate that adventitial fibroblasts support epithelial repair as they proliferate extensively, produce pro-inflammatory factors, downregulate secreted ligands known to be cytostatic on lung epithelial cells, and support epithelial organoid growth in culture.

## Post-injury transcriptional changes imply immune activation by club, AT2, and ciliated cells

A deeper understanding of the molecular changes in epithelial cells during lung regeneration can aid the development of therapies for lung diseases. We therefore characterized the cellular and transcriptional responses by scRNA-seq in dividing and non-dividing epithelial cells, sorted as viable (DAPI[-]) Epcam[+] cells from distal lungs of SRC mice without treatment and two, three, and four days after tamoxifen administration (Fig. 3a). During sorting, the very abundant AT2 cell numbers were reduced by approximately 90%, while all GFP[+] epithelial cells were enriched (Fig. 3a and Supplementary Fig. 3a). The steady post-injury increase of GFP[+] dividing cells was confirmed by FACS and IF (Supplementary Fig. 3a, b).

We identified AT1, AT2, basal, club, goblet, ciliated, and DTA[+] cells (Fig. 3b, Supplementary Fig. 3c, d, and Supplementary Data 4). Basal, club, and goblet cells were again the main cell types dividing early after epithelial injury, while AT2 cells became the dominant dividing cell type as injury progressed (Fig. 3b). We then characterized the transcriptional changes between the cells isolated from uninjured lungs and the cells at the three time points after tamoxifen injection for each cell type (Supplementary Data 5). In goblet, basal, and AT2 cells, the top-upregulated DEGs were predominantly associated with cell cycle (Supplementary Fig. 3e and Supplementary Data 5). While basal cells increased expression of these genes as early as day 2, their expression in AT2 cells did not increase until day 3, which is in accordance with AT2 cells starting to divide later (Supplementary Fig. 3f and Supplementary Data 5). In addition, AT2 cells also displayed an upregulation of genes associated with immune response and inflammation, such as *Il33*, *Chia1*, *Cd14*, *Lgasl3*, *Ly6e*, and *Ly6c1*, suggesting a role of these cells in immune activation (Supplementary Fig. 3f).

Club cells showed the highest number of DEGs (FC > 2, *p*_adj <0.05) post-injury at all time points, reaching more than 300 DEGs on day 4 (Supplementary Fig. 3g). Upregulated DEGs were mainly related to cell cycle, cytoplasmic ribosomal proteins, protein folding, mRNA processing, and genes associated with leukocyte chemotaxis (e.g.

*Ccl20*, *Cxcl15*, *Cxcl5*), the latter suggesting a role of club cells in immune activation (Fig. 3c, d and Supplementary Data 5).

Ciliated cells elevated the expression of interferon-stimulated genes (Fig. 3e, f and Supplementary Data 5), which are involved in viral response mechanisms and are crucial in driving and maintaining lung inflammation, for example, by promoting antigen presentation or stimulating cytokine production[43]. Ciliated cells also upregulated genes implicated in antigen processing and presentation, including MHC class II molecules (MHCII), which are normally found exclusively in professional antigen-presenting cells[44]. Unlike AT2 cells, which express high levels of MHCII independent of inflammatory stimuli (Fig. 3g)[45], ciliated cells demonstrated very low expression of these molecules in homeostasis (Fig. 3g). Thus, the upregulation of MHCII molecules in ciliated cells after epithelial damage suggests that they might function as antigen-presenting cells upon lung epithelial injury.

Together, these data suggest that club, AT2, and ciliated cells may contribute to the initiation of an inflammatory response following lung injury.

## The DTA[+] cell transcriptome reveals a COVID-19 lung cell population

Genetic injury mouse models using inducible DTA expression have been widely used since they allow for rapid and controlled cell type-specific ablation[46]. We found that viable DTA[+] cells can persist in the injured tissue and express a distinct gene expression signature (Fig. 1g), which may provide insights into mechanisms of cellular response to pathological conditions. To characterize these cells in more detail, we further enriched rare cell types by partially depleting the most abundant cell types, i.e. AT2 and ciliated cells, by ~90%, and used a more sensitive 3′ RNA sequencing protocol (Fig. 4a and Supplementary Fig. 4a). In addition, GFP[+] epithelial cells were sorted in their totality in samples from day 0 and day 2 and added before sequencing (Fig. 4a).

Unsupervised clustering revealed the same dividing and non-dividing epithelial cell types as described above, neuroendocrine cells, and five clusters of DTA[+] cells (Fig. 4b, c and Supplementary Fig. 4b). Again, DTA[+] cells did not have a higher percentage of mitochondrial genes or a lower number of counts or genes per cell, supporting the notion that these are viable cells (Supplementary Fig. 4c). Tamoxifen-treated Rosa26R-DTA x CycB1-GFP control mice lacked DTA[+] cells (Supplementary Fig. 4d) and had unchanged lung morphology, number of GFP[+] dividing cells, and number and distribution of club and AT2 cells compared with normal lung (Supplementary Fig. 4e), ruling out non-specific tamoxifen effects.

To determine the cells of origin or the closest relatives of the DTA[+] populations, we calculated the correlation coefficient of gene expression between the different cell types on day 2 (Supplementary Fig. 4f). One DTA[+] cell cluster showed high expression of AT2 canonical marker genes (Fig. 4d). We hypothesized that these cells, which we named DTA[+]_Sftpc[+], were either derived from a *Scgb1a1*[+] AT2 subpopulation[9] and could therefore activate DTA expression (Supplementary Fig. 4g), or from club cells that differentiated into AT2 cells[47]. Two clusters transcriptionally resembled secretory cells and were accordingly named DTA[+]_club and DTA[+]_goblet, and DTA[+] dividing cells included a mixture of cells resembling goblet and club cells, so we refer to them as DTA[+] secretory dividing (DTA[+]_sec_div) (Fig. 4d and Supplementary Fig. 4f). The remaining cluster, which we termed DTA[+]_Foxj1[+], expressed ciliated cell genes (Fig. 4d and Supplementary Fig. 4f) and likely evolved from club cells that differentiated into ciliated cells[9]. A complete list of the marker genes for each population at day 2 is provided in Supplementary Data 6.

Besides their cell type-specific gene signatures, the DTA[+] populations shared gene expression patterns and had several marker genes in common (Fig. 4d). The co-upregulated genes showed a prominent

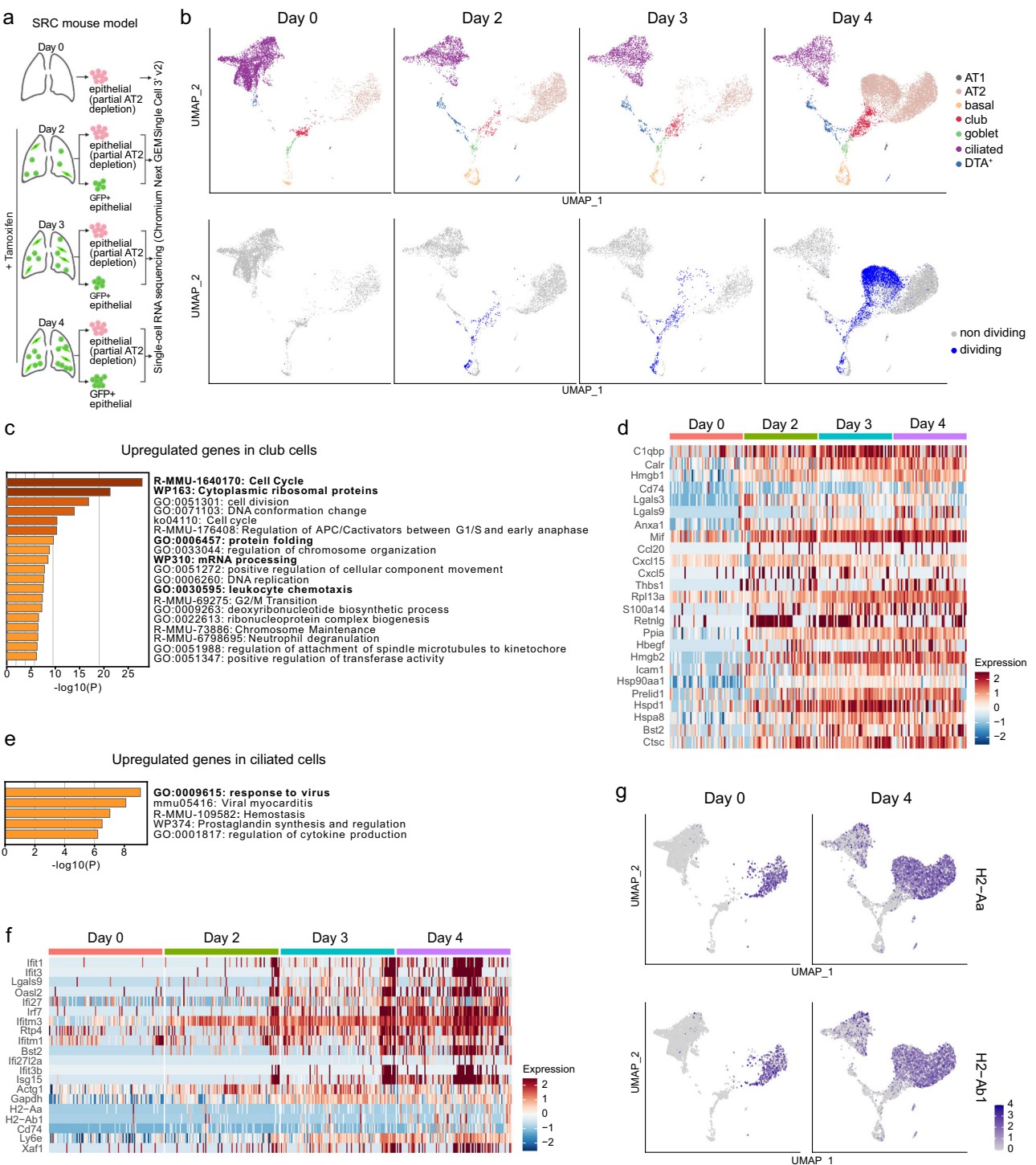

**Fig. 3 | Transcriptional changes in epithelial lung cells after *Scgb1a1*⁺ cell depletion.** scRNA-seq analysis was done using lung epithelial cells from SRC mice harvested on day 0 (n = 2 mice), day 2 (n = 7 mice), day 3 (n = 5 mice), and day 4 (n = 5). **a** Scheme of cell sorting and scRNA-seq strategy. Created with BioR-ender.com. **b** UMAP embedding showing epithelial cell type (upper panels) and dividing and non-dividing epithelial cells (lower panels) on the indicated days. **c** Gene set enrichment analysis (GSEA) of upregulated genes in club cells on day 2, 3, and 4 compared to day 0 (MAST test, FC > 2, *p*_adj <0.05). The terms mentioned in the text are shown in bold. **d** Heatmap of genes from the"leukocyte chemotaxis" cluster of the GSEA in **c** showing the expression in club cells. **e** GSEA of upregulated genes in ciliated cells on day 2, 3, and 4 compared to day 0 (MAST test, FC > 2, *p*_adj < 0.05). The terms mentioned in the text are shown in bold. **f** Heatmap of genes from the"response to virus" cluster of the GSEA in **e** showing the expression in ciliated cells. **g** UMAP embedding showing the expression of H2-Aa and H2-Ab1 on day 0 and day 4 (p_ adj <0.05 in ciliated cells with MAST test).

enrichment in mRNA processing genes, including mRNA splicing-related genes (Fig. 4e and Supplementary Data 7). In addition, DTA⁺ populations displayed upregulation of genes involved in ribonucleo-protein complex biogenesis, cellular response to stress, induction of apoptosis, and downregulation of protein processing-related genes (Fig. 4e and Supplementary Data 7), which could be a consequence of DTA-mediated protein synthesis inhibition[48]. Curiously, all DTA⁺ populations, apart from DTA⁺_Foxj1⁺ cells, displayed increased levels of

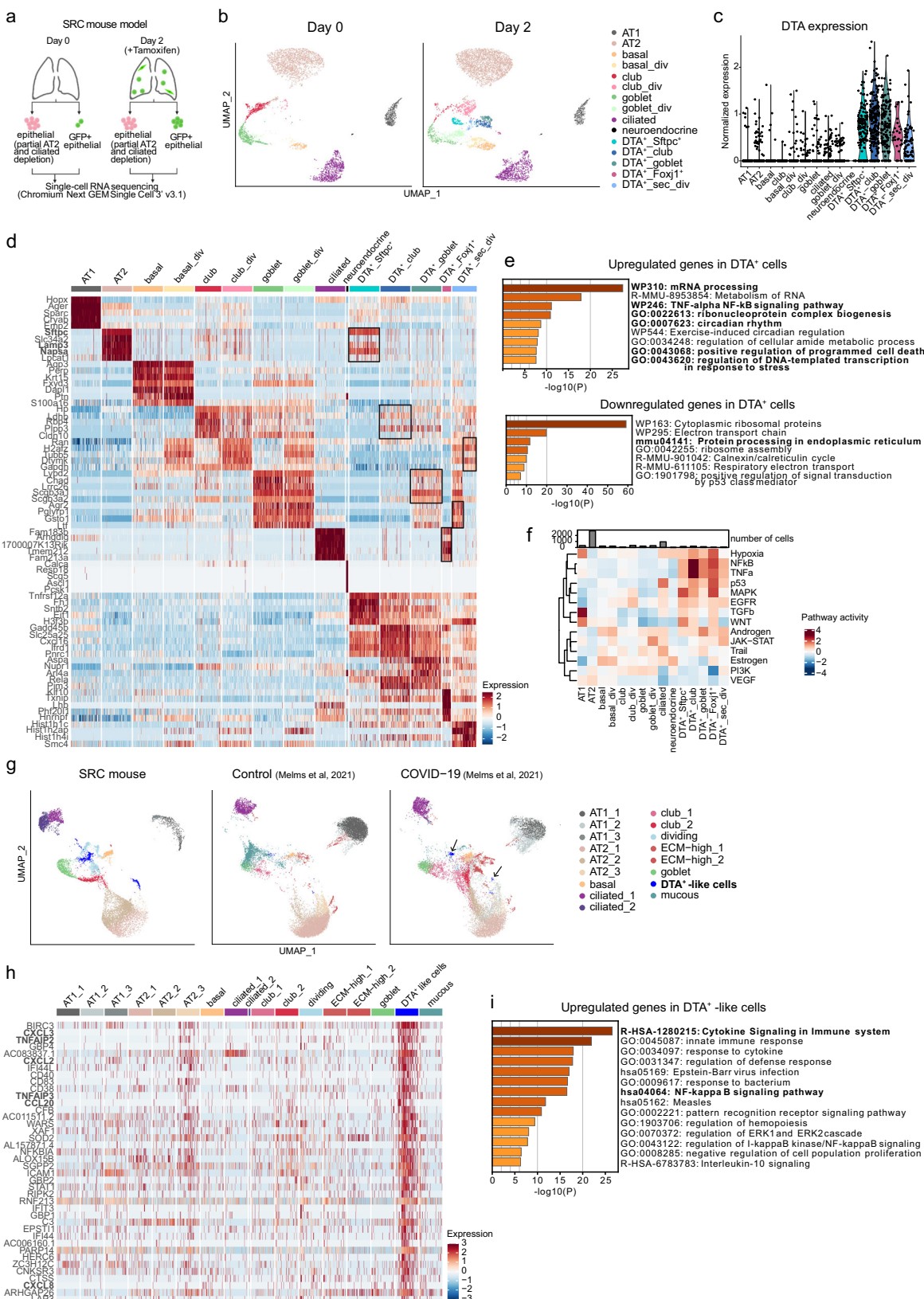

*Cftr*, a transmembrane conductance regulator that is mutated in cystic fibrosis (Supplementary Fig. 4h). This gene is highly expressed by ionocytes found in the proximal airways of human and mice[11,31]. However, DTA+ cells did not express other ionocyte marker genes, implying that these are different cell types (Supplementary Fig. 4h). Finally, DTA+ cells showed an enrichment in genes involved in pro-

inflammatory TNF-alpha/NF-κB signaling and circadian rhythm, which could influence the inflammatory response[49] (Fig. 4e and Supplementary Data 7). A complementary pathway activity analysis based on core pathway responsive genes confirmed that DTA+ populations had increased activation of several inflammatory pathways such as TNF, NF-κB, MAPK, and EGFR (Fig. 4f).

**Fig. 4 | Characterization of DTA⁺ cells and comparison to lung cells from COVID-19 patients. a** Scheme of cell sorting and scRNA-seq strategy. Created with BioRender.com. **b** UMAP embedding showing epithelial cell type assignment before (day 0, *n* = 8 mice) and after tamoxifen administration (day 2, *n* = 10 mice). **c** Violin plot with normalized DTA expression in all epithelial cells. **d** Heatmap of the top five upregulated genes across epithelial cell types on day 2 ranked by power (roc test). Scaling of expression was done after downsampling to 100 cells per cell type. Black boxes highlight the expression of the marker genes of the closest epithelial population in DTA⁺ cells. **e** GSEA of common DEGs in DTA⁺ populations in relation to all non-DTA⁺ cells (MAST test, *p*_adj <0.05). The terms mentioned in the text are shown in bold. **f** Heatmap of the activities of 14 pathways inferred with PROGENy for all epithelial populations on day 2. **g** UMAP embedding showing cell type assignment of COVID-19 and control human epithelial cells integrated with SRC mouse epithelial cells (day 0 and day 2). Arrows point to DTA⁺-like cells present in the COVID-19 patient samples. **h** Heatmap of the top 40 DEGs in DTA⁺-like cells from COVID-19 epithelial cells. **i** GSEA of upregulated genes in the DTA⁺-like cells (MAST test, *p*_adj < 0.05). The terms in bold depict the clusters that are mentioned in the text.

Considering these specific transcriptional changes in DTA⁺ cells, we hypothesized that they might mirror characteristics of damaged cells in the context of human lung diseases. Using gene signatures from DisGeNET[50] and MSigDB[51,52] databases for several human lung diseases, we found that DTA⁺ cells had particularly high enrichment for genes expressed in epithelial lung cells from COVID-19 patients[53] (Supplementary Fig. 4i). Integration of our murine dataset with a human scRNA-seq dataset from a COVID-19 study[4] revealed an epithelial cell population unique to COVID-19 samples, which we named DTA⁺-like cells (Fig. 4g and Supplementary Fig 4j–l). This cell population showed differential upregulation of genes involved in TNF and NF-κB signaling, supporting the similarity to DTA⁺ cells (Fig. 4h, i and Supplementary Data 8).

Together, epithelial cells expressing DTA can persist in the lung and share a distinct gene expression profile that exhibits activation of inflammatory signaling pathways and remarkable transcriptional similarity to a previously undescribed lung cell population in patients with COVID-19.

## DTA⁺ cells signal to immune, epithelial, and mesenchymal cells

Inflammatory pathways, such as the ones described above, can be activated in epithelial cells by microbial components and lead to the initiation of an immune response through chemokine secretion[54]. Upregulated genes in the COVID-19 DTA⁺-like population were enriched for genes related to cytokine signaling in the immune system (Fig. 4i), and several chemokines were among the top 40 DEGs (Fig. 4h), suggesting that the identified COVID-19 cell population can contribute to the inflammatory response in these patients. To investigate this further, we used the murine DTA⁺ cells as a model for the human COVID-19 cell population and investigated their potential role in immune cell activation with focus on chemokines.

After *Scgb1a1*⁺ cell depletion, chemokine expression increased only slightly in regular epithelial cells, with the exception of *Ccl20* in club cells and *Cxcl17* in club_div and ciliated cells (Fig. 5a and Supplementary Fig. 5a). In contrast, DTA⁺ cells were a major source of chemokine production, and DTA⁺_club and DTA⁺_Foxj1⁺ even dedicated a large proportion of their total transcriptome to chemokine production (2% in DTA⁺_club at day 2 vs. 0.7% in club cells at day 0; 0.4% in DTA⁺_Foxj1⁺ at day 2 vs. 0.1% in ciliated cells at day 0) (Fig. 5a and Supplementary Fig. 5b). Among the chemokines significantly upregulated in at least one DTA⁺ cell type, but not in any regular epithelial cell type, were *Ccl9*, *Ccl20*, *Cxcl1*, *Cxcl2*, *Cxcl3*, *Cxcl10*, and *Cxcl16* (Fig. 5b). *Ccl9*, *Cxcl2*, and *Cxcl16* were shown to be expressed in immune cells during homeostasis, whereas *Ccl20*, *Cxcl1*, *Cxcl3*, and *Cxcl10* are barely expressed in any cell type of the lung under healthy conditions (Supplementary Fig. 5c)[55], suggesting that they are specifically produced under damage-inducing conditions. These DTA⁺-specific chemokines are known to recruit neutrophils (CXCL1, CXCL3), dendritic cells (CCL20), T cells (CCL20, CXCL10), and B cells (CCL20)[56].

To characterize the interactions between epithelial and mesenchymal cells, we analyzed the ligand-receptor expression of all epithelial and mesenchymal cells in lungs of SRC mice at day 0 and day 2. After injury, new or increased signal transmission occurred in several pathways (Fig. 5c), mainly as a result of new ligand-receptor pairings involving DTA⁺ epithelial cells (Supplementary Fig. 5d, e). Specifically, signaling networks involved in the formation of tight junctions such as MARVELD, OCLN, JAM, and CLDN were activated in epithelial cells, with particularly high levels in DTA⁺ cell types (Fig. 5c and Supplementary Fig. 5f). Tight junctions are crucial during lung repair as they are responsible for sealing the epithelial barrier and preventing the entry of pathogens and the free diffusion of solutes into airspace[57]. *Edn1* was also specifically expressed in DTA⁺ epithelial cells and was predicted to signal to alveolar fibroblasts, adventitial fibroblasts, smooth muscle cells, and pericytes (Fig. 5d and Supplementary Fig. 5f, g). EDN1 was shown to stimulate fibroblast replication, migration, and collagen synthesis[58], suggesting a role of DTA⁺ cells in the activation of mesenchymal cells after epithelial injury. Finally, we found that DTA⁺ cells highly expressed several ligands that can signal to other epithelial cells and support lung regeneration (Fig. 5e–g and Supplementary Fig. 5f, g). Specifically, IL6, predicted to signal from DTA⁺_Sftpc⁺ cells to AT1, basal, goblet, and ciliated cells, was found to be crucial for lung repair after influenza-induced lung injury in mice[59]. LIFR pathway, which was previously shown to protect lung tissue during pneumonia in mice[60], was predicted to be activated in AT2 and club cells by DTA⁺ cells through the expression of LIF. DTA⁺ cells also expressed high levels of *Areg* and *Hbegf* that can signal to basal cells through EGFR, previously shown to be essential for basal cell proliferation[61].

Overall, DTA⁺ cells express specific chemokines and growth factors that can trigger an immune response and contribute to lung regeneration through epithelial and mesenchymal cell support.

## *Scgb1a1*⁺ and *Sftpc*⁺ cell loss unveils *Krt13*⁺ basal and *Krt15*⁺ club cells

We next characterized the consequences of increased epithelial damage in the peripheral lung through additional injury of AT2 cells. Therefore, we crossed SRC mice with mice expressing tamoxifen-inducible Cre recombinase in AT2 cells under the *Sftpc* promoter[25]. The resulting Sftpc-CreER x Scgb1a1-CreER x Rosa26R-DTA x CycB1-GFP (SfSRC) mice (Fig. 6a) allow simultaneous damage of *Scgb1a1*⁺ and *Sftpc*⁺ cells after tamoxifen administration, thus of both bronchiolar and alveolar progenitor cells, and parallel analysis of GFP⁺ dividing cells. IF analysis showed that *Scgb1a1*⁺ and *Sftpc*⁺ cells progressively disappeared from day 2 to 14, and numerous dividing cells emerged in the airways, underlying tissue, and alveoli (Fig. 6b). Airway regeneration was not complete at day 28 based on cell morphology and Cyp2f2 club cell staining, despite only few dividing cells remaining, but was finally achieved at day 56 (Fig. 6b). Lung morphology on days 14, 28, and 56 showed no evidence of fibrosis (Fig. 6b), and Masson staining at day 14 confirmed the absence of collagen deposition (Supplementary Fig. 6a). A mixed inflammatory infiltrate composed mainly of lymphocytes and plasma cells and a few neutrophil granulocytes, with perivascular and peribronchial accentuation, was observed on day 4. This pattern became more pronounced on day 14, and, in addition, the lungs exhibited an accumulation of intra-alveolar macrophages in the form of foam cells, which was absent in controls (Supplementary Fig. 6b).

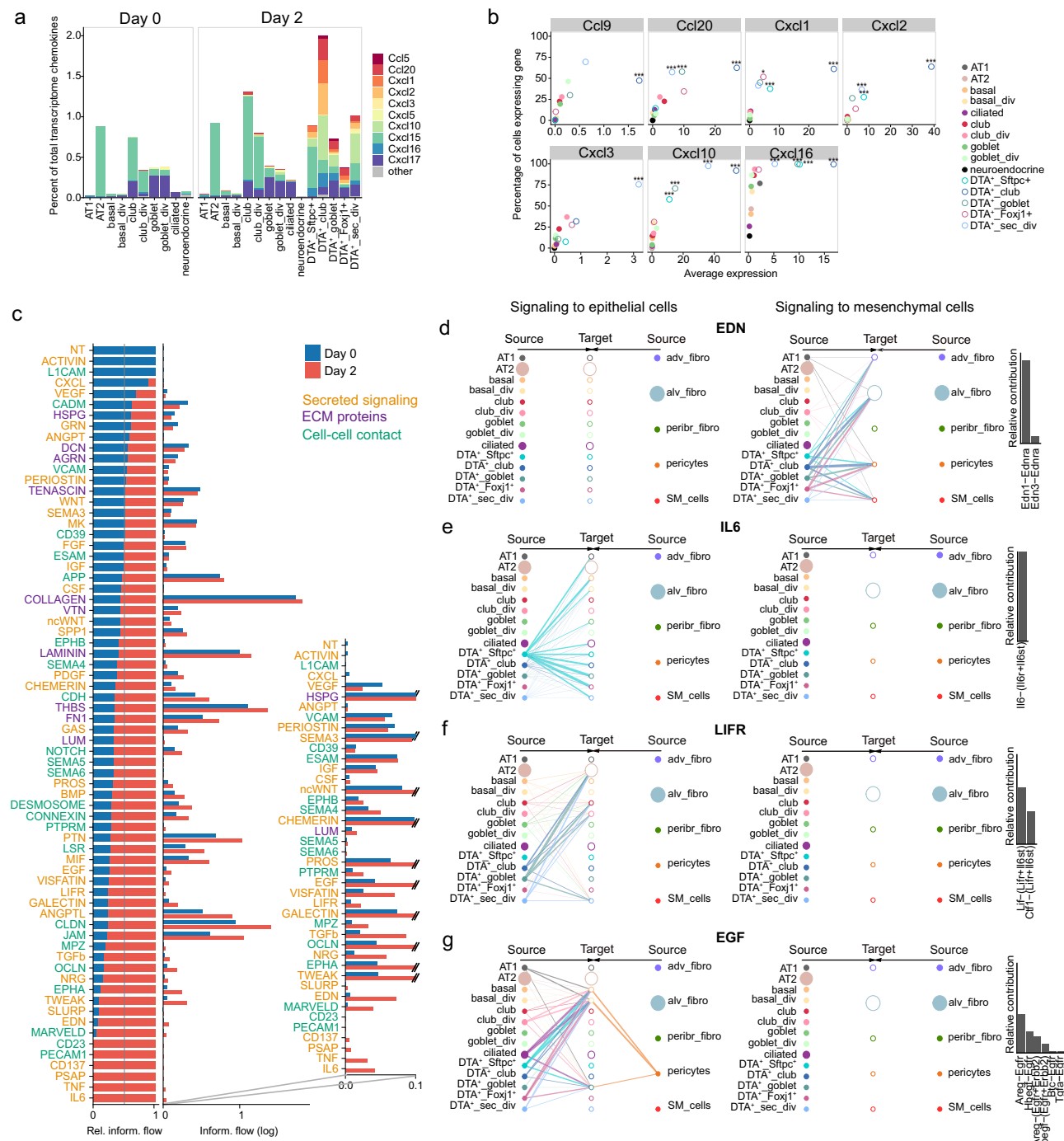

**Fig. 5 | Chemokine expression and crosstalk to mesenchymal and epithelial cells of DTA⁺ cells. a** Chemokine expression in epithelial lung cells from SRC mice before (day 0) and after injury (day 2). Genes with a mean expression of at least three counts per 10,000 in any cell type are colored. **b** Expression of chemokines upregulated in DTA⁺ cells compared to other epithelial cells after injury (MAST test, FC > 2 and p_adj < 0.05 in any DTA⁺ cell type). Asterisks denote p_adj value for significant DEGs (***p_adj < 0.001, *p_adj = 0.01). The percentage of cells expressing the gene are plotted against the average expression per cell type. **c** Ranking of significant signaling pathways based on differences of overall information flow (i.e. summed interaction strengths) within the inferred networks on day 0 and day 2.

Left: relative information flow. Middle: absolute information flow (log scale). Right: zoom in on the absolute information flow of pathways with log(information flow) <0.1 at any time point. Signaling pathways are colored by the type of interaction (yellow: secreted signaling, purple: ECM-receptor, green: cell-cell contact). **d–g** Hierarchical plots showing the inferred communication network for EDN (**d**), IL6 (**e**), LIFR (**f**) and EGF (**g**) signaling. Circle sizes are proportional to the number of cells in each cell type, and line width corresponds to the interaction strength. The relative contributions of ligand-receptor pairs making up at least 1% of the pathway strength are shown on the right.

We analyzed lung epithelial cells from SfSCR mice with scRNA-seq before (day 0) and 2 and 3 days after tamoxifen injection (Fig. 6c). Cells were sorted analogously to the SRC model (Supplementary Fig. 4a) to enrich viable, rare EPCAM⁺ and dividing GFP⁺ cells (Supplementary

Fig. 6c). We were able to distinguish all previously identified cell types and two additional small clusters (Fig. 6d, Supplementary Fig. 6d–f, and Supplementary Data 9). One cluster comprised *Krt13*⁺ basal cells (n = 88 cells; named basal_Krt13⁺) and the other *Krt15*⁺ club cells (n = 19

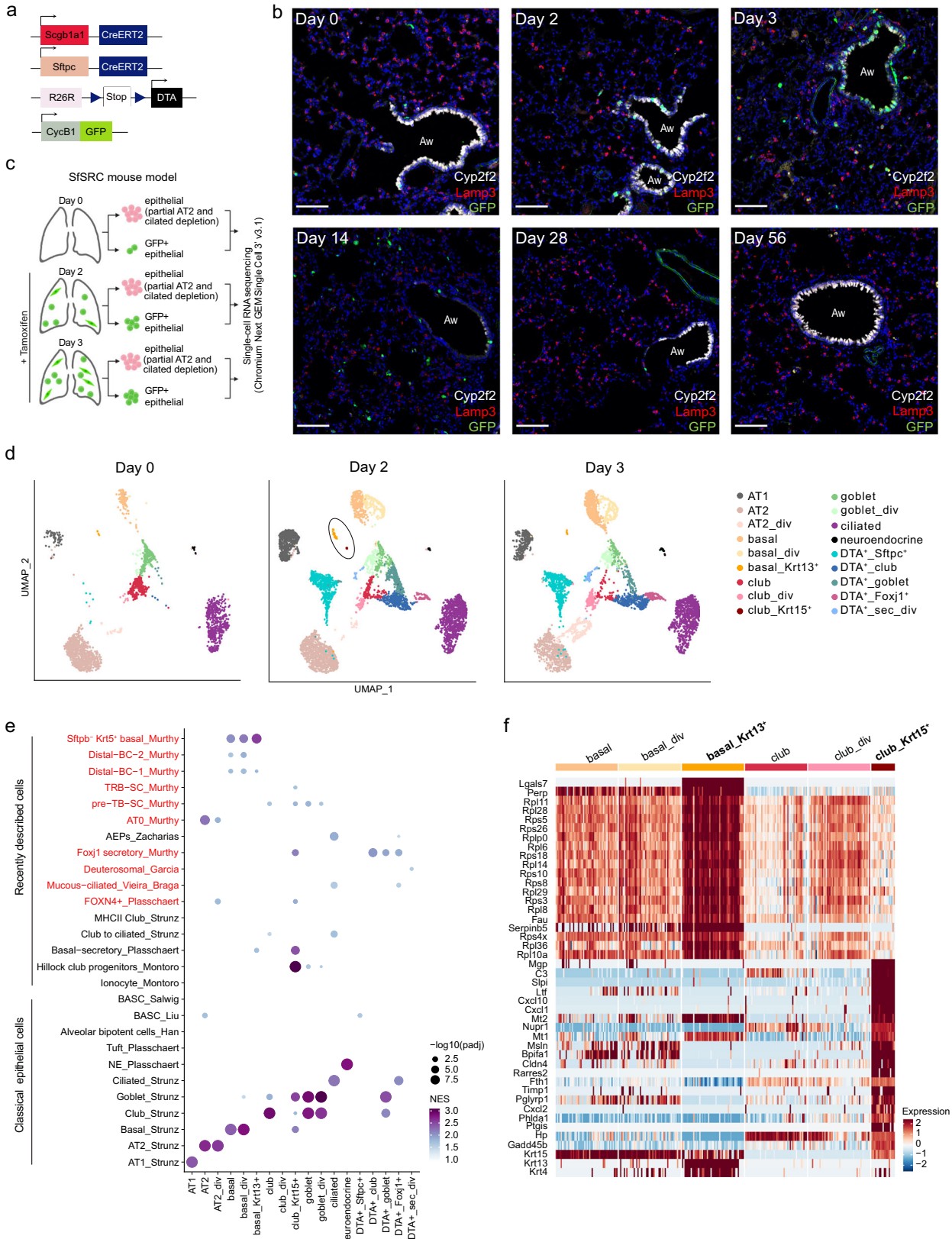

cells; named club_Krt15+) (Fig. 6d). Projection of cells from SfSRC mice onto cells from SRC mice and integration of all datasets generated in this study showed that basal_Krt13+ cells were present in both models, while club_Krt15+ cells were unique to the SfSRC model (Supplementary Fig. 6g, h). Since club_Krt15+ cells were found in the SfSRC model only at day 2 and we cannot distinguish from which of the nine mice

used at this time point these cells originated, we cannot rule out the possibility that they were derived from a single mouse.

To systematically assess the similarity of basal_Krt13+ and club_Krt15+ cell populations with reported cell types, we performed fast gene set enrichment analysis (FGSEA) with gene signatures of canonical lung epithelial cell types and recently described lung

**Fig. 6 | Characterization of lung epithelial cells in the SfSRC mouse model.**
**a** Schematic of Sftpc-CreER, Scgb1a1-CreER, Rosa26R-DTA, and CycB1-GFP transgenes in SfSRC mice. **b** IF staining of SfSRC mouse lungs with Cyp2f2 (white; club cells), Lamp3 (red; AT2 cells), CycB1-GFP (green; dividing cells) and DAPI (blue; nuclei), on days 0, 2, 3, 14, 28 and 56 after tamoxifen injection. Aw: airway. Scale bar: 100 μm. Images are representative of three independent experiments with $n = 3$ animals analyzed per timepoint. **c** Scheme of cell sorting and scRNA-seq strategy. Created with BioRender.com. **d** UMAP embedding showing cell type assignment of epithelial cells before (day 0, $n = 5$ mice), and two ($n = 9$ mice) and three days ($n = 6$ mice) after tamoxifen injection. At day 2, basal_Krt13[+] and club_Krt15[+] cells are highlighted. **e** Dot plot showing normalized enrichment score (NES) and significance for published signatures of epithelial cell types in all SfSRC mouse epithelial populations (fgsea package[91], all timepoints merged). DATP damage-associated transient progenitors, PATS pre-alveolar type-1 transitional cell state, RAS respiratory airway secretory, BC basal cell, TRB-SC terminal and respiratory bronchioles secretory cell, pre-TB-SC pre-terminal bronchiole secretory cell, AEP alveolar epithelial progenitor, BASC bronchioalveolar stem cell. Cell types described in the human lung are depicted in red. **f** Heatmap of the top 20 marker genes for basal_Krt13[+] and club_Krt15[+] cells, and expression of Krt15, Krt13, and Krt4, in basal and club cells of SfSRC mice (roc test, all three timepoints merged).

epithelial cells[7,11,13,14,31,33,55,62–64]. Although *Krt13*[+] basal cells were previously described in the mouse trachea as basal-secretory cells[11], our basal_Krt13[+] cells were transcriptionally distinct (Fig. 6e). Instead, they closely resembled *SFTPC KRT5*[+] basal cells found in human airways[7] (Fig. 6e). Club_Krt15[+] cells were transcriptionally similar to mouse hillock club cells, despite the fact that they did not consistently express the main marker genes *Krt4* and *Krt13* (Fig. 6e, f). Since hillock club cells were found exclusively in the trachea[31], and we removed trachea and main bronchi before performing scRNA-seq, our data indicate that hillock-like cells can be present in the distal airways.

Finally, we examined whether basal-goblet cells identified only in the separate analysis of GFP[+] cells (Fig. 1) were also present in the other analyses. By projecting the cells from each scRNA-seq dataset onto the cells identified in Fig. 1, we found that basal-goblet cells (dividing and non-dividing) were indeed present in the SfSRC model dataset (Supplementary Fig. 6i) but were assigned to basal or goblet cells during clustering. Additionally, basal-goblet cells from the SfSRC model are located between basal and goblet cells in the UMAP embedding (Supplementary Fig. 6j), expressed both basal and goblet marker genes, and had a similar transcriptional profile to basal-goblet_div from Fig. 1 (Supplementary Fig. 6k), supporting the idea that basal-goblet cells represent an intermediate state between these two cell types.

Overall, the combined depletion of *Scgb1a1*[+] and *Sftpc*[+] cells revealed two cell types, basal_Krt13[+] resembling human *SFTPC KRT5*[+] basal cells, and club_Krt15[+] with similarity to hillock club cells.

### Trajectory modeling predicts goblet cell differentiation into basal and club cells

Finally, we explored the relations and predicted biological trajectories between the identified epithelial cell types in response to *Scgb1a1*[+] and *Sftpc*[+] cell loss. To get a better understanding of the data topology in the SfSRC model, we visualized the cells before and after injury in a force-directed graph (Fig. 7a) and summarized the connectivities with partition-based graph abstraction (PAGA) (Fig. 7b). The edges in the PAGA graph represent possible differentiation paths, which we analyzed using diffusion maps of different cell type subsets to obtain finer resolution.

According to the basic lineage model of the lung epithelium, basal cells are considered to be progenitors of club cells, which in turn give rise to goblet and ciliated cells[34]. However, in the PAGA map, basal cells were strongly connected to goblet cells and not to club cells (Fig. 7b). A diffusion map restricted to basal, goblet, and club cells confirmed that goblet cells serve as bridge between basal and club cells (Fig. 7c). The directionality inferred by RNA velocity even suggests that goblet cells give rise to basal cells (Fig. 7d and Supplementary Fig. 7a), which is in accordance with a recently described dedifferentiation trajectory from goblet to basal cells following bleomycin lung injury[65]. Basal-goblet cells, identified by projecting SfSRC cells onto GFP[+] SRC cells (Supplementary Fig. 6i), were distributed between goblet and basal cells in the diffusion map, reinforcing that they are an intermediate cell type between goblet and basal cells (Supplementary Fig. 7b). We modeled a pseudotime trajectory from the tip of the goblet cell cluster and calculated DEGs along the goblet-basal and goblet-club axes separately.

The transition from goblet to basal cells is primed by a decrease of the goblet cell markers *Muc5b*, *Bpifb1*, *Scgb3a1*, *Scgb3a2*, and *Bpifa1*, and an increase of the basal cell markers *Aqp3*, *Krt15, Krt5*, and *Dcn* (Fig. 7e–g and Supplementary Data 10).

Next, we followed the goblet cell trajectory to club cells (Supplementary Fig. 7c–f). The key transcription factors required for goblet cell differentiation and mucus production *Spdef* and *Foxa3*[66] are among the first downregulated genes, followed by goblet cell markers including *Agr2*, *Bpifb1*, *Muc5b*, and *Scgb3a1* (Supplementary Fig. 7e, f and Supplementary Data 11). At the same time, the expression of club cell markers such as *Cckar*, *Scgb1c1*, and *Sftpb* increased. As in the goblet-basal cell trajectory, the differentiation between goblet and club cell clusters occurs without cell division (Fig. 7d and Supplementary Fig. 7c). PAGA and diffusion maps also showed a strong connection between DTA[+]_goblet and DTA[+]_club cells, suggesting that DTA[+]_goblet cells differentiate into club cells similar to uninjured goblet cells (Fig. 7b and Supplementary Fig. 7g).

Club cells are known to give rise to ciliated cells[9]. We also observed a trajectory from club to ciliated cells, which occurred almost exclusively through DTA[+]_club (Supplementary Fig. 7h, i), suggesting that differentiation into ciliated cells occurs preferentially through stressed club cells in our mouse model. While the expression of club cell markers is gradually reduced during the transition to DTA[+]_club and completely lost in DTA[+]_Foxj1[+] cells, the expression of ciliated markers already started to increase in DTA[+]_club cells (Supplementary Fig. 7j–l and Supplementary Data 12).

Finally, club_div cells were predicted to give rise to both club and AT2 cells (Fig. 7h, i). As observed in SRC mice, club_div cells had decreased secretory cell markers, such as *Scgb3a2* and *Scgb1c1*, and increased AT2 markers, such as *Napsa* and *Lamp3* (Fig. 7j–l and Supplementary Data 13). *Cldn18*, a tight junction protein highly expressed in the alveolar epithelium, was also among the top DEGs upregulated in club_div, corroborating the club to AT2 cell transition (Fig. 7l).

Taken together, these data suggest secretory cells as the main epithelial progenitor cell population in the distal lung following *Scgb1a1*[+] and *Sftpc*[+] cell loss. In particular, goblet cells, previously thought to be terminally differentiated, are predicted to be progenitors for basal and club cells, and club_div cells for club and AT2 cells. Stressed (DTA[+]) club cells seem to be driven to differentiate into ciliated cells (Fig. 7m).

## Discussion

This study provides a detailed single-cell transcriptome analysis of the lung epithelial and mesenchymal responses to targeted epithelial damage using genetically modified mouse models and cell sorting strategies to enrich rare and functionally dividing cells. We reveal and characterize cell types and provide insights into the functional roles of lung epithelial cells and their differentiation paths.

We showed that adventitial fibroblasts actively proliferate early after in vivo injury of *Scbg1a1*[+] cells, and have the ability to support epithelial organoid formation in vitro. Several studies have previously characterized mesenchymal cell populations of the adult mouse lung, often using different terminologies for the cell types. We chose to use

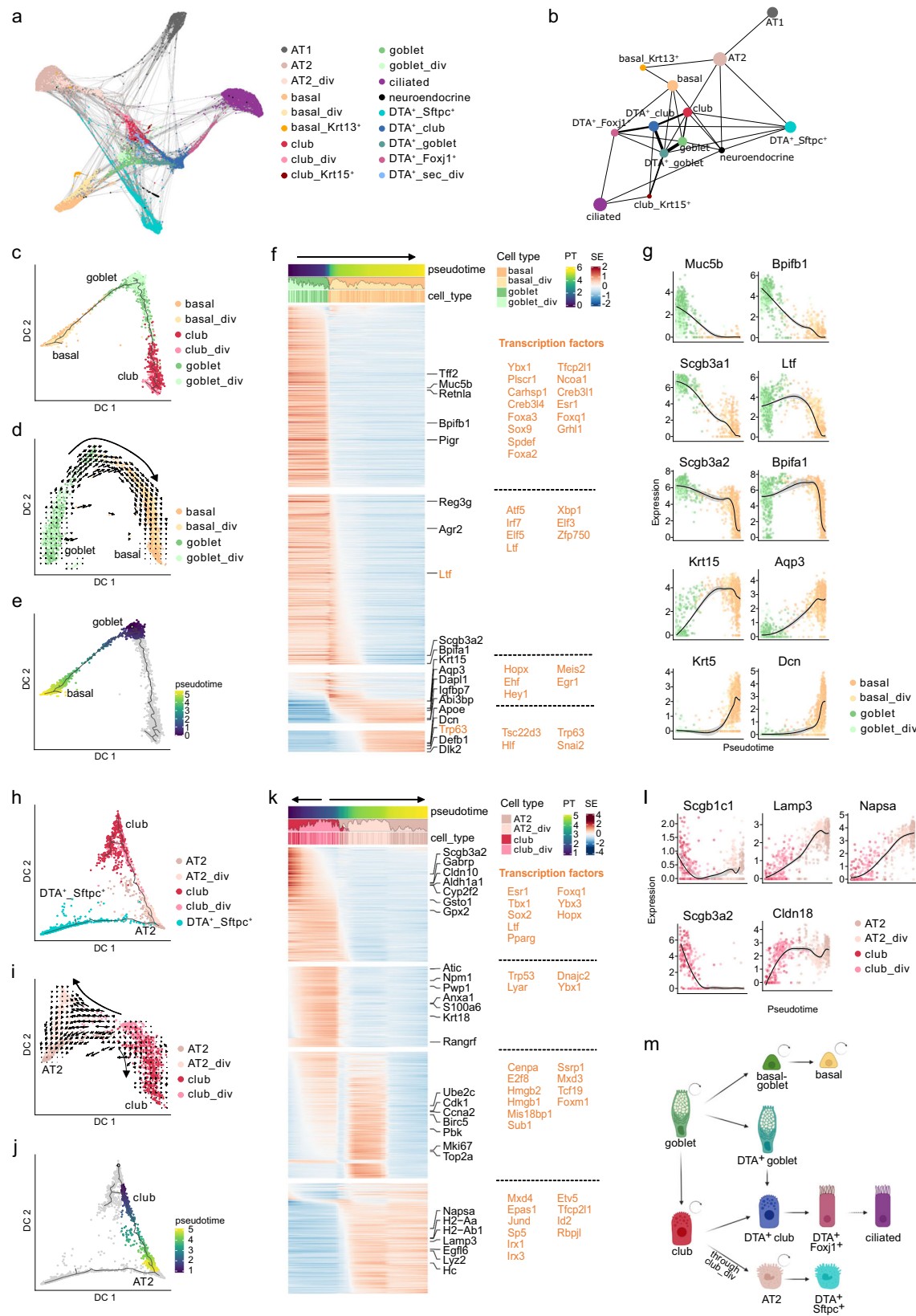

the annotation from a study that identified mesenchymal cell types based on scRNA-seq and immunofluorescence, and in which the identified populations strongly overlap with ours (Tsukui et al.)[28]. The adventitial fibroblasts of Tsukui et al. [28] (*Dcn*[+] *Ly6a*[+]) resemble the mesenchymal stromal cells (MSCs) described by Hurskainen et al. [67] (*Dcn*[+] *Ly6a*[+]), as well as the matrix fibroblasts of Xie et al. [35] (*Col14a1*[+],

*Dcn*[+]), making it likely that these are all the same cell type. Although PDGFRA[+] cells were previously shown to be able to support epithelial organoid formation in vitro[15], this has not yet been demonstrated for adventitial fibroblasts, which are a subset of PDGFRA[+] cells.

Elimination of *Scgb1a1*[+] cells or both *Scgb1a1*[+] and *Sftpc*[+] cells resulted in immediate proliferation of most epithelial cell types with

**Fig. 7 | Differentiation trajectories between epithelial lung cells in SfSRC mice.**
**a** Force-directed graph of single cells with edges to the 10 nearest neighbors.
**b** PAGA graph summarizing the connectivities between cell types. The thickness of the lines represents connectivity between cell types. **c** Diffusion map of basal, goblet and club cells with pseudotime trajectory colored by cell type. **d** Diffusion map of basal and goblet cells with RNA velocity vectors indicating differentiation of goblet into basal cells. **e** Diffusion map of basal and goblet cells with pseudotime of cells selected to characterize the goblet to basal cell trajectory in **f. f** Smoothed expression heatmap of the top 1000 altered genes along the differentiation trajectory from goblet to basal cells. The names of the top five genes in each cluster are annotated and transcription factors are indicated in orange. All transcription factors of each cluster are listed on the right. **g** Normalized expression along the goblet to basal cell pseudotime trajectory for exemplary genes. The black line is the smoothed expression using local polynomial regression fitting, with the 95%

confidence interval shown in gray. **h** Diffusion map of club, AT2, and DTA$^+$_Sftpc$^+$ cells with pseudotime trajectory colored by cell type. **i** Diffusion map of club and AT2 cells with RNA velocity vectors indicating that dividing club cells give rise to club cells and AT2 cells. **j** Diffusion map of club, AT2, and DTA$^+$_Sftpc$^+$ cells with pseudotime trajectory colored by pseudotime of the cells selected to characterize the club to AT2 cell trajectory in **k. k** Smoothed expression heatmap of the top 1000 altered genes along the differentiation trajectory from club to AT2 cells. **l** Normalized expression along the club to AT2 cell pseudotime trajectory for exemplary genes. Smoothed expression and confidence interval are shown as in **g**. Figures were calculated on merged data from all timepoints (day 0, 2, and 3). **m** Proposed model of differentiation routes (straight arrows) and self-renewal capacities (curved arrows) of lung epithelial cells based on data and mouse models in this study. Created with BioRender.com. PT pseudotime, SE scaled expression.

the exception of ciliated and AT1 cells known to be terminally differentiated. In particular, AT2 cells showed a remarkable increase in cell division even when only *Scgb1a1*$^+$ club cells were targeted, which is unexpected since AT2 cells do not contribute to bronchiolar epithelium regeneration. The reason for the pronounced AT2 cell proliferation is unclear. Since inflammatory signals were previously shown to induce AT2 proliferation and differentiation[68], it is possible that the expression of pro-inflammatory genes seen in our mouse model stimulates AT2 proliferation. The expansion of AT2 cells could also be related to an indirect role of these cells in lung regeneration. For example, a subset of AT2 cells was recently shown to play a critical role in maintaining barrier immunity in the lung by regulating the function of memory CD4$^+$ T cells[69]. In agreement with an immune modulatory role, AT2 cells displayed an upregulation of genes associated with immune response and inflammation.

Based on our idea that the use of chemical or infectious agents commonly used for mouse lung regeneration studies could destroy yet unknown cells involved in regeneration, one of our goals was to search for new cell types through the application of precise genetic cell disruption. Indeed, our study revealed rare cell populations that we named basal-goblet_div, club_div, basal_Krt13$^+$, and club_Krt15$^+$ cells, which were not previously described in the mouse distal lung. Club_div cells did not display gene expressing signatures of known club progenitor cells but were predicted to be the source of club and AT2 cells upon *Scgb1a1*$^+$ and *Sftpc*$^+$ cell depletion, so they might represent a progenitor cell population. Basal_Krt13$^+$ cells were closely related to human *SFTPC KRT5*$^+$ basal cells but not to mouse basal cells, and club_Krt15$^+$ cells displayed a transcriptional profile similar to hillock cells that are located in the trachea, suggesting their presence also in the distal mouse lung. Further analyses are required to validate the presence, precise location, and function of these identified cells.

The identification of basal-goblet_div cells expressing both basal and goblet cell marker genes as a distinct cell population was only possible by analyzing dividing cells separately, and we could subsequently confirm their presence in all datasets through data projection. These cells might represent an intermediate cell state or a cell type in the differentiation path from goblet to basal cells (see also below). In line with this, an intermediate cell state between basal and goblet cells was previously predicted by scRNA-seq differentiation trajectory, and cells co-expressing the markers BPIFB1 (goblet) and KRT5 (basal) in IF analysis were observed in the intrapulmonary airways of bleomycin-injured mice[65]. These data, together with our finding that goblet cells and basal-goblet cells are capable of self-renewal and differentiation, argue for the existence of basal-goblet cells as an intermediate cell state between goblet and basal cells during regeneration.

Goblet cells are the main mucus-producing cells in the airways and are thought to be terminally differentiated, therefore cannot divide, and originate from club cells[34]. We found that goblet cells are capable of dividing in response to epithelial injury, and trajectory modeling indicates that they differentiate into club and basal cells

instead of the other way around as previously assumed. The latter observation is supported by a single-cell transcriptomics study that predicted dedifferentiation from goblet to basal cells in a bleomycin injury mouse model[65]. Furthermore, Tata et al. showed that basal cell ablation using Krt5$^+$-DTA mice induced dedifferentiation of secretory cells into basal cells by in vivo lineage tracing of *Scgb1a1*$^+$ cells[17], a marker shared by club and goblet cells[64]. The authors also demonstrated that cells expressing *Atp6v1b1*, a marker for goblet cells in our study (Supplementary Data 6 and 9), dedifferentiated into basal cells in an organoid culture system. Together, these findings strongly suggest that goblet cells can function as progenitor cells in the distal mouse lung and actively contribute to epithelial repair.

Severe COVID-19 is characterized by persistent lung inflammation that can lead to the development of acute respiratory distress syndrome and death, but it is not clear what triggers inflammation[70]. Recently, it has been shown that SARS-CoV-2 can infect peripheral blood monocytes and tissue-resident macrophages in COVID-19 patients, and activate an inflammatory cascade with a unique transcriptome that results in inflammatory cell death[71,72]. The authors suggested that the release of inflammatory factors upon cell death may be a trigger for the overactive immune inflammatory response seen in COVID-19. In our study, the transcriptional profile of persistent DTA$^+$ cells in the mouse lungs allowed us to identify a subpopulation of lung epithelial cells from COVID-19 patients, as they share increased expression of key inflammatory factors and pathways. The presence of this population in the lungs of COVID-19 patients suggests that epithelial cells can also be a source of immune activation in these patients. Further characterization of the immune response in injured lungs of SRC and SfSRC mice and the identified cell population in SARS-CoV-2-infected patient lungs may help to better understand the overshooting immune response in COVID-19 patients and identify crucial interactions between immune and epithelial cells during lung regeneration. Of note, the DTA$^+$ cell population in the mouse lungs was defined by the presence of DTA mRNA, and validation of DTA protein expression remains to be performed. Likewise, the presence of the DTA$^+$-like cells in the tissue of COVID-19 lungs must still be verified.

In summary, our study yields perspectives on how lung cells react to pathological conditions and the cellular and molecular mechanisms involved in epithelial regeneration. We also identified cell types and/or cell states, although we would like to point out that further studies are needed to determine their exact functional and mechanistic role. In addition, we present models and a comprehensive resource that serve as a solid foundation for future research specifically focused on functional studies.

## Methods
### Mouse experiments
All experiments were approved by the regional authority in Karlsruhe, Germany, under protocol numbers 35-9185.81/G-238/14 and 35-

9185.81/G-303/19 and were performed according to federal and institutional guidelines. Mice were housed under strict specified pathogen-free (SPF) conditions at 22.0 ± 2.0 °C and 55.0 ± 10.0 % relative humidity. The light/dark cycle was adjusted to 14 h lights on and 10 h lights off with the beginning of the light and dark period set at 6 am and 8 pm, respectively. Mice of both sexes between 8 and 16 weeks old were used. Scgb1a1^tm1(cre/ERT)Blh/J (Scgb1a1-CreER; C57BL/6 N background), B6.129P2-Gt(ROSA)26Sor^tm1(DTA)Lky/J (Rosa26R-DTA; C57BL/6 background), and Tg(Pgk1-Ccnb1/EGFP)1Aklo/J (CycB1-GFP; C57BL/6 J background) mice were purchased from The Jackson Laboratory and used to generate the SRC mouse line. Sftpc^tm1(cre/ERT2,rtTA)Hap (Sftpc-CreER) mice[25], kindly provided by Rocio Sotillo, DKFZ, Heidelberg, Germany (originally provided by Harold A. Chapman, University of California, San Francisco, USA), were crossed with SRC mice to generate the SfSRC mouse line. Rosa26R-DTA x CycB1-GFP (RC) mice were used as controls. The mice were homozygous for all transgenes except CycB1-GFP, which has multiple transgenes integrated in the genome. These CycB1-GFP transgenic mice, previously developed and extensively validated by Klochendler et al. [29], constitutively express a fusion protein of the 105 N-terminal residues of the cyclin B1 protein and eGFP, whose protein expression changes similar to the cyclin B1 protein during the cell cycle. If cells are in G0 or G1 stages of the cell cycle, the cyclin B1 portion is ubiquitinated by the APC/C complex, directing the fusion protein for degradation, thereby rendering cells GFP negative. In the S/G2/M stages of the cell cycle, the activity of APC/C is low, the CycB1-GFP protein is not degraded, and the cells are GFP⁺. SRC, SfSRC, and control RC mice were injected intraperitoneally with 4 mg tamoxifen dissolved in corn oil once (Figs. 1 and 2 and Supplementary Figs. 1 and 2) or on two consecutive days with 4 mg each (all other figures). Control animals (day 0) were not injected. Analysis was done at the indicated timepoints, counting from the first day of tamoxifen injection. Mice were sacrificed by cervical dislocation, the abdominal aorta and vena cava were severed, and lungs were perfused with PBS though the right ventricle. Further processing of the lungs depended on the type of downstream analysis.

## Immunofluorescence

Lungs were carefully inflated with 1% PFA (Thermo Fischer Scientific) though the trachea, fixed overnight at 4 °C, washed in PBS, subjected to a 10-30% sucrose gradient, embedded in OCT compound (Sakura), and snap-frozen in ethanol/dry ice. Cryosections with 10 μm thickness were placed on SuperFrost UltraPlus Gold adhesion slides (Menzel) and stained using immunostaining chambers (Thermo Fischer Scientific). Briefly, sections were washed with PBS and incubated with a perm/block solution containing 0.3% Triton X-100 (Sigma) and 5% BSA (Sigma) in PBS for 30 min at room temperature (RT). Primary antibodies were incubated at RT for 60 min, sections were washed with 0.1% Tween 20 (Sigma) in PBS, and secondary antibodies were incubated for 30 min at RT followed by washing with 0.1% Tween 20 in PBS. The following antibodies were used: acetylated tubulin (1:1000, #T7451, Sigma), Agr2 (1:1000, #12275-1-AP, Proteintech), CC10 (1:500, #07-623, Millipore), CD34 (1:200, #553731, BD Biosciences), Cyp2f2-Alexa Fluor 647 (1:1000, #sc-374540 AF647, Santa Cruz), GFP-FITC (1:500, #600-102-215, Rockland), Krt5 (1:500, # ab53121, Abcam), Lamp3 (1:500, #DDX0192-100, Dendritics), Npnt (1:50, #PA547610, Invitrogen), Pdpn (1:2000, #ab11936, Abcam), SPC (1:1000, #AB3786, Millipore), donkey anti-rabbit Alexa Fluor 568 (1:500, #A10042, Invitrogen), goat anti-rat Alexa Fluor 568 (1:500, #A11077, Invitrogen), and goat anti-hamster Alexa Fluor 647 (1:500, #A21451, Invitrogen). The acetylated tubulin antibody was conjugated with Pacific Blue Antibody Labelling Kit (Thermo Fisher Scientific) prior to use. All antibodies were diluted using the perm/block solution. When indicated, sections were incubated with 0.6 μg/ml of DAPI solution (BD Biosciences) and washed with PBS before mounting with Prolong Diamond Antifade mounting media (Thermo Fischer Scientific). Images were acquired using the A1R confocal microscope (Nikon) and processed using Fiji (ImageJ 1.53c)[73].

## Histopathology

PBS-perfused lungs were inflated with 1% PFA (Thermo Fischer Scientific) and fixed for 24 h at 4 °C. Lungs were dehydrated in increasing concentration of ethanol and embedded in paraffin. Sections of 3 μm were placed on SuperFrost UltraPlus Gold adhesion slides (Menzel), deparaffinized with histoclear (Linaris), and hydrated in decreasing concentrations of ethanol before staining. Hematoxylin and eosin staining was done using the H&E staining kit (Abcam), according to the manufacturer's instructions. Masson staining was performed with kit trichrome de Masson Anilin Blue variation (RAL diagnostics), according to the manufacturer's instructions. Briefly, sections were incubated in Mayer Haemalum for 10 min, rinsed in running tap water, stained with Ponceau Fuchsin for 5 min, rinsed in 2 baths of 1% acetic water, dipped in phosphomolybdique, stained in Aniline blue for 5 min, and rinsed in 2 baths of 1% acetic water. Sections were dehydrated and dipped in histoclear before mounting with Entellan (Millipore). Images were acquired using the Ni-E microscope (Nikon) and processed using Fiji (ImageJ 1.53c)[73].

## Generation of lung single-cell suspensions

PBS-perfused lungs were inflated with 2 ml of digestion cocktail containing 50 U/ml dispase (Corning), 250 U/ml collagenase type I (Worthington), 5 U/ml elastase (Worthington), and 60 U/ml DNAse I (Roche). The trachea was clipped distally and lungs were dissected in a petri dish on ice to remove extrapulmonary airways (trachea and main bronchi). Lung lobes were placed in a C tube (Miltenyi) containing 3 ml of digestion cocktail, and the m_lung_01 program was run on gentle-MACS (Miltenyi). C tubes were placed in a rotating incubation oven at 37 °C for 30 min. The m_lung_02 program was run again, and the tubes were placed on ice for the next steps. The lung cells were passed through a 70 μm cell strainer (Corning) and centrifuged at 400 g for 5 min. The pellet was resuspended in a red blood lysis buffer solution (0.15 M NH4Cl, 10 mM KHCO3, 0.1 mM EDTA), incubated for 2 min on ice, and washed with EasySep buffer (STEMCELL Technologies) at 400 g for 5 min. To isolate mesenchymal cells (Fig. 2, Supplementary Fig. 2), a milder digestion cocktail was used containing only 375 U/ml collagenase and 60 U/ml DNase I. In all experiments, lung single-cell suspensions from several mice were pooled before fluorescence-activated cell sorting was performed.

## Fluorescence-activated cell sorting

Lung single-cell suspensions were resuspended in EasySep buffer, incubated with primary antibodies for 20 min at 4 °C, washed with EasySep buffer at 400 g for 5 min, and, in the case of unconjugated primary antibodies, secondary antibodies were added, incubated for 15 min at 4 °C and washed again. The following antibodies were used, according to the cell populations sorted: CD31 APC (1:200, #551262, BD Biosciences), CD45 APC (1:200, #559864, BD Biosciences), Ter119 APC (1:200, #17-5921-82, eBioscience), Epcam PE-Cy7 (1:300, #118216, BD Biosciences), CD24 PE (1:200, #12-0242-81, eBioscience), MHCII Alexa Fluor 700 (1:200, #56-351-82, eBioscience), Npnt (1:40, #PA547610, Invitrogen), Pdgfra BV711 (1:100, #740740, BD Biosciences), Sca-1 APC-Cy7 (1:200, #560654, BD Biosciences), CD34 PerCP-Cy5.5 (1:100, #119327, Biolegend), and donkey anti-goat Alexa Fluor 680 (1:500, #A32860, Invitrogen). DAPI at 0.6 ug/ml (BD Biosciences) was added before acquisition to exclude dead cells. Hematopoietic (CD45⁺ and Ter119⁺) and endothelial (CD31⁺) cells were excluded. Dividing cells were sorted based on their expression of GFP. For mesenchymal cell analysis, GFP⁺ dividing cells were run on Chromium as separate samples. For epithelial cell analysis, dividing GFP⁺ EPCAM⁺ cells were sorted and added to the EPCAM⁺ cells before running the samples on Chromium. To enable enrichment of infrequent cell types, AT2 cells

were partially depleted using the anti-MHC II antibody[74] or AT2 and ciliated cells were partially depleted using the anti-CD24 antibody. With partial depletion, only approximately 10% of AT2 and ciliated cells were kept, but GFP⁺ AT2 or ciliated cells were not depleted as they were isolated during the separate GFP⁺ cell sorting. Gating strategies are depicted for each experiment. Sorting was performed with a BD FACS Aria II (BD Biosciences) with a 100 μm nozzle.

## Lung organoid cultures

Lung single-cell suspensions from non-injured SRC lungs were sorted according to the sorting strategy described in Figure S2. Approximately 1,000 progenitor epithelial cells (DAPI⁻ CD45⁻ CD31⁻ Ter119⁻ EPCAM^high CD24^dim)[36] were resuspended with approximately 20,000 adventitial fibroblasts (DAPI⁻ CD45⁻ CD31⁻ Ter119⁻ EPCAM⁻ PDGFRA⁺ CD34⁺ SCA-1⁺) or the same number of alveolar fibroblasts (DAPI⁻ CD45⁻ CD31⁻ Ter119⁻ EPCAM⁻ CD34⁻ SCA-1⁻ PDGFRA⁺ NPNT⁺) in 50% Growth Factor Reduced Matrigel (Corning). Cell suspensions were carefully dispensed into 0.4 μm pore polyethylene terephthalate Falcon cell culture inserts (Corning), placed in 24-well plates with DMEM-F12 medium supplemented with 5% FBS, 1% insulin/transferrin/selenium (Thermo Fisher Scientific), and 0.05 μg/ml FGF10 (GenScript). 0.1% ROCK inhibitor (Sigma-Aldrich) was added for the first 48 h of culture. Medium was changed every 2–3 days and organoids were cultured for 3-4 weeks. Imaging of the organoids was performed with the Lionheart FX Automated Microscope (BioTek) by performing tiles and z-stack images, and the Gen5 3.05 software was used for image stitching and z-stack projections.

## Mass spectrometry

**Sample preparation.** 90,000-140,000 adventitial fibroblasts (DAPI⁻ CD45⁻ CD31⁻ Ter119⁻ EPCAM⁻ PDGFRA⁺ CD34⁺ Sca-1⁺), 200,000 alveolar fibroblasts (DAPI⁻ CD45⁻ CD31⁻ Ter119⁻ EPCAM⁻ CD34⁻ SCA-1⁻ PDGFRA⁺), and 160,000-230,000 PDGFRA⁻ cells (DAPI⁻ CD45⁻ CD31⁻ Ter119⁻ EPCAM⁻ CD34⁻ SCA-1⁻ PDGFRA⁻) were sorted from uninjured lungs in three independent experiments ($n = 6$, $n = 6$, and $n = 4$ mice, respectively), and lysed in RIPA buffer (Thermo Fisher Scientific) supplemented with protease inhibitor cocktail (Roche). Further sample preparation and data analysis was performed by the DKFZ Genomics and Proteomics Core Facility as follows: SDS-PAGE gel-based protein purification was performed before trypsin digestion of the proteins on a DigestPro MSi robotic system (INTAVIS Bioanalytical Instruments AG) according to an adapted protocol by Shevchenko et al.[75]. Peptides were separated on a cartridge trap column, packed with Acclaim PepMap300 C18, 5 μm, 300 Å wide pore (Thermo Fisher Scientific) in a three step, 180 min gradient from 3% to 40% ACN on a nanoEase MZ Peptide analytical column (300 Å, 1.7 μm, 75 μm x 200 mm, Waters) carried out on a UltiMate 3000 UHPLC system. Eluting peptides were analyzed online by a coupled Q-Exactive-HF-X mass spectrometer (Thermo Fisher Scientific) running in data depend acquisition mode, where one full scan at 120 k resolution (375–1500 m/z,3e6 AGC tagert, 54 ms maxIT) was followed by up to 35 MSMS scans at 15 k resolution (1e5 AGC tagert, 22 ms maxIT) of eluting peptides at an isolation window of 1.6 m/z and a collision energy of 27% NCE. Unassigned and singly charged peptides were excluded from fragmentation and dynamic exclusion was set to 60 sec to prevent oversampling of same peptides.

**Processing of mass spectrometry data and statistical analysis.** Data analysis was carried out with MaxQuant v1.6.14.0[76] using an organism-specific database extracted from Uniprot.org under default settings. Identified false discovery rate (FDR) cutoffs were 0.01 on peptide level and on protein level. The match between runs (MBR) option was enabled to transfer peptide identifications across RAW files based on accurate retention time and m/z. The fractions were set in a way that MBR was only performed within each condition. Label-free

quantification (LFQ) was done using a label free quantification approach based on the MaxLFQ algorithm[77]. A minimum of two quantified peptides per protein was required for protein quantification. Adapted from the Perseus recommendations[76], protein groups with a non-zero LFQ intensity in 70% of the samples of at least one of the conditions were used for statistics. LFQ values were normalized via variance stabilization normalization[78]. Based on the Perseus recommendations, missing LFQ values being completely absent in one condition were imputed with random values drawn from a downshifted (2.2 standard deviation) and narrowed (0.3 standard deviation) intensity distribution of the individual samples. For missing LFQ values with no complete absence in one condition, the R package missForest v1.4 was used for imputation[79].

## scRNA-seq analysis

**scRNA-seq library preparation and next-generation sequencing.** Sorted cells from pooled lungs were counted using the Luna-FL automated cell counter (Logos Biosystems), and 18,000–20,000 cells per channel were loaded onto a Chromium controller (10x Genomics), except for GFP⁺ cells, which were loaded completely without counting due to low cell number. scRNA-seq libraries were prepared using Chromium Next GEM Single Cell 3′ v2 (Figs. 1–3 and Supplementary Fig. 1–3) or Chromium Next GEM Single Cell 3′ v3.1 (all other figures) and following the manufacturer's protocol (10x Genomics). Libraries were analyzed and quantified using TapeStation D1000 screening tapes (Agilent) and Qubit HS DNA quantification kit (Thermo Fisher Scientific) before sequencing with a NextSeq 500 (Illumina) (Figs. 1–3 and Supplementary Figs. 1–3) or NovaSeq 6000 (Illumina) (all other figures). Detailed information for each scRNA-seq run can be found in Supplementary Data 1.

**Processing of scRNA-seq data.** Raw sequencing data was processed with 10x Genomics Cell Ranger v3.1.0[80]. Reads were aligned to a custom reference genome, which was created based on the mouse mm10 reference genome v1.2.0 provided by 10x Genomics. Firstly, sequences for transgenes Sgcb1a1 (3′ of ER + 130 bp linker + 3′ UTR of Scgb1a1), Sftpc (rtTA-M2 coding sequence), and DTA were added to the reference genome FASTA file. Subsequently, the endogenous Esr1 (genomic position Chr10:4611989-5005614) and 3′ UTR of Scgb1a1 (genomic position Chr19:9083642-9083739) were masked from the genome. Then, three lines corresponding to the newly-added transgene sequences were added to the reference genome GTF file. Lastly, Cell Ranger was used to create the reference genome package from the modified FASTA and GTF files. SoupX v1.5.0[81] was used to remove ambient RNA contamination. To avoid underestimation of the global contamination fraction, manual gene lists were used for the sequencing runs of GFP⁺ mesenchymal cells on day 2 and 3 (*Sftpc, Sftpa1, Sftpb, Sftpd, Dcn, Col1a1, Col1a2, Cldn3, Cldn18, Cldn2, Cldn4*) and epithelial cells on day 4 (*Sftpc, Sftpa1, Sftpb, Sftpd, Dcn, Scgb3a1, Foxj1*) after tamoxifen administration in the SRC model. SoupX was otherwise run with default parameters. Further processing and analysis of count tables was performed with Seurat v3.2.1[82]. Poor quality cells were filtered out based on high content of mitochondrial genes (4–7.5%, depending on the sample) and low total number of features (500 to 2500, depending on the sample), before integrating different samples. Detailed information for each scRNA-seq run can be found in Supplementary Data 1.

**Sample integration, cell cycle regression, dimensionality reduction, clustering, and doublets exclusion.** Individual samples were integrated with Seurat with IntegrateData() using 2000 anchor features and integrating all common features between samples. The cell cycle phase was calculated by adapting the Seurat function CellCycleScoring() to use the GFP⁺ sorted cells as a reference. Cell cycle scores were then regressed out during data scaling with ScaleData() to

mitigate the effects of cell cycle heterogeneity in the datasets from Figs. 3, 4, and 7 and Supplementary Fig. 3, 4, and 7. UMAP dimensionality reduction and nearest-neighbor graphs were calculated using the top 30 principle components. Cells were then clustered with FindClusters() with a resolution between 0.3 and 1.5, depending on cell number. Cell doublets were calculated with scDblFinder() (scDblFinder v1.2.0[83]) using the default parameters and providing a vector of the runs id in the samples parameter. Clusters of cells composed of doublets were excluded from the analysis. Similar clusters were merged, taking into consideration the phylogenetic tree calculated with BuildClusterTree(). Cells were identified based on the expression of known marker genes and, when present, clusters composed of endothelial (*Pecam*⁺) or hematopoietic (*Ptprc*⁺) cells were excluded from the analysis. In the case of mesenchymal cell analysis, only clusters expressing Col1a1 were kept.

**Differential gene expression and gene set enrichment analysis.** Differential gene expression between different cell types (markers) was calculated with FindMarkers(test.use = "roc", only.pos=TRUE, min.pct=0.2) and differentially gene expression between cells from non-injured and injured lungs was calculated with FindMarkers(test.use = "MAST", min.pct=0.2). Common marker genes of DTA⁺ epithelial cells were calculated separately for downregulated and upregulated genes by intersecting the DEGs previously calculated for each population with FindMarkers(test.use = "MAST", ident.1 = "DTA⁺_cells", ident.2=c("all populations excluding DTA⁺ cells", min.pct=0.2)). Only DEGs with a p_adj <0.05 were considered. Heatmaps for DEGs were generated with Seurat's function DoHeatmap(). GSEA was performed using Metascape v3.5[84] with a p-value cutoff of 10E-6. Activities of 14 pathways were inferred with PROGENy v1.10.0[85] using organism = "Mouse", scale=FALSE and otherwise default values. Scores were scaled and centered using Seurat's ScaleData(). Heatmap showing pathway activity for each cell cluster at day 2 was drawn with ComplexHeatmap v2.4.3[86].

**Transcriptome correlation between cell types.** For every cell type combination of day 2 samples, Spearman's rank correlation coefficient was calculated from the average normalized expression of all genes. The heatmap was drawn with ComplexHeatmap v2.4.3[86].

**Expression of human lung diseases signatures.** To infer the expression of human lung diseases signatures in mouse epithelial cells, SRC mouse features were converted to their human orthologs using bioDBnet[87] and a humanized Seurat object was generated using the same parameters as for the mouse object. Single-cell scores were calculated with AddModuleScore() using gene signatures for asthma, non-small cell lung carcinoma, and influenza from DisGeNET v7[50], and chronic obstructive pulmonary disease[88], COVID-19 bronchial epithelial cells[53], and pulmonary fibrosis[89] from MSigDB v7.5.1[51,52] databases. Expression of the different signatures by each SRC epithelial population at day 0 and day 2 was drawn with the Seurat's RidgePlot() function.

**Integration with COVID-19 dataset and DEGs analysis.** Humanized Seurat objects from SRC mouse epithelial cells at day 0 and day 2 were generated by replacing mouse genes with their human orthologs. Conversion from mouse to human genes was done using bioDBnet[87]. Epithelial cells from controls and COVID-19 patients[4], selected based on the author's "Epithelial cells" annotation, were used to integrate with our mouse datasets following Seurat package guidelines. Seurat objects for each patient sample and each mouse run were normalized and FindVariableFeatures() was run before calculating anchors. To identify integration anchors, FindIntegrationAnchors() was run with a k.filter of 100 (due to the low cell number in some patient samples) and SRC mouse samples were used as reference. Integration anchors

were then used to integrate the objects with IntegrateData(). The integrated Seurat object was scaled, UMAP dimensional reduction and nearest-neighbor graphs were calculated using the top 30 principle components, and clusters were calculated with FindClusters() and a resolution of 0.5. Cell type identification of the COVID-19 dataset was done according to the authors' annotation ("cell_type_fine" identity class) and clusters were annotated based on the identity of the majority of the cells in the cluster. When several clusters with the same cell type were identified, a number was added as a suffix. The cluster containing mouse DTA⁺ cells, together with DTA⁺-like cells from the COVID-19 dataset, was annotated as DTA⁺-like cells. DEGs in DTA⁺-like cells were calculated with FindMarkers(test.use = "MAST", min.pct=0.2, ident.1 = "DTA⁺-like cells") and considering only epithelial cells from COVID-19 patients (excluding cells from the SRC dataset and from human controls). Only DEGs with a p_adj <0.05 were considered.

**Similarity to previously described epithelial cell types.** To assess the similarity between our cell populations and mouse lung epithelial cells previously identified, fast gene set enrichment analysis was done as described previously[90]. Briefly, marker genes for all SfSRC mouse epithelial cell populations using all timepoints were calculated with FindMarkers(test.use = "MAST", logfc.threshold = -Inf, min.pct = -Inf), and housekeeping genes listed in Laughney et al. [90] were excluded. Genes were ranked according to average log FC, and the top 50 genes from previously described epithelial cell types were used to calculate the normalized enrichment score (NES) for each SfSRC mouse epithelial population using fgsea v1.14.0 package[91].

**Projection of cells between mouse models and experiments.** To compare the cell populations present in different experiments, SingleCellExperiment[92] objects were generated and cells from one experiment were projected onto cells from another experiment using scmapCell()[93]. Cell assignment was done with scmapCell2Cluster() and drawn with getSankey() from scmap v1.10.0 package[93].

**Expression of chemokines.** Chemokine genes considered for analysis were taken from MGI GO TERM "Chemokine activity". scRNA-seq data to assess chemokine expression in homeostasis by lung cell types, including mesenchymal, immune, and endothelial cells, was taken from the Mouse Cell Atlas (Supplementary Fig. 5c)[55].

**Cell-cell communication analysis.** Intercellular interactions were inferred in mesenchymal (Fig. 2 cells) and epithelial cells (Fig. 4 cells) of the SRC model before (day 0) and after tamoxifen administration (day 2) with CellChat v1.1.0[37] following the official workflow and using standard parameters. The analysis was based on expression of ligand-receptor pairs from the CellChat mouse database, which we manually adjusted to exclude ligand-receptor pairs not supported by literature and to include interactions playing a role in intercellular junctions and mesenchymal cells of the lung (adjusted database is provided in Supplementary Data 14). Neuroendocrine cells were excluded from the analysis due to low cell numbers. For each timepoint, preprocessing of normalized count tables was performed with identifyOverExpressedGenes(), identifyOverExpressedInteractions() and projectData(). The cell-cell communication network was inferred using computeCommunProb(), which by default requires 25% of the cells per group to express the ligand or receptor gene. Summarizing analyses were performed with computeCommunProbPathway(), aggregateNet() and netAnalysis_computeCentrality(). The summed incoming and outgoing interactions strengths were obtained with netAnalysis_signalingRole_scatter() and scaled to the maximum summed interaction strength at the respective time point. For comparison of the signaling pathways between the two timepoints, the CellChat objects were merged with liftCellChat() and mergeCellChat().

**Force-directed graph and PAGA analysis.** Connectivities between different cell types were analyzed with the Scanpy package[94]. PCA and a neighborhood graph were computed with pp.pca() and sc.pp.neighbors() using n_neighbors=10 and n_pcs=20. The force-directed graph was drawn with sc.tl.draw_graph(). For the PAGA analysis, the data was restricted to non-dividing cell clusters, on which the PCA and neighborhood graph were re-calculated using the same functions and parameters as above. Connectivities were quantified with tl.paga()[95].

**Diffusion maps.** Diffusion maps restricted to cell-type subsets of interest were calculated with Scanpy[94]. The data was restricted to the cell type subset and the top 2,000 variable genes with a minimum count of 10. To avoid obtaining cell cycle-related genes, dividing cell clusters and cells in the G2/M or S phase were removed for the calculation of the top variable genes. PCA and the neighborhood graph were calculated with pp.pca() and sc.pp.neighbors() using n_neighbors=15 and n_pcs=20. The diffusion map was calculated with tl.diffmap().

**RNA velocity.** Spliced and unspliced read counts per gene were obtained with Velocyto v0.17.17[96] and merged with the pre-processed normalized and log-transformed count data from Seurat. To reduce the influence of possible variable kinetic rates between different cell types or states[97], RNA velocities were calculated separately on a reduced data set containing only cell types along the transition pathway using scVelo v0.2.2[98]. First and second order moments were calculated using scvelo.pp.moments() with default parameters, velocities were calculated with scvelo.tl.velocity() in "dynamical" mode, and a velocity graph was constructed with scvelo.tl.velocity_graph(). The velocities were projected and visualized on diffusion map embeddings (see above) using scvelo.pl.velocity_embedding_grid(density = 0.5).

**Trajectory inference and differential expression analysis.** In the diffusion maps, trajectory analysis was performed with Monocle 3 v0.2.2[99] using default parameters, unless otherwise specified. The scale of the diffusion map DC values was adjusted by multiplying by 100. Cells in the diffusion map were clustered using cluster_cells() with parameters partition_qval=0.05 and num_iter=1. Trajectories were inferred using learn_graph(). Cells were assigned a pseudotime with the order_cells() function, for which root cells were manually chosen according to prior knowledge. The differential expression analysis was manually restricted to cells along the respective branch of interest and carried out using the function graph_test() with the parameter neighbor_graph = "principal_graph". The expression was then scaled to the cell subset and smoothed using the loess() function in R with span set to 0.5. Heatmaps of the top 1,000 DEGs (sorted by q-value and Morans's I) were drawn with Complex-Heatmap v2.4.3[86]. Mouse transcription factors in the DEGs were annotated with AnimalTFDBv3.0[100].

### Statistics and reproducibility
No statistical method was used to predetermine the sample size. No data were excluded from the analyses. Mice were randomly distributed between groups. The investigators were not blinded to allocation during experiments and outcome assessment.

### Reporting summary
Further information on research design is available in the Nature Portfolio Reporting Summary linked to this article.

## Data availability
Gene signatures for asthma, non-small cell lung carcinoma, and influenza were obtained from DisGeNET v7 [https://www.disgenet.org/][50], and gene signatures for chronic obstructive pulmonary disease,

COVID-19 bronchial epithelial cells, and pulmonary fibrosis from MSigDB v7.5.1 [https://www.gsea-msigdb.org/gsea/msigdb/][51,52] databases. Processed single-nuclei RNA-seq data from the lungs of COVID-19 and control patients was obtained through the Gene Expression Omnibus (GEO) database under the accession code GSE171524[4]. A manually curated database of ligand-receptor pairs was generated from the CellChat mouse database and is provided in Supplementary Data 14. Chemokine genes considered for analysis were taken from MGI GO TERM "chemokine activity" [https://www.informatics.jax.org/go/term/GO:0008009]. scRNA-seq reads were aligned to a custom reference genome published with the DOI 10.5281/zenodo.10478745 [https://zenodo.org/records/10478745]. Proteome ID UP000000589 from Uniprot was used for mass-spectrometry data analysis. The mass spectrometry proteomics data generated in this study have been deposited in the ProteomeXchange Consortium via the PRIDE[101] partner repository with the dataset identifier PXD039508. The scRNA-seq data generated in this study have been deposited in the GEO database under the accession code GSE223816.

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

## Acknowledgements

We thank the Heidelberg University Nikon Imaging Center, the DKFZ Transgenic Core Facility, the DKFZ Center for Preclinical Research, the DKFZ Light Microscopy Core Facility, the DKFZ Flow Cytometry Core Facility, the DKFZ Genomics and Proteomics Core Facility, the EMBL Genomics Core Facility, and the Single-cell Open Lab for valuable technical assistance and discussions. We thank Roman Kurley, Jana Kress, Gelsomina Kaufhold, Dorothee Terhardt, Alexandra Buse and Lea Jopp-Saile for technical assistance, and Rocio Sotillo for providing the Sftpc$^{tm1(cre/ERT2,rtTA)Hap}$ mice (originally provided by Harold A. Chapman, University of California, San Francisco, USA). This study was supported by the German Research Foundation (DFG) grant SFB873 to L.R.M., C.S. and H.G.; L.V. acknowledges the support of the Spanish Ministry of Science and Innovation to the EMBL partnership, the Centro de Excelencia Severo Ochoa and the CERCA Programme/Generalitat de Catalunya. T.G.P.G. acknowledges support from the Barbara and Wilfried Mohr foundation. S.T. acknowledges the support of The Darwin Trust of Edinburgh.

## Author contributions

Cl.S. and H.G. designed the study; Cl.S. supervised the work; L.V. provided expertise for and co-supervised data analysis; L.R.M. performed most experiments and computational analyses; M.M.

performed experiments and computational analyses related to mesenchymal cells; L.S. aligned reads and generated feature-barcode matrices, performed computational analyses related to cell-cell interaction, chemokine expression, and differentiation trajectories; S.T. performed preliminary computational analyses related to differentiation trajectories; Ch.S. pre-processed previously published datasets and curated the data used to evaluate cell-cell interaction; C.E. provided computational support for the scRNA-seq data analysis; T.G.P.G. performed histopathological analyses; S.F. contributed valuable resources; L.R.M., M.M., L.S. and Cl.S. wrote the manuscript with input from other authors.

## Funding

## Competing interests
The authors declare no competing interests.

## Additional information

[1]Division of Applied Functional Genomics, German Cancer Research Center (DKFZ), Heidelberg, Germany. [2]National Center for Tumor Diseases (NCT), NCT Heidelberg, a partnership between DKFZ and Heidelberg University Hospital, Heidelberg, Germany. [3]Division of Translational Medical Oncology, DKFZ, Heidelberg, Germany. [4]Faculty of Biosciences, Heidelberg University, Heidelberg, Germany. [5]Institute for Computational Biomedicine, Faculty of Medicine, Heidelberg University Hospital and Heidelberg University, Heidelberg, Germany. [6]Hopp-Children's Cancer Center (KiTZ), Heidelberg, Germany. [7]Division of Translational Pediatric Sarcoma Research, DKFZ, Heidelberg, Germany. [8]Institute of Pathology, Heidelberg University Hospital, Heidelberg, Germany. [9]German Cancer Consortium (DKTK), Heidelberg, Germany. [10]Structural and Computational Biology, European Molecular Biology Laboratory (EMBL), Heidelberg, Germany. [11]Broad Institute of Harvard and MIT, Cambridge, USA. [12]Department of Chemistry, Institute for Medical Engineering and Sciences (IMES), and Koch Institute for Integrative Cancer Research, MIT, Cambridge, USA. [13]Institute of Human Genetics, Heidelberg University, Heidelberg, Germany. [14]Centre for Genomic Regulation (CRG), The Barcelona Institute of Science and Technology, Barcelona, Spain. [15]Universitat Pompeu Fabra (UPF), Barcelona, Spain. [16]Department for Translational Medical Oncology, National Center for Tumor Diseases Dresden (NCT/UCC), a partnership between DKFZ, Faculty of Medicine and University Hospital Carl Gustav Carus, TUD Dresden University of Technology, and Helmholtz-Zentrum Dresden - Rossendorf (HZDR), Dresden, Germany. [17]Translational Medical Oncology, Faculty of Medicine and University Hospital Carl Gustav Carus, Technische Universität Dresden, Dresden, Germany. [18]Translational Functional Cancer Genomics, DKFZ, Heidelberg, Germany. [19]DKTK, partner site Dresden, Dresden, Germany. ✉e-mail: leila.martins@nct-heidelberg.de; claudia.scholl@nct-heidelberg.de

