## [Peer Review File · Nature Communications]

Single-cell division tracing and transcriptomics reveal cell types and differentiation paths in the regenerating lungReviewers' comments:

Reviewer #1 (Remarks to the Author):

General Comments:

This manuscript is emphasized the differential epithelial and mesenchymal cell contributors to the lung tissue regeneration under induced-injury condition, which is measured according to their proliferative activity. As a lung injury mouse model researchers in this project created Scgb1a1+ cell induction by DTA dependent tamoxifen treatment and observed the change in gene expression in different sub clusters with single cell sequencing analysis. The group was successfully identified DTA+ cell clusters of the known epithelial and mesenchymal cell origins which indicates resistant cells in similar with other lung diseases such as Covid19. Furthermore, they also revealed damage associated DTA+ cells can induce pro-inflammatory signals due to Scgb1a1+ cell depletion which is verified by increased cytokine signaling in similar with Covid19. Furthermore, they found the association between epithelial and mesenchymal cells in terms of ligand-receptor pairing to understand their involvement in the lung tissue repairing process and greatly illustrated the relationship in Fig5D-G. At the end they performed double depletion of two cell types to increase the damage, as AT2 and club cells and unidentified trachea associated cell types as Krt13+ basal cells and hillock-like club cells. It is promising for the future research to understand change in morphology of strong lung injury in terms of keratinization process. It is really an interesting outcome about dedifferentiation of goblet cells back to basal and furtherly club cells which deserves further investigation. Overall, the research makes a remarkable contribution in to the missing fields of lung regeneration process in terms of having a potential to target the DTA+ cells.

Major Comments:

- 1- In the introduction part writers were well established what is known and missing in the `lung epithelial regeneration` in terms of rare cell types and their possible functions in lung injuries. Yet it would be better to include literature references to support inhibition of Scgb1a1+ cells.
- 2- It is really interesting that the group was discovered similarities for rare cell types between their DTA lung injury model and Covid19 disease. In order to clarify why this study would be beneficial to understand rare populations in different diseases following questions needs to be answered: What is the relationship with treatment resistant cells, Covid19 lung tissue damage and regeneration? Covid19 is not mentioned until the GSEA results in line 273 therefore the main focus of the paper is confused at the end of the introduction section.
- 3- Data or Figure from control tissue with Scgb1a1+ cells and knockout verification must be perform to verify the mouse model.
- 4- What other genes are upregulated in 37% of club_div cells that express Sftpc and Scgb1a1? Which pathways involved? What happened to these cells when both Sftpc and Scgb1a1 positive cells were depleted? Inhibition of these genes has any additional influence on BASCs proliferation or existence?
- 5- In Fig2.E, writers mentioned that they performed organoid formation with co-culture method on Matrigel with epithelial progenitor+ adv-fibroblast and observed that adv-fibroblasts increased the organoid formation. There is a high similarity between the defined adv-fibroblasts and previously marked resident MSCs gene expression profile (Dcn+, Ly6a+) (PMID: 33692365). Therefore downstream analysis required to understand which protein or which mechanism (cell-to-cell contact etc.) involved in the organoid formation.
- 6- Why for epithelial cells Day 4 included into the analysis after tamoxifen administration and in the others 2-3 days? Additional day might effect on the gene expression of mesenchymal cells in the previous sections too (If they are also active in the late division like AT2)
-As "AT2 cell division is visible after 3-4 days due to GFP expressed cells", would it be

possible in Day5 ciliated cells gives GFP signal? Or in very early stages of tamoxifen administration (e.g. 12-24h) would it be possible that they were already divided and loss GFP signal?

-Especially for the ciliated cells, as there are no GFP signal in any of the days but upregulation of several genes as in Fig3F, what are the genes responsible for blocking the proliferation process? Is it because of loss of progenitor club cells? Yet it is defined in the end that, stressed DTA+ club cells can differentiate into ciliated cells in pseudo time analysis.

-Moreover, Upregulation of Ly6e gene in day 3 also an important inflammatory marker.

Minor Comments:

1- Introduction Line 68: Scgb1a1+ club cells are responsible for the regeneration of alveoli. Sentence needs to be clarified

2- In line 141, "self-renewal" term is a strong statement based on Scgb1a1 upregulation in goblet cells. It must be verified with downstream in vitro analysis.

3- In Line 200, Writers mentioned for the first time "double tamoxifen administration", it must be clarified in the method part why two different dosage was preferred.

4- It would be better to include dividing and not dividing cell populations figure in Fig4B as in the previous sections

5- To understand the contribution of AT2 cells proliferation in epithelium regeneration, it would be better to create a similar heatmap (Fig3F) showing the differentially expressed genes in different days (e.g. immune regulation as it is suggested)

6- It would be better to refer Figure 1B as "canonical club cells".

Reviewer #2 (Remarks to the Author):

In this study, Martins et al. performed single-cell transcriptomic profiling coupled to cell division tracing after diphtheria toxin (DTA)-mediated depletion of specific (Scgb1a1+ and/or Sftpc+) epithelial cell types to study the regeneration of the distal mouse lung. They describe putative new epithelial states which may have progenitor properties, but experimental proof is lacking. In organoid assays They also characterize changes in the mesenchyme after specific epithelial cell depletions, and report on a possible pro-regenerative role of adventitial and not Pdgfra+ alveolar fibroblasts based on organoid assays with of sorted cells from uninjured mice. The organoid findings for alveolar (Lipo-)fibroblasts with no colony forming capacity are very surprising given the fact that previous study showed that Lipofibroblast can support (alveolar) organoid formation (Barkauskas et al. J Clin Invest. 2013 Jul;123(7):3025-36 and reviewed in Wu & Tan, Development (2021) 148 (2): dev193458).

They further found non-depleted DTA-expressing cells in their data that showed markers of cell stress with activation of inflammatory signaling pathways and chemokine expression, which presumably could also be found in COVID19 patients from one study. It is unclear whether these cells are indeed viable and persisting -as claimed by the authors, or whether they represented dying cells. Lastly, the authors computationally infer epithelial cell differentiation trajectories describing a goblet-to-basal dedifferentiation that have been previously already been described in bleomycin-induced injury (Lange et al. Nat. Methods 2022), indicating a novel progenitor role for goblet cells in the distal lung.

Although the study provides very interesting single-cell datasets that are of relevance for the

community, the study does not go beyond a descriptive level and greatly lacks experimental proof and validation. Given the plethora of different cell sorting and scRNAseq profiling strategies for the SRC mouse model with often only incremental added value, parts of the study are very repetitive and lack clarity. In general, most findings are only shallowly described and most importantly not experimentally followed up and validated, why they largely remain speculative. A major concern is also that no information about cell numbers for the newly described cell states is provided (based on the UMAP numbers seem very low) and it is not clear from how many different animals the cells come from or in other words, whether the identified rare states are derived from more than 1 biological replicate.

Major points:

- In the method section no information regarding QC filtering of single cell data is provided, including %Mito, Min/Max of genes/counts per cell, and based on which criteria doublets were removed. This is also nowhere reported in the manuscript. This information is extremely critical especially for the goblet-basal_div cell type and DTA+ cell findings
- Regarding the goblet-basal_div cell type please provide evidence that is not a doublet and thus an artefact (stainings, showing doublet scores). This state is only described in Fig1; in all subsequent figures it is not detected anymore? Why is that? Given the putative goblet-to-basal trajectory could that be an intermediate state?
- Apart from cell cycle scoring, have you assessed proliferation markers in your single cell data as well as on tissue level?
- From Fig3 on you perform AT2 and or ciliated cell “depletions”, yet still large amounts of both cell types are still present. Please comment and rephrase accordingly
- The high fraction of dividing AT2 cells with emergence of a novel cluster at Day 4 p.i. in the SRC model is very interesting. How does this cluster differ from the other AT2 cluster? You state in the manuscript that the high number of dividing AT2 cells (starting d2) was also surprising to you. What could be the mechanism? Could cell depletion trigger inflammatory events?
- What is the phenotype of the SRC mouse? For the SfSRC mouse should provide histopathological data, but not for the SRC. Is there inflammation happening?
- Regarding the DTA+ cells, it is not clear whether that indeed are viable or simply dying cells. Please provide info about their quality (%Mito, No of counts/gene per cell etc). Please provide stainings together with apoptosis markers.
- To compare similarity in cell type signature between datasets/species, it is highly recommend to perform matchScore analysis and/or marker gene signature scorings. That does not require dataset integration.
- The identification of DTA+-like cells in COVID-19 in Fig 4c is not all clear. It is 1) not clear how you did the integration and which cell annotation was used and 2) what the marker signature of the DTA-like cells in the human disease setting is. Here indeed it is critical to not integrate but perform matchScore and/or marker gene signature scorings (see previous point). Please also confirm this in other COVID-19 datasets and provide evidence on the tissue level.
- For the cell trajectory inference, it would be critical compare the results to CellRank (which does not need direction information).
- As stated in the general comments, the organoid assays with adventitial and alveolar fibroblasts are not convincing and contradictory. Please provide as a control and reference also data with the “whole” mesenchyme. Please report on colony size and forming capacity. How did you distinguish alveolar from bronchial/alveolar organoids? I have not seen any IF stainings?

Reviewer #3 (Remarks to the Author):

Summary

Transgenic mice were engineered to express DTA in Scgb1a1-CreER – CycB1-GFP mice to express DTA in response to tamoxifen, thereby activating apoptosis and causing lung injury. Cell division was monitored by the expression of GFP during injury and repair and followed for three days. GFP+ dividing cells were sorted on days 2 and 3, isolating diverse proliferating epithelial cells GFP+ mesenchymal cells, club, goblet, basal, and AT2 cells were identified. Surviving DTA+ epithelial cells were identified. Goblet cells were proposed as epithelial progenitors. In vitro cultures were used to demonstrate the selective growth of organoids by co-culture with adventitial fibroblasts. Extensive bioinformatic analyses identified temporal and cell-type gene expression patterns, including inflammatory mediators. Comparative analyses with human lung data sets were used to conclude that DTA+ cells identified in mice were similar to those in the lungs of Covid-19 patients. A diversity of cell types were described by sub-clustering of the epithelial cells; trajectory analyses were used to predict progenitors. [SPC-CreERT2], Scgb1a1-CreERT2 mice were used to delete both airway and alveolar cells, and repair processes followed for 2-56 days, from which distinct Krt13 and Krt15 cells were identified.

The Authors conclude that the use of selective DTA lung epithelial cell injury identifies new cell types, that DTA+ expressing cells survive and resemble the populations of epithelial cells from the lungs of patients succumbing from Covid-19.

Overview

This is an extensive data set with complex bioinformatic analyses using well-established algorithms. The authors' major conclusions focused to the identification of a number of distinct epithelial cell types and progenitors which were identified during DTA-induced lung injury. While it is not surprising that a diversity of conducting airway cells, e.g., club, goblet, basal, and AT2 cells, are proliferative, the authors have not demonstrated that they proliferate and undergo asymmetric cell division and therefore contribute to the expansion of cell lineages. This conclusion needs further validation. Present conclusions are drawn primarily from extensive bioinformatic analyses and single-cell RNA profiles but are generally without experimental validation. The present experiments are complicated by many variables, including the timing of driver gene expression and allele recombination and the possible variability of recombination. Since the TAM may be active throughout the repair process during a period in which differentiation of epithelial cells is changing dynamically with injury, differentiation, and inflammation, it is unclear that specific cell states are being targeted during the course of the experiments. Rather, it appears that multiple cell types, as defined at homeostasis, may be continuously labeled throughout the experimental course. The conditional DTA experiments depend upon knowledge of the timing of DTA expression, thus specific cell types expressing DTA and perhaps their susceptibility to apoptosis. Regarding the proposal of goblet cells as progenitors, secretory cells differentiate reversibly during inflammation and metaplasia; thus, it may not be surprising that goblet cells or other secretory cells are in the process of dynamic differentiation during injury, they share club, basal, and goblet cell characteristics and therefore serve as proliferative cells and may undergo recombination during the experiments since Scgb1a1 may be expressed during

differentiation and activate Cre.

The organoid experiments with adventitial fibroblasts likely include Pdgfra+ mesenchymal cells, findings consistent with previously published work. Present experiments are not accompanied by quantitative data; a more careful characterization of the organoid and the evaluation of the purity of the fibroblast after isolation is needed. The authors have also identified the survival of a number of cells expressing DTA; is the DTA protein expressed? What are the proposed mechanisms underlying their resistance, and what biological insights are to be derived from the resistance? While gene expression studies suggest similarities between DTA+ epithelial subsets and cells identified in lungs from Covid-19 patients, mechanistic insights regarding their role in Covid-19 pathogenesis are not provided.

Editorial note: Panel b of Author Response Fig. 4 and that of Author Response Fig. 11 in this Peer Review File has been amended to remove third-party material where no permission to publish could be obtained.

Reviewer #1 (Remarks to the Author):

General Comments:

This manuscript is emphasized the differential epithelial and mesenchymal cell contributors to the lung tissue regeneration under induced-injury condition, which is measured according to their proliferative activity. As a lung injury mouse model researchers in this project created Scgb1a1+ cell induction by DTA dependent tamoxifen treatment and observed the change in gene expression in different sub clusters with single cell sequencing analysis. The group was successfully identified DTA+ cell clusters of the known epithelial and mesenchymal cell origins which indicates resistant cells in similar with other lung diseases such as Covid19. Furthermore, they also revealed damage associated DTA+ cells can induce pro-inflammatory signals due to Scgb1a1+ cell depletion which is verified by increased cytokine signaling in similar with Covid19. Furthermore, they found the association between epithelial and mesenchymal cells in terms of ligand-receptor pairing to understand their involvement in the lung tissue repairing process and greatly illustrated the relationship in Fig5D-G. At the end they performed double depletion of two cell types to increase the damage, as AT2 and club cells and unidentified trachea associated cell types as Krt13+ basal cells and hillock-like club cells. It is promising for the future research to understand change in morphology of strong lung injury in terms of keratinization process. It is really an interesting outcome about dedifferentiation of goblet cells back to basal and furtherly club cells which deserves further investigation. Overall, the research makes a remarkable contribution in to the missing fields of lung regeneration process in terms of having a potential to target the DTA+ cells.

Author response: We thank the reviewer for the careful review of our manuscript and for the positive evaluation as a "remarkable contribution to the missing fields of lung regeneration." We addressed the reviewer's concerns by conducting further experiments and adjusting the manuscript text to improve comprehension. Changes in the manuscript are highlighted in red and the rebuttal letter features references to the main revised passages by line number.

Major Comments:

1- In the introduction part writers were well established what is known and missing in the `lung epithelial regeneration` in terms of rare cell types and their possible functions in lung injuries. Yet it would be better to include literature references to support inhibition of Scgb1a1+ cells.

Author response: We appreciate the reviewer's suggestion to include references on studies that show the importance of Scgb1a1+ cells in the introduction. We changed the sentence in line 71 of the introduction to clarify that club cells are thought to be lung progenitors due to lineage tracing of Scgb1a1+ cells. Furthermore, we included additional references to previous studies, in which club (Scgb1a1+) and AT2 cells were targeted by naphthalene and bleomycin, respectively (line 75-77). Targeted removal of Scgb1a1+ cells using genetic approaches, as performed in our study, has not been reported previously.

2- It is really interesting that the group was discovered similarities for rare cell types between their DTA lung injury model and Covid19 disease. In order to clarify why this study would be beneficial to understand rare populations in different diseases following questions needs to be answered: What is the relationship with treatment resistant cells, Covid19 lung tissue damage and regeneration? Covid19 is not mentioned until the GSEA results in line 273 therefore the main focus of the paper is confused at the end of the introduction section

Author response: We agree with the reviewer that the similar transcriptional profile of a small cell population from COVID-19 patients and DTA⁺ cells is quite interesting. We addressed the reviewer's questions with a more comprehensive discussion in the manuscript (lines 529-534 of the revised manuscript). Specifically, we added the following text:

“One hypothesis for the similarity between COVID-19 cells and DTA⁺ cells is that the intracellular expression of DTA triggers a non-self response mechanism, as seen in virus infections. In accordance, some DTA⁺ cells also show high expression of an influenza gene signature (**Fig. S4I**). Nonetheless, DTA⁺ cells are transcriptionally much closer to epithelial COVID-19 cells due to overexpression of specific cytokines and chemokines.”

Why lung epithelial cells from COVID-19 patients express more inflammatory factors than an epithelial cell from an influenza patient, and the relationship between tissue injury in COVID-19 patients and lung regeneration are indeed interesting questions. However, answering them would require intensive investigations on primary human lung material, which was not our focus and is far beyond the scope of this study.

The relationship between human COVID-19 cells and mouse DTA⁺ cells was mentioned later in the results section (now line 303), when we analyzed whether cells expressing DTA could be used as a model of damaged cells to reveal cell populations and mechanisms in human lung diseases. Although this was not the main focus of our study but was an interesting result, we mentioned this at the end of the introduction and do not find it confusing. Does the reviewer suggest removing this result in the introduction?

3- Data or Figure from control tissue with Scgb1a1⁺ cells and knockout verification must be perform to verify the mouse model

Author response: We are not clear what the reviewer means by “knockout verification” since we have not knocked out the Scgb1a1 gene but used it to express DTA driven by the endogenous Scgb1a1 promoter. We showed verification of the progressive loss of Scgb1a1⁺ club cells after DTA induction by immunofluorescence in **Fig. S3D**. In addition, we now performed a new control experiment in which Rosa26R-DTA x CycB1-GFP mice (lacking the Scgb1a1-CreER genotype) were injected with tamoxifen. In this experiment, we did not detect DTA⁺ cells by scRNA-sequencing or epithelial lung injury by immunofluorescence. These new data are mentioned in the results section in lines 267-269 and are shown in **Fig. S4D and E**.

4- What other genes are upregulated in 37% of club_div cells that express Sftpc and Scgb1a1? Which pathways involved?

Author response: We thank the reviewer for this interesting question. After analyzing the differentially expressed genes (DEGs) of Scgb1a1⁺ Sftpc⁺ club_div cells in more detail, we found that they are transcriptional similar to the other club dividing cells and do not differentially upregulate any gene compared to these cells (p_{adj} < 0.05, min.pct=0.2, test.use= “MAST”). We have included this information in lines 148-149 of the revised manuscript.

What happened to these cells when both Sftpc and Scgb1a1 positive cells were depleted?

Author response: BASCs express both Scgb1a1 and Sftpc, so it is likely that DTA is activated in these cells and most of them are killed. Nonetheless, because the recombination efficiency to remove the STOP cassette for expression of DTA is not 100%, some BASCs are expected to proliferate because they have been described as progenitor cells or stem cells. However, we did not identify a distinct BASC population (as defined by clustering) in the SRC or SfSRC mouse models. Nonetheless, if we consider all Scgb1a1⁺ Sftpc⁺ cells in the SfSRC mice, which

are mainly present in the AT2 and club cell clusters (**Author Response Fig. 1a**), they represent 2.7%, 0.5%, and 0.1% of the total amount of cells sequenced on day 0, day 2, and day 3, respectively. These numbers suggest that *Scgb1a1*⁺ *Sftpc*⁺ cells are being targeted in the SfSRC model, although one should be careful with their interpretation, since we partially removed some cell types during sorting. Just as in the SRC model, these *Scgb1a1*⁺ *Sftpc*⁺ cells do not differentially express previously identified BASC markers ^{1, 2} when compared to non-*Scgb1a1*⁺ *Sftpc*⁺ cells of the same cell type (**Author Response Fig. 1b**).

Inhibition of these genes has any additional influence on BASCs proliferation or existence?

Author response: We did not inhibit these genes in our mouse models but used their expression to activate DTA for killing the respective cells. Therefore, we cannot determine the effect of *Scgb1a1* and *Sftpc* inhibition in cells.

Author Response Fig. 1: Analysis of *Scgb1a1*⁺ *Sftpc*⁺ cells in the SfSRC model. (a) Expression of the *Scgb1a1* and *Sftpc* transgenes in all cell types and timepoints of the SfSRC model shown in Fig. 6 of the manuscript. **(b)** Expression of BASC marker genes in *Scgb1a1*⁺ *Sftpc*⁺ cells (top panel) and non-*Scgb1a1*⁺ *Sftpc*⁺ (other) cells (bottom panel).

5- In Fig2.E, writers mentioned that they performed organoid formation with co-culture method on Matrigel with epithelial progenitor+ adv-fibroblast and observed that adv-fibroblasts increased the organoid formation. There is a high similarity between the defined adv-fibroblasts and previously marked resident MSCs gene expression profile (*Dcn*⁺, *Ly6a*⁺) (PMID: 33692365).

Author response: The reviewer raises an excellent point. Several studies have previously characterized the mesenchymal populations of the adult mouse lung. We chose to use the annotation from a study that identified mesenchymal cell types based on scRNA-seq and immunofluorescence, and in which the identified populations strongly overlap with ours (Tsukui *et al.*, 2020) ³ (now mentioned in line 180 of the results). Indeed, the adventitial fibroblasts of Tsukui *et al.* ³ (*Dcn*⁺ *Ly6a*⁺) resemble the MSCs of the mentioned study by Hurskainen *et al.* ⁴ (*Dcn*⁺ *Ly6a*⁺), as well as the matrix fibroblasts of Xie *et al.* ⁵ (*Col14a1*⁺, *Dcn*⁺), making it likely that these are all the same cell type. However, the ability of adventitial cells, MSCs, or matrix fibroblasts to support epithelial cells, as determined in our study, and

our observation that they proliferate after epithelial injury, has not been demonstrated by any of the above studies.

We now included these considerations into the discussion (lines 461-472), pointing out the overlap between previously described populations, with the goal of harmonizing annotations, minimize misunderstandings, and acknowledge the authors' contribution to the field.

Therefore downstream analysis required to understand which protein or which mechanism (cell-to-cell contact etc.) involved in the organoid formation.

Author response: We agree with the reviewer that it is of interest to better understand the signals involved in epithelial support by adventitial fibroblasts. To address this, we identified genes encoding proteins implicated in these processes through differentially expressed gene analysis in adventitial fibroblasts compared with all other mesenchymal cells, using CellChat and scRNA-seq (**Fig. 2F**). Furthermore, we validated these results at the protein level in FACS-sorted adventitial fibroblasts, alveolar fibroblasts, and Pdgfr⁺ mesenchymal cells by mass spectrometry and confirmed expression of the identified ligands CD34 and DCN (**Fig. S2D**), which are differentially expressed in adventitial fibroblasts. We identified further signaling proteins, for example *Bmp4* and *Tgfb3*, that might play a role in the support of epithelial cells by adventitial fibroblasts (see chapter in lines 195 – 208).

To finally prove the functional contribution of the identified proteins in cell-cell contacts, extensive experimentation would be required, such as isolation of mesenchymal and epithelial cells, removal/inhibition of the proteins of interest, e.g. by CRISPR, and organoid formation or *in vivo* studies, which would itself represent an entirely new project. We hope that the reviewer will agree with us that such experiments are not within the scope of this manuscript and that our downstream analyses shown are sufficient.

6- Why for epithelial cells Day 4 included into the analysis after tamoxifen administration and in the others 2-3 days? Additional day might effect on the gene expression of mesenchymal cells in the previous sections too (If they are also active in the late division like AT2).

Author response: The focus of our study was to understand the early events after targeted epithelial injury, i.e., to identify and analyze cell types, particularly epithelial cells that become activated, i.e., divide. Since almost no dividing cells were observed on day 1 after tamoxifen administration (**Author Response Fig. 2**), we focused our analysis on day 2 and 3. For analysis of DEGs in epithelial cells over time (**Fig. 3**), we added a one-time scRNA-seq analysis at day 4 along with immunofluorescence imaging (**Fig. S3D**) to get a sense of whether there were significant changes in cell proliferation and DEGs after an additional day. Although AT2 cells expanded more at day 4, they were already identified among the dividing cells from day 2. Also, the type of dividing mesenchymal cells did not change between day 2 and day 3. Therefore, we omitted day 4 in the other experiments, also in the sense of saving animals and costs.

Author Response Fig. 2: Immunofluorescent analysis of SRC mouse lung with Cyp2f2 (white, club cells), SPC (red, AT2 cells), GFP (green, dividing cells), and DAPI (blue, nuclei) on day 1 after tamoxifen administration (n=2 mice). Two exemplary images with different magnifications of one mouse are shown.

-As "AT2 cell division is visible after 3-4 days due to GFP expressed cells", would it be possible in Day5 ciliated cells gives GFP signal? Or in very early stages of tamoxifen administration (e.g. 12-24h) would it be possible that they were already divided and loss GFP signal?

Author response: Very early after tamoxifen administration (12-24 hours), we can barely detect any GFP⁺ dividing cells, similar to uninjured lungs (**Author Response Fig. 2**). It is well established that ciliated cells are terminally differentiated cells that have lost their ability to divide. Accordingly, we could not detect dividing ciliated cells at any time point, and we would not expect them to divide at day 5 or later.

-Especially for the ciliated cells, as there are no GFP signal in any of the days but upregulation of several genes as in Fig3F, what are the genes responsible for blocking the proliferation process? Is it because of loss of progenitor club cells? Yet it is defined in the end that, stressed DTA+ club cells can differentiate into ciliated cells in pseudo time analysis.

Author response: Thank you for sharing these considerations, which at first sight do indeed seem contradictory and require further explanation. It is well studied that club cells are the precursors of ciliated cells⁶, meaning that club cells differentiate into ciliated cells, which then lose their ability to divide due to terminal differentiation. Accordingly, the observed lack of proliferation of ciliated cells is due to their terminal differentiation state, but not to gene expression changes in response to the loss of *Scgb1a1*⁺ club cells.

We suspect that cellular stress through expression of DTA in club cells promotes differentiation into ciliated cells without cell division, as shown by our trajectory analysis (**Fig. S7H**). The finding that ciliated cells upregulated genes included in the "response to virus" cluster of the GSEA shows that they can definitely respond to epithelial injury (**Fig. 3E and F**).

-Moreover, Upregulation of *Ly6e* gene in day 3 also an important inflammatory marker.

Author response: We agree with the reviewer that *Ly6e* is an important interferon-inducible protein that was shown to restrict or enhance the entry of a variety of viruses⁷ and is contained in the GSEA "response to virus" cluster of upregulated genes in ciliated cells (**Fig. 3F**).

Minor Comments:

1- *Introduction Line 68: Scgb1a1⁺ club cells are responsible for the regeneration of alveoli. Sentence needs to be clarified*

Author response: We apologize for the incomplete explanation. This statement refers to the referenced study by Zheng et al. (2012)⁸, in which the authors performed lineage tracing of Scgb1a1⁺ cells and observed that they gave rise to AT2 and AT1 cells following damage with bleomycin or influenza. The authors concluded that the "... findings strongly suggest that Scgb1a1-expressing cells, most likely Clara cells, are a major cell type that gives rise to alveolar type I and II cells during the regeneration of alveolar epithelia in response to severe pulmonary damage in mice". We have now added a half-sentence that the authors performed in vivo Scgb1a1⁺ cell lineage tracing (line 71).

2- *In line 141, "self-renewal" term is a strong statement based on Scgb1a1 upregulation in goblet cells. It must be verified with downstream in vitro analysis.*

Author response: The self-renew statement in goblet cells is not based on Scgb1a1 upregulation but on their ability to divide, which is proven by the detection of the CycB1-GFP transgene in AGR2⁺ cells by immunofluorescence (**Fig. S1B**), their isolation by FACS using CycB1-GFP expression (**Fig. 1**), and by their upregulation of cell cycle related genes (**Fig. S3E and Supplementary Table 5**).

3- *In Line 200, Writers mentioned for the first time "double tamoxifen administration", it must be clarified in the method part why two different dosage was preferred.*

Authors' response: We apologize for the lack of clarity in this matter. In the methods section (lines 562-563), we describe in which experiments single or double tamoxifen injection was performed. Although we increased the frequency of tamoxifen administration from Fig. 3 onward from one to two injections (hence the statement in line 200), aiming to increase epithelial damage, we found that the depletion of Scgb1a1⁺ cells was comparable to a single injection. Accordingly, the immunofluorescence stainings in **Fig. 1B** (single injection) and **Fig. S3D** (double injection) show a similar response to single and double tamoxifen injections.

We now removed "double" from line 200 (now line 218) because it is not important for understanding the results, while keeping the number of injections in the methods sections and figure legends. In the legend of **Fig. 1B**, we now wrote "single tamoxifen injection" instead of "tamoxifen injection".

4- *It would be better to include dividing and not dividing cell populations figure in Fig4B as in the previous sections.*

Author response: We agree with the reviewer that it would be nice to show the UMAPs with dividing and non-dividing cells in **Fig. 4B**. However, since each population is already annotated as dividing and non-dividing (for example, club and club_div) and the entire figure is already quite crowded, we decided to show these data in the supplement (**Fig. S4B**). But if the reviewer finds that it would be essential for understanding, we can include the UMAP into **Fig. 4B** and instead move another panel to the supplement.

5- *To understand the contribution of AT2 cells proliferation in epithelium regeneration, it would be better to create a similar heatmap (Fig3F) showing the differentially expressed genes in different days (e.g. immune regulation as it is suggested).*

Author response: We thank the reviewer for this excellent suggestion, which prompted us to generate a heatmap with all DEGs in AT2 cells at the different timepoints following tamoxifen administration. This heatmap (included as **Fig. S3F**) shows an increase in AT2 cells with upregulated cell cycle-related genes (in red), in particular on day 4, which is in accordance with the cell cycle analysis in **Fig. 3B**. Furthermore, we found that several genes associated with immune response and inflammation, such as *Il33*, *Chia1*, *Cd14*, *Lgasl3*, *Ly6e*, and *Ly6c1* (in bold), were increasingly upregulated in the majority of AT2 cells already from day 2 onward. These new data were included in the manuscript text in lines 229-233, and the title of the chapter (line 212) was adapted accordingly.

6- *It would be better to refer Figure 1B as "canonical club cells".*

Author response: **Fig. 1B** shows the depletion of club cells and the increase in dividing cells after tamoxifen administration to validate our mouse model with immunofluorescence. We are not sure about the reviewer's suggestion of where to write "canonical club cells" and would like to ask the reviewer to clarify.

Reviewer #2 (Remarks to the Author):

In this study, Martins et al. performed single-cell transcriptomic profiling coupled to cell division tracing after diphtheria toxin (DTA)-mediated depletion of specific (Scgb1a1+ and/or Sftpc+) epithelial cell types to study the regeneration of the distal mouse lung. They describe putative new epithelial states which may have progenitor properties, but experimental proof is lacking. In organoid assays They also characterize changes in the mesenchyme after specific epithelial cell depletions, and report on a possible pro-regenerative role of adventitial and not Pdgfra+ alveolar fibroblasts based on organoid assays with sorted cells from uninjured mice. The organoid findings for alveolar (Lipo-)fibroblasts with no colony forming capacity are very surprising given the fact that previous study showed that Lipofibroblast can support (alveolar) organoid formation (Barkauskas et al. J Clin Invest. 2013 Jul;123(7):3025-36 and reviewed in Wu & Tan, Development (2021) 148 (2): dev193458). They further found non-depleted DTA-expressing cells in their data that showed markers of cell stress with activation of inflammatory signaling pathways and chemokine expression, which presumably could also be found in COVID19 patients from one study. It is unclear whether these cells are indeed viable and persisting -as claimed by the authors, or whether they represented dying cells. Lastly, the authors computationally infer epithelial cell differentiation trajectories describing a goblet-to-basal dedifferentiation that have been previously already been described in bleomycin-induced injury (Lange et al. Nat. Methods 2022), indicating a novel progenitor role for goblet cells in the distal lung.

Although the study provides very interesting single-cell datasets that are of relevance for the community, the study does not go beyond a descriptive level and greatly lacks experimental proof and validation. Given the plethora of different cell sorting and scRNAseq profiling strategies for the SRC mouse model with often only incremental added value, parts of the study are very repetitive and lack clarity. In general, most findings are only shallow described and most importantly not experimentally followed up and validated, why they largely remain speculative. A major concern is also that no information about cell numbers for the newly described cell states is provided (based on the UMAP numbers seem very low) and it is not clear from how many different animals the cells come from or in other words, whether the identified rare states are derived from more than 1 biological replicate.

Author response: We would like to thank the reviewer for the careful consideration of our manuscript and for providing insightful feedback. We have addressed the reviewer's questions and concerns through further experimentation, references to relevant published data, and adjusting the manuscript to improve comprehension. Changes in the manuscript are highlighted in red and the rebuttal letter features references to the main revised passages by line number.

The reviewer raised the concern “that no information about cell numbers for the newly described cell states is provided”. We thank the reviewer for bringing this missing information to our attention, which is now included in the manuscript as follows. In the SfSRC model, the number of basal_Krt13+ cells and club_Krt15+ cells was 88 and 19 cells, respectively, which is now mentioned in line 374 and 375 of the results. Furthermore, 56 basal-goblet_div cells were identified in the GFP+ SRC model (**Fig. 1**), and we included this information in the results section (line 156).

In addition, the reviewer lacked information on the number of animals from which the cells were originated, or in other words, whether the identified rare conditions were derived from more than one biological replicate. The number of mice used in each run was indicated in the

legend of each figure. To increase the visibility of this information, we have now also included the mouse numbers in **Supplementary Table 15** (now **Supplementary Table 1**). Given that club_Krt15⁺ cells are only observed in the SfSRC model on day 2 for which we used nine mice, we cannot exclude the possibility that they originate from a single mouse, although it seems unlikely. We now included this information in line 378-381 of the results. On the other hand, basal_Krt13⁺ cells and basal-goblet cells are present at different timepoints (**Fig. 6D and Fig. 1H, respectively**), as well as in different mouse models (**Fig. S6G**), and are therefore detected in multiple biological replicates.

An important point that we would like to briefly address at the beginning is that the reviewer finds that we do not show functional data in our study, but remain descriptive only. This misunderstanding is based on an inadequate explanation from our side of the CycB1-GFP mouse model used, which we would like to clarify briefly here, and in more detail later in the response letter. This model allows the functional measurement of dividing cells by expressing the proliferation marker CycB1-GFP in all cells throughout the organism (including the lung). CycB1-GFP is a fusion protein of the N-terminal portion of the cyclin B1 protein and eGFP, and behaves like the natural full-length cyclin B1 protein in terms of expression during the cell cycle. It is degraded in the G0/G1 phase of the cell cycle via the ubiquitin-proteasome system so that cells are GFP negative. In the S/G2/M stages of the cell cycle, the fusion protein is stable and cells are GFP positive. Therefore, in each experiment we isolated and enriched actively proliferating cells based on GFP positivity and examined them by scRNA-seq, which is a unique feature of our study and has never been done before.

The reviewer also felt that the variety of different cell sorting and scRNA-seq profiling strategies for the SRC mouse model often had little added benefit, and that parts of the study were highly repetitive and unclear. We respectfully disagree with this statement, as none of the experiments performed are redundant, but of course accept the criticism that they seem to be presented in a clear way in some places. However, we could correct these inaccuracies more effectively if the reviewer had made them more explicit. We feel that we have taken much care to clearly present the different scRNA-seq experiments and why we have performed them. For example, we show a scheme of the experimental setup in each figure (**Fig. 1C, 2A, 3A, 4A, 6C**). Nevertheless, we indicate in the following the specific approach and the most important value for each experiment:

- **Fig. 1** (SRC model): This is the only scRNA-seq experiment in which we isolated epithelial GFP⁺ cells exclusively and acquired single cell transcriptomes of actively dividing lung cells based on biological data as opposed to computationally predicted data. We subsequently identified the type of dividing cells that were activated upon selective loss of Scgb1a1⁺ cells (which are mainly club cells) and characterized their transcriptome.
- **Fig. 2** (SRC model): This is the only experiment in which we isolated and analyzed mesenchymal GFP⁺ dividing cells (enriched with the functional cell division marker) and non-dividing cells.
- **Fig. 3** (SRC model): We sorted and analyzed enriched GFP⁺ dividing cells and non-dividing cells allowing their integration and comparison. This is the only experiment in which we analyzed five time points (day 0 until day 4) and identified differentially expressed genes (DEGs) in the different epithelial cell populations after injury compared to homeostasis. Apart from the upregulation of cell-cycle related genes in

all the previously identified dividing populations, our data suggested a role of club, AT2, and ciliated cells in immune activation following epithelial injury.

- **Fig. 4** (SRC model): By greatly reducing the number of the most abundant cell types, i.e. ciliated and AT2 cells, during FACS sorting, we substantially enriched for rare cell types. In addition, GFP⁺ dividing cells were enriched. This allowed us to characterize distinct DTA⁺ populations, to identify a DTA-like population in COVID-19 patients, and we showed that DTA⁺ cells express specific chemokines and growth factors that could trigger an immune response and contribute to lung regeneration through epithelial and mesenchymal cell support.
- **Fig. 6** (SfSRC model): Here we added AT2 cell depletion (Sftpc⁺ cells; alveolar progenitors) to the club cell depletion (Scgb1a1⁺ cells; bronchiolar progenitors) and applied the sorting strategy as in Fig. 4. This allowed us to uncover two new rare epithelial populations in the distal mouse lung: basal_Krt13⁺ that resembled human Sftpc⁻ Krt5⁺ basal cells, and a population of hillock-like club cells expressing Krt15 (club_Krt15⁺). Additionally, this detailed and comprehensive dataset allowed us to predict differentiation trajectories.

The reviewer also argues that we describe a dedifferentiation from goblet to basal cells that has been described previously in bleomycin-induced injury (Lange *et al.* Nat. Methods 2022), suggesting a new progenitor role for goblet cells in the distal lung. While we acknowledged the previous finding of Lange *et al.* in line 372 (now line 418) of the results section and line 435 (now line 511) of the discussion section, we now included a more detailed description of their results, and explained the innovation that our study brings (line 496-511). Lange *et al.* predicted a differentiation path from goblet to basal cells through trajectory inference, and showed the existence of an intermediate cell state between goblet and basal (BPIFB1⁺ KRT5⁺) in the intrapulmonary airways. The existence of this intermediate cell type confirms a differentiation path between basal and goblet, but it does not show that goblet cells can act as progenitors, since differentiation can be occurring in either direction. This is particularly critical between these cell types, because, while basal cells are known epithelial progenitors, goblet cells are thought to be terminally differentiated cells. More specifically, the ability to divide *in vivo* is intrinsic to non-terminally differentiated cells and was never shown before for goblet cells. Therefore, we were quite intrigued by our finding that goblet cells, as well as the identified basal-goblet cells, indeed proliferate upon epithelial injury in several of our *in vivo* experiments. Additionally, we isolated these dividing cells and characterized their transcriptome by scRNA-seq. In accordance with the study by Lange *et al.*, our directionality inferred by RNA velocity suggests that goblet cells give rise to basal cells, thereby confirming the results obtained in bleomycin-injured lungs by Lange *et al.* with a completely different injury model used by us. This strongly suggests that goblet to basal dedifferentiation could indeed be a general mechanism of the lung during regeneration.

We address the remaining reviewer comments regarding the (lipo)fibroblasts and the viability of DTA-expressing cells below, as these were again raised as major points.

Major points:

- *In the method section no information regarding QC filtering of single cell data is provided, including %Mito, Min/Max of genes/counts per cell, and based on which criteria doublets were removed. This is also nowhere reported in the manuscript. This information is extremely critical especially for the goblet-basal_div cell type and DTA+ cell findings.*

Author response: We agree with the reviewer that showing details on QC filtering is very important and therefore provide this information in line 621-623 (now line 714-716) in the Method section and more detailed for each cell sample in **Supplementary Table 15** (now **Supplementary Table 1**). This table features the percentage of mitochondrial genes, the number of genes, and the number of counts for which each sample was filtered. Unfortunately, we did not point out this additional data until the Methods section, which is easy to miss. For more clarity, we now already refer to **Supplementary Table 1** in the Results section (line 126).

In addition, we now provide new QC data showing the number of counts, percentage of mitochondrial genes, and number of genes for each cell type (**Fig. S1H** and **Fig. S4C**). These figures include DTA⁺ cells and show that these cells have neither a higher percentage of mitochondrial genes nor a lower number of genes per cell and therefore represent viable cells (see below for more information on this topic).

Concerning the analysis of cell doublets, we had previously written in the Methods section: “Cell doublets were identified using scDbfFinder v1.2.0⁹, and clusters of cells composed of doublets were excluded from the analysis.” To improve clarity, we have now changed this sentence (line 719) to: “Cell doublets were calculated with scDbfFinder() (scDbfFinder v1.2.0⁹) using the default parameters and providing a vector of the runs id in the samples parameter. Clusters composed of doublets were excluded from the analysis.” Additionally, we added a UMAP embedding depicting the doublet classification for each cell (**Fig. S1F**) to demonstrate that all populations, and in particular the basal-goblet_div cells, are not composed by doublets (line 158 in the manuscript text).

• *Regarding the goblet-basal_div cell type please provide evidence that is not a doublet and thus an artefact (stainings, showing doublet scores). This state is only described in Fig1; in all subsequent figures it is not detected anymore? Why is that? Given the putative goblet-to-basal trajectory could that be an intermediate state?*

Author response: As briefly mentioned above, we added a new UMAP embedding showing the doublet classification for each cell type, including basal-goblet_div cells, which consist of only single cells (**Fig. S1F**; line 158 in the manuscript text). We are pleased that the reviewer agrees with our hypothesis that the basal-goblet_div cells could be an intermediate cell state between goblet and basal cells, and have described this now even more clearly in the manuscript text (line 157 and line 419-421). Additionally, we noticed that cells expressing both goblet (BPIFB1) and basal (KRT5) markers were previously noticed in the intrapulmonary airways of bleomycin-injured mice¹⁰. We added this information, which supports the existence of a basal-goblet intermediate state, to the discussion section (line 496-505).

We thank the reviewer for pointing out that we only mention basal-goblet_div cells in **Fig. 1** and then do not mention them any further leading to confusion. Basal-goblet_div cells are described solely in **Fig.1** because they formed a distinct cluster only when GFP⁺ proliferating cells were selectively sorted and analyzed. Once non-dividing epithelial cells are included in the analysis, basal-goblet_div cells are assigned to basal and/or goblet cell clusters. The analysis of dividing GFP⁺ cells in isolation was therefore critical to identify specific cell types. When all datasets are integrated, one can observe that basal-goblet cells from **Fig.1** colocalize with cells from other datasets in the UMAP embedding (**Fig. S6G**). We now improved **Fig. S6G** by including an UMAP with all integrated datasets, showing the annotation of the epithelial cell types in the different “Fig. UMAPs” without discriminating cell cycle status (i.e. removed “_div”), changing the size and color of the points to allow for better visualization, and pointing out that the UMAP of **Fig.1** only shows dividing cells. In addition, we projected the

cells from **Fig. 3** and **Fig. 4** (SRC model) and from **Fig. 6** (SfSRC model) into the annotated cells from **Fig. 1** (dividing SRC) using scmap¹¹. These analyses show that some basal and basal_div cells from the SRC and SfSRC model are, in fact, more similar to basal-goblet_div cells than to basal cells (**Author Response Fig. 3a-c**). Additionally, we show that basal-goblet cells of the SfSRC model cluster between basal and goblet cells in the UMAP embedding, express basal and goblet cell genes, and display a similar transcriptional profile to the basal-goblet_div cells from **Fig.1** (**Author Response Fig. 3d-e**). The **Author Response Fig. 3c-f** are also included as Supplementary Figures in the manuscript (**Fig. S6H-J** and **Fig. S7B**).

Author Response Fig. 3: Projection of cells used in Fig. 3 analysis (SRC model) **(a)**, Fig. 4 analysis (SRC model) **(b)**, and Fig. 6 analysis (SfSRC model) **(c)** onto Fig. 1 (SRC dividing) cells using scmap. Green: highlighted similarity of basal and basal_div cells (left) with basal-goblet_div cells (right). **(d)** UMAP embedding showing the distribution of basal-goblet cells (dark green) in relation to the other cell types in the SfSRC model. **(e)** Diffusion map of dividing and non-dividing basal and goblet cells, and

basal-goblet cells with RNA velocity vectors indicating transition from goblet to basal through basal-goblet cells. **(f)** Heatmap of the top 30 upregulated genes in basal-goblet cells from the SfSRC model ranked by fold change (test= MAST, $p_{adj} < 0.05$). Genes in bold are common markers to basal-goblet_div cells from Fig.1.

• *Apart from cell cycle scoring, have you assessed proliferation markers in your single cell data as well as on tissue level?*

Author response: We apologize for not properly explaining the CycB1-GFP mouse model used in our study. As briefly mentioned above, this model allows the functional measurement of proliferating cells through expression of a cyclin B1-derived proliferation marker. In each experiment, actively dividing cells were identified, isolated, and enriched based on the protein expression of the CycB1-GFP transgene. We realized that, to avoid misunderstandings, we needed to better describe the functionality of the transgene in these mice and now provide a more detailed explanation in the methods section (lines 555- 561) and rephrased the sentence in line 108-113 where the mouse line is first introduced.

In detail, CycB1-GFP transgenic mice, previously developed and extensively validated by Klochendler et al (2012) ¹², constitutively express a fusion protein of the 105 N-terminal residues of the cyclin B1 protein and eGFP. If cells are in G0 or G1 stages of the cell cycle, the cyclin B1 portion is ubiquitinated by the APC/C complex, directing the fusion protein for degradation, thereby rendering cells GFP negative. In the S/G2/M stages of the cell cycle, the activity of APC/C is low, the CycB1-GFP protein is not degraded, and the cells are GFP⁺. Therefore, we apply a functional analysis rather than a bioinformatics analysis of dividing cells, which is a unique feature of our study.

In **Fig. 1D**, we calculated cell cycle scoring to confirm that the isolated GFP⁺ cells were indeed in S/G2/M phases of the cell cycle. On a tissue level, we provide co-immunofluorescence staining of dividing GFP⁺ cells combined with markers for club cells (CC10), goblet cells (AGR2), basal cells (KRT5), AT1 cells (PDPN), AT2 cells (SPC), adventitial fibroblasts (CD34), and alveolar fibroblasts (NPNT), demonstrating their proliferation in lung tissue (**Fig. S1B**). Additionally, we show that basal, goblet, club, and AT2 cells upregulate the expression of cell cycle-related genes upon epithelial injury (**Fig. 3C, Fig. S3E, and Supplementary Table 5**).

As a further functional readout, we now performed immunofluorescence analysis of injured lungs at day 3 showing that GFP⁺ cells are also Ki-67⁺ (**Author Response Fig. 4a**). Because Ki-67 is also expressed by cells in the G1 stage of the cell cycle, some Ki-67⁺ GFP⁻ cells are expected, as was observed in the original publication (**Author Response Fig. 4b**) ¹². Finally, we show that the clusters annotated as dividing in **Fig. 1, Fig. 4, and Fig. 6** have higher Mki67 expression (**Author Response Fig. 4c**).

Author Response Fig. 4: (a) Immunofluorescence staining of lungs from SRC mice with CycB1-GFP (yellow), Ki-67 (red), and DAPI (blue) on day 3 after tamoxifen injection. Scale bar: 50 μ m. The two images shown are representative of at least three animals analyzed. (b) Flow cytometry analysis of Ki-67 (Cy5-Ki67) and GFP (Cy2-GFP) expression in hepatocytes from a 25 days-old CycB1-GFP mouse. Image is taken from Klochendler *et al.*¹². (c) UMAP embedding showing Mki67 expression (top panels) of cells analyzed in Fig. 1, Fig. 4, and Fig. 6.

• From Fig3 on you perform AT2 and or ciliated cell “depletions”, yet still large amounts of both cell types are still present. Please comment and rephrase accordingly

Author response: We thank the reviewer for this comment, which makes us realize that this was not sufficiently explained, and in particular the schematic images in **Fig. 3A, 4A and 6C**, where we only indicate “depletion”, are misleading. From **Fig. 3** on, we “partially depleted” AT2, and from **Fig. 4** on, we partially depleted AT2 and ciliated cells. The goal was to enrich rare cell types by reducing the most common ones. Since we did not want to completely exclude AT2 and ciliated cells from our analysis, hence the “partially depleted” wording in the manuscript text, we still included around 10% of each population in relation to the total amount of cells. Additionally, dividing epithelial cells, sorted on the base of GFP expression, were included in their totality, and added to the epithelial cells before running Chromium. Since

GFP⁺ dividing cells independent of their cell types were sorted separately, dividing AT2 cells were not depleted.

In **Fig. 3A, 4A, and 6C**, we changed “depleted” to “partial depletion”, the text in the results section was modified accordingly (lines 258-260), and in the Methods section we included a more detailed explanation of the sorting strategy (lines 635-639). We hope that this sorting strategy is now clearer with the terms "partial depletion" or "reduction in cell number" as also used in the manuscript text.

• *The high fraction of dividing AT2 cells with emergence of a novel cluster at Day 4 p.i. in the SRC model is very interesting. How does this cluster differ from the other AT2 cluster? You state in the manuscript that the high number of dividing AT2 cells (starting d2) was also surprising to you. What could be the mechanism? Could cell depletion trigger inflammatory events?*

Author response: We agree with the reviewer that this is an interesting and unexpected observation. A direct comparison of the transcriptomes of dividing and non-dividing cells is challenged by the fact that most of the differentially expressed genes (DEGs) in dividing cells are related to cell division. We could remove classical cell cycle-related genes from the transcriptome of the dividing cells before comparing them to non-dividing cells, but there might still be genes present that are indirectly changed by cell division.

To understand whether the dividing AT2 cells differed from the non-dividing AT2 cells, we decided to filter out all AT2 cells from Fig. 3 at day 4 and recalculate the UMAP distribution and clustering before and after regressing out cell cycle scores. After regressing cell cycle scores, both dividing and non-dividing AT2 cells formed a unique cluster (**Author Response Fig. 5a**). Moreover, differential expression analysis (fold change ≥ 2 , adjusted p value < 0.05) between dividing and non-dividing clusters revealed only cell cycle-related genes (**Author Response Fig. 5b and c**). These calculations suggest that AT2_div and AT2 cells are the same cells and differ only in cell cycle status.

Additionally, to better characterize the role of AT2 cells following epithelial injury, we now show the expression of all DEGs in AT2 cells through time (**Fig. S3F**). This heatmap shows, apart from the upregulation of cell cycle-related genes (in red), that several genes associated with immune response and inflammation, such as *Ii33*, *Chia1*, *Cd14*, *Lgasl3*, *Ly6e*, and *Ly6c1* (in bold), were increasingly upregulated in the majority of AT2 cells, suggesting a role of these cells in immune activation. These new data are now included in the manuscript text in lines 228-233, and the title of the chapter (line 212) was adapted accordingly. Based on the excellent point raised by the reviewer and previously shown data that inflammatory signals induce AT2 proliferation and differentiation (Choi et al, 2020) that are in line with our new data on upregulation of pro-inflammatory genes in our mouse model, we also included this hypothesis in the discussion (lines 477-483).

Author Response Fig. 5: (a) UMAP embedding showing clustering of dividing and non-dividing AT2 cells at day 4 after tamoxifen injection in SRC mice, before and after regressing out cell cycle scores. (b) Expression of all DEGs (fold change ≥ 2 , adjusted p value < 0.05) between cluster 0 (dividing AT2) and cluster 1 (non-dividing AT2) from UMAP without cell cycle regression (panel a, lower left UMAP). (c) Gene set enrichment analysis (GSEA) of DEGs between dividing and non-dividing AT2 cells on day 4.

• *What is the phenotype of the SRC mouse? For the SfSRC mouse should provide histopathological data, but not for the SRC. Is there inflammation happening?*

Author response: This is an interesting point that we addressed with additional analyses. Overall, the SRC mice showed no symptoms at any time point after tamoxifen administration compared to the control mice. Of the SfSRC mice, 26% (5 out of 19 mice followed for more than 5 days) suddenly died or had to be sacrificed 5-7 days after tamoxifen administration because they were moribund.

Masson staining of lungs 14 days after tamoxifen administration in SfSRC mice (**Fig. S6A**) provided evidence that there are no signs of fibrosis. To assess whether inflammatory infiltrates were present in the lungs of SfSRC mice, we performed H&E staining of lungs at different time points after tamoxifen administration. Histological analysis of these whole lung sections from SfSRC mice four days after tamoxifen injection revealed a mixed inflammatory infiltrate composed mainly of lymphocytes and plasma cells and a few neutrophil granulocytes, with perivascular and peribronchial accentuation. This pattern became more pronounced two weeks after injury induction, and in addition, the lungs exhibited an accumulation of intraveloar macrophages in form of foam cells that was virtually absent in controls (**Author Response Figure 6**). If the reviewer wishes, we are happy to include these new data as Supplementary Material in the manuscript.

Author Response Fig. 6: H & E staining of SfSRC mouse lungs before (day 0), and 4 and 14 days after two consecutive tamoxifen injections. Images are representative of at least 2 animals per group. Black arrowhead: plasma cell; white arrowhead: lymphocyte; gray arrowhead: neutrophil granulocyte; red arrowhead: alveolar macrophage. Scale bar: 200 μ m. Insets on the top right on each picture are 2x magnifications of the demarcated area in the main picture.

• *Regarding the DTA⁺ cells, it is not clear whether that indeed are viable or simply dying cells. Please provide info about their quality (%Mito, No of counts/gene per cell etc). Please provide stainings together with apoptosis markers.*

Author response: We fully understand the reviewer's concerns about the viability of DTA⁺ cells, which we had questioned ourselves during our analyses. Although we do not know the lifespan of DTA⁺ cells, we are confident that these cells were viable during sorting and scRNA-seq analysis and survived long enough to divide and/or differentiate. Moreover, the time that these cells are viable might be sufficient to play a role in triggering an inflammatory response. The following data support that DTA⁺ cells are alive, at least at the time of analysis:

1. During cell sorting, we gate on viable cells (DAPI negative) to exclude dead cells (DAPI positive) (**Fig. S3A and S4A**).
2. QC data for DTA⁺ cells show that they do not have a higher percentage of mitochondria genes or less genes or lower counts per cell, and therefore represent viable cells at the time of analysis. These new data are now included in the manuscript (**Fig. S1H and S4C**).

We agree that co-staining of DTA⁺ cells with apoptosis markers would be a great addition to the data we already provide. Unfortunately, this experiment is complicated by the fact that there are no working antibodies to detect the DTA protein to specifically identify DTA⁺ cells. We therefore hope that the reviewer will find the data now provided sufficient.

• *To compare similarity in cell type signature between datasets/species, it is highly recommend to perform matchScore analysis and/or marker gene signature scorings. That does not require dataset integration.*

Author response: Thank you for this suggestion. Marker gene signature scorings and matchScore are useful to compare already identified populations, but do not allow the identification of new populations such as DTA-like cells in the COVID-19 dataset. Instead, we ran matchScore2 to assess the similarity between cells from COVID-19 patients and populations from the SRC mouse model, since it can assess the expression of previously calculated cell type signatures in single cells. For this, we generated "humanized" Seurat objects from the SRC mouse epithelial cells at day 0 and day 2 (experiment from Fig. 4) by replacing mouse genes with their human orthologs, and calculated the 100 top marker genes for each population (DTA⁺_Sftpc⁺, DTA⁺_club, DTA⁺_goblet, and DTA⁺_Foxj1⁺ cells were considered as a single DTA⁺ population). After training the model with the default parameters, its accuracy was calculated to be 0.96. However, cell type identification of the cells in the COVID-19 dataset using the SRC model as a reference was not accurate (**Author Response**

Fig. 7a). A large proportion of cells was wrongly annotated, even when using a probability threshold of 0.9 (**Author Response Fig. 7b**), hampering the identification of rare cell types such as DTA-like cells.

Integration of datasets from different species using Seurat has been shown to successfully enable the identification of similar cell populations across species, even for rare cell types¹³. Accordingly, integration of human lung datasets with SRC mouse data allowed us to identify a small population of cells in COVID-19 patient samples that otherwise would be scattered throughout the dataset. We now included in the Methods section of the manuscript (lines 763-782) a more detailed explanation on how this integration was achieved (see next point). In **Author Response Fig. 7**, we show the UMAP embedding after integration of the datasets, with the original cell annotation of the SRC (**Author Response Fig. 7c**) and COVID-19 (**Author Response Fig. 7d**) cells. Clustering and annotation after integration, done according to the identity of the majority of cells in each cluster (based on the COVID-19 dataset annotation), are shown in **Author Response Fig. 7e** and **7f**, respectively. **Author Response Fig. 7g** shows that there is a high correspondence between the previous annotation of SRC cells (**Fig. 4G**) and the annotation done after integration. This is also true for the different DTA⁺ clusters, as all their cells were included in the new DTA-like population (**Author Response Fig. 7g**). Finally, **Author Response Fig. 7h** shows that there is also a high correspondence between the previous annotation of the COVID-19 cells and the annotation after integration with the SRC dataset, which was expected, since the original COVID-19 annotation was used to annotate the new clusters

In conclusion, we think that Seurat integration, which corrects for batch effect, followed by re-clustering of cells based on their nearest neighbors represents the best strategy for the identification of rare common populations between datasets of different species.

Author Response Fig. 7: (a,b) Sankey diagrams showing the correspondence between the original annotation of cells from the Melms *et al.* dataset and the cell type determined by matchScore2 using the top 100 signature genes from the SRC cell populations and a probability threshold (p.threshold) of 0.5 (a) or 0.9 (b). (c-f) UMAP embedding after integration of SRC and Melms *et al.*¹⁴ datasets showing the original annotation of cells from the SRC dataset (c), the original annotation of cells from the Melms *et al.* dataset (d), re-clustering of both datasets (e), and annotation of the clusters based on the identity of the majority of cells in each cluster (according to “cell_type_fine”, Melms *et al.*) (f). (g,h) Sankey diagrams showing the correspondence between the original annotation of SRC cells and their annotation after integration with the Melms *et al.* dataset (g), and the original annotation of cells from the Melms *et al.* dataset and their annotation after integration with the SRC dataset (h).

• *The identification of DTA⁺-like cells in COVID-19 in Fig 4c is not all clear. It is*

1) *not clear how you did the integration and which cell annotation was used*

Author response: We apologize for not explaining this clearly enough. We now included a more detailed explanation on the integration and annotation of the DTA⁺-like cells in COVID-19 lungs in the paragraph “Integration with COVID-19 dataset and DEGs analysis” of the methods section (lines 763-782). Additionally, we included UMAP embeddings after integration of SRC and Melms *et al.*¹⁴ datasets showing the original annotation of cells from the SRC dataset, the original annotation of cells from the Melms *et al.* dataset, and the re-clustering of both datasets (**Fig. S4J-L**). We hope that these changes help clarify how the integration and annotation were done.

2) *what the marker signature of the DTA-like cells in the human disease setting his. Here indeed it is critical to not integrate but perform matchScore and/or marker gene signature scorings (see previous point).*

Author response: Although the human DTA⁺-like population was identified by integration with our mouse dataset, the gene signature for these cells (**Supplementary Table 8**) was calculated using cells from the COVID-19 dataset only. Meaning that DEGs were calculated comparing DTA⁺-like cells from the COVID-19 dataset with all the other cells from the same dataset. We have now explained this in more detail in the Methods section (lines 763-782) to hopefully make this point more understandable. Concerning matchScore analysis, please see previous point.

Please also confirm this in other COVID-19 datasets and provide evidence on the tissue level.

Author response: This is another great suggestion by the reviewer and we confirmed our findings using another COVID-19 dataset¹⁵. After integrating the SRC dataset from **Fig. 4** with healthy and COVID-19 samples from Chua *et al.*¹⁵, a cluster including DTA⁺ cells and human epithelial cells was identified (cluster 7, **Author Response Fig. 8a-c**). COVID-19 cells included in this cluster differentially expressed genes related to cytokine, NF-κB, and MAPK signaling when compared to all other cells from the human COVID-19 lungs, supporting their similarity to DTA⁺ cells (**Author Response Fig. 8d-e**). We would also like to mention that in the initial analysis with different lung diseases in which we detected the high expression of the COVID-19 signature by DTA⁺ cells (now **Fig. S4I**), a third COVID-19 dataset by Blanco-Melo *et al.*¹⁶ was used, although this was somewhat hidden in the text (line 301).

The detection of lung epithelial cells sharing the gene signature with murine DTA⁺ cells in lung tissue from COVID-19 patients would indeed be an interesting additional piece of information. However, this would require intensive studies in primary human lung material, which would be a separate study in itself. We hope that the reviewer agrees with us that a detailed analysis of human COVID-19 lungs is not our focus and is far beyond the scope of this study.

Author Response Fig. 8: (a-c) UMAP embedding after integration of SRC and Chua *et al*¹⁵ datasets showing annotation of cells from the SRC dataset (a), annotation of cells from healthy and COVID-19 patient cells (b), and re-clustering of integrated data. Arrow points to the cluster containing DTA⁺ SRC cells and cells from Chua *et al.* dataset (cluster 7) (c). **d**) Expression of top 50 DEG in cluster 7 compared to all cells in the COVID-19 samples from Chua *et al* dataset. **e**) GSEA of DEG in cluster 7 compared to all cells in the COVID-19 samples from Chua *et al* dataset (FC>1.5, *p*_{adj} < 0.05). Terms that were also enriched in DTA⁺ cells are in bold.

• For the cell trajectory inference, it would be critical compare the results to CellRank (which does not need direction information).

Author response: Thank you for the suggestion. We now ran CellRank¹⁰ analysis for trajectory inference and compared it with our current results. Initial and terminal states were calculated using the default parameters, and the overall trajectory results are shown using a directed PAGA graph (**Author Response Fig. 9**). As in our previous analysis, goblet cells are predicted to differentiate into basal cells, club_div cells are predicted to be the source for club and AT2 cells, and DTA⁺ club cells are predicted to differentiate into DTA⁺_FoxJ1⁺ cells (blue arrows).

Author Response Fig. 9: UMAP embedding of SfSRC cells (all time points) with PAGA directed graph calculated with CellRank¹⁰. Pie charts show cell fates averaged per cluster, dashed lines denote connectivities, and arrows denote transitions. Arrow thickness indicates transition probability. Blue arrows denote transitions mentioned in the Author Response.

- As stated in the general comments, the organoid assays with adventitial and alveolar fibroblasts are not convincing and contradictory. Please provide as a control and reference also data with the “whole” mesenchyme. Please report on colony size and forming capacity. How did you distinguish alveolar from broncholalveolar organoids? I have not seen any IF stainings?

Author response: We appreciate the reviewer’s comment, which also relates to comment #5 of reviewer #1. Several studies have previously characterized mesenchymal populations of the adult mouse lung that are, unfortunately, often annotated differently. We chose to use the annotation from a study that identified the mesenchymal cell types based on scRNA-seq and immunofluorescence, and in which the identified populations strongly overlap with ours (Tsukui et al. 2020)³ (now indicated in line 180 of the manuscript). As mentioned by the reviewer in the general comments, Barkauskas et al.¹⁷ showed that PDGFRA⁺ mesenchymal cells were able to support lung epithelial organoid formation. Since both adventitial and alveolar fibroblasts express Pdgfra (Tsukui et al.³ and **Author Response Fig. 10a**), these are rather subpopulations of the previously described PDGFRA⁺ cells. Accordingly, we did not use PDGFRA to distinguish between adventitial and alveolar fibroblasts during sorting, but used CD34 and SCA-1 (*Ly6a* gene), which are highly expressed in adventitial fibroblasts (**Fig. 2C and S2D**). In the CD34⁺ SCA-1⁻ population, alveolar fibroblasts were discriminated from the rest of the cells by their expression of PDGFRA and NPNT (see sorting strategy in **Fig. S2C**).

Furthermore, Barkauskas et al. used AT2 cells as epithelial progenitors, while in our study, we used epithelial progenitor cells¹⁸, which are closer to bronchiolar cells.

Another important difference is the number of cells used in both studies. While Barkauskas *et al.* co-cultured 5,000 AT2 cells with 100,000 PDGFRA^{high} cells, we used 1,000 progenitor epithelial cells with 20,000 adventitial or alveolar fibroblasts.

To avoid confusion with previously described mesenchymal populations, we added the following changes (underlined) to the text in line 190: “To test this, we performed organoid cultures with epithelial progenitor cells (EPCAM^{high} CD24^{dim}) in co-culture with adventitial fibroblasts (PDGFRA⁺ CD34⁺ SCA-1⁺) or alveolar fibroblasts (CD34⁻ SCA-1⁻ PDGFRA⁺ NPNT⁺).”

Additionally, we now included a paragraph in lines 460-471 of the discussion, pointing out the overlap between previously described populations, with the goal of harmonizing annotations, minimize misunderstandings, and acknowledge the authors’ contribution to the field.

Alveolar and bronchioalveolar organoids were identified based on morphology, which is now indicated in the manuscript text (line 192). However, we can remove this classification, if the reviewer does not find it convincing. When co-cultured with adventitial cells, 1,000 epithelial progenitor cells gave rise to, on average, 29 organoids (**Author Response Fig. 10b**) of different sizes, ranging from 0.125 to 1.075 mm in diameter. This information is now included in the manuscript text (lines 193-194).

Overall, we showed that adventitial cells, a PDGFRA⁺ mesenchymal subpopulation, proliferate upon epithelium injury and is able to support epithelial organoid formation from epithelial progenitor cells. This is a new finding and is not contradictory with previous studies.

Author Response Fig. 10: (a) UMAP embeddings of SRC mesenchymal cells at day 0 showing cell annotation and expression of *Pdgfra*. (b) Number of epithelial organoids per well when 1,000 epithelial progenitors are co-cultured with adventitial or alveolar fibroblasts for 3 weeks.

Reviewer #3 (Remarks to the Author):

Summary

Transgenic mice were engineered to express DTA in Scgb1a1-CreER – CycB1-GFP mice to express DTA in response to tamoxifen, thereby activating apoptosis and causing lung injury. Cell division was monitored by the expression of GFP during injury and repair and followed for three days. GFP+ dividing cells were sorted on days 2 and 3, isolating diverse proliferating epithelial cells GFP+ mesenchymal cells, club, goblet, basal, and AT2 cells were identified. Surviving DTA+ epithelial cells were identified. Goblet cells were proposed as epithelial progenitors. In vitro cultures were used to demonstrate the selective growth of organoids by co-culture with adventitial fibroblasts. Extensive bioinformatic analyses identified temporal and cell-type gene expression patterns, including inflammatory mediators. Comparative analyses with human lung data sets were used to conclude that DTA+ cells identified in mice were similar to those in the lungs of Covid-19 patients. A diversity of cell types were described by sub-clustering of the epithelial cells; trajectory analyses were used to predict progenitors. [SPC-CreERT2], Scgb1a1-CreERT2 mice were used to delete both airway and alveolar cells, and repair processes followed for 2-56 days, from which distinct Krt13 and Krt15 cells were identified.

The Authors conclude that the use of selective DTA lung epithelial cell injury identifies new cell types, that DTA+ expressing cells survive and resemble the populations of epithelial cells from the lungs of patients succumbing from Covid-19.

Overview

This is an extensive data set with complex bioinformatic analyses using well-established algorithms. The authors' major conclusions focused to the identification of a number of distinct epithelial cell types and progenitors which were identified during DTA-induced lung injury. While it is not surprising that a diversity of conducting airway cells, e.g., club, goblet, basal, and AT2 cells, are proliferative, the authors have not demonstrated that they proliferate and undergo asymmetric cell division and therefore contribute to the expansion of cell lineages. This conclusion needs further validation. Present conclusions are drawn primarily from extensive bioinformatic analyses and single-cell RNA profiles but are generally without experimental validation.

Author response: We would like to thank the reviewer for carefully reading our manuscript and for providing constructive feedback on our data. We have addressed the reviewer's questions and concerns through further experimentation, references to relevant published data, and adjusting the manuscript to improve comprehension. Changes in the manuscript are highlighted in red and the rebuttal letter features references to the main revised passages by line number.

Although it might not be surprising that a variety of conducting airway cells are proliferative, it has never been shown experientially that club, goblet, basal, and AT2 cell types divide simultaneously *in vivo* after epithelial injury. In particular, it has never been shown that goblet cells, which were considered to be terminally differentiated, divide at all. Cell division is essential during tissue regeneration and it is an intrinsic feature of non-terminally differentiated cells.

An important point we would also like to address is the reviewer's comment that we have not demonstrated that the cells proliferate/divide. This misunderstanding is based on an inadequate explanation from our side of the CycB1-GFP mouse model used, which we would like to clarify. This model, previously developed and extensively validated by Klochendler *et*

*al.*¹², allows the functional measurement of dividing cells by expressing the proliferation marker CycB1-GFP in all cells throughout the organism (including the lung). CycB1-GFP is a fusion protein of the N-terminal portion (first 105 amino acids) of the cyclin B1 protein and eGFP, and behaves like the natural full-length cyclin B1 protein in terms of expression during the cell cycle. Therefore, it is degraded in the G0/G1 phase of the cell cycle via the ubiquitin-proteasome system so that cells are GFP negative. In the S/G2/M stages of the cell cycle, the fusion protein is stable and cells are GFP positive. Therefore, GFP⁺ dividing cells can be found in all tissues of CycB1-GFP mice, it is not an inducible system, and GFP expression is dynamic rather than a static labelling of cells that divide at a certain stage. Accordingly, we apply a functional analysis in each experiment rather than a bioinformatic analysis of actively proliferating cells based on GFP positivity, which is a unique feature of our study and has never been done before. We realized that to avoid misunderstandings we needed to better describe the functionality of the CycB1-GFP transgene in these mice and now provide a more detailed explanation in the methods section (lines 555- 561) and rephrased the sentence in line 108-113 where the mouse line is first introduced.

In **Fig. 1D**, we calculated cell cycle scoring to confirm that GFP⁺ sorted cells are indeed in the S/G2/M stages of the cell cycle. On a tissue level, we provide co-immunofluorescence staining of dividing GFP⁺ cells combined with markers for club cells (CC10), goblet cells (AGR2), basal cells (KRT5), AT1 cells (PDPN), AT2 cells (SPC), adventitial fibroblasts (CD34), and alveolar fibroblasts (NPNT), demonstrating their proliferation in lung tissue (**Fig. S1B**).

As a further functional readout, we now performed immunofluorescence analysis of injured SRC lungs at day 3 showing that GFP⁺ cells are also Ki-67⁺ (**Author Response Fig. 11a**). Because Ki-67 is also expressed by cells in the G1 stage of the cell cycle, some Ki-67⁺ GFP⁻ cells are expected, as was observed in the original publication¹² (**Author Response Fig. 11b**). Finally, we found that the clusters annotated as dividing in **Fig. 1**, **Fig. 4**, and **Fig. 6** have higher Mki67 expression (**Author Response Fig. 11c**).

To prove asymmetric cell division or to validate the differentiation trajectories that we computationally predicted in this study, we would need to establish multiple new mouse models with complex genotypes that would allow parallel lineage tracing. Such experiments would take many years and are in our opinion beyond the scope of this manuscript.

Author Response Fig. 11 (same as Author Response Fig. 4 in response to reviewer #2): (a) Immunofluorescence staining of lungs from SRC mice with CycB1-GFP (yellow), Ki-67 (red), and DAPI (blue) on day 3 after tamoxifen injection. Scale bar: 50 μ m. The two images shown are representative of at least three animals analyzed. (b) Flow cytometry analysis of Ki-67 (Cy5-Ki67) and GFP (Cy2-GFP) expression in hepatocytes from a 25 days-old CycB1-GFP mouse. Image is taken from Klochendler *et al.*¹². (c) UMAP embedding showing Mki67 expression (top panels) of cells analyzed in Fig. 1, Fig. 4, and Fig. 6.

The present experiments are complicated by many variables, including the timing of driver gene expression and allele recombination and the possible variability of recombination. Since the TAM may be active throughout the repair process during a period in which differentiation of epithelial cells is changing dynamically with injury, differentiation, and inflammation, it is unclear that specific cell states are being targeted during the course of the experiments.

Author response: We acknowledge the reviewer's comment that our models are complicated by the variables mentioned, but we respectfully disagree, which we would like to explain in more detail in the following.

The presence of tamoxifen during the repair process does not invalidate the results obtained in this study or complicate their interpretation. For example, when using the SRC model, only

cells expressing Scgb1a1 are targeted, and the ones undergoing Cre recombination are expected to die. Cells that become Scgbb1a1⁺ during the course of data collection (e.g. by differentiation) can also undergo recombination, express DTA, and die. However, this is selectively true for Scgb1a1⁺ cells, and no other cell type can be damaged by DTA. The same is true for the SfSRC model, where the same considerations apply to Sftpc⁺ cells. Therefore, we consider our model quite controlled and predictable. In contrast, the use of chemicals or virus to induce lung injury, which has been broadly used in the field of lung regeneration, is highly unspecific concerning targeted cells, as their effect on epithelial progenitors, the micro-environment, or any other cell type is unknown.

Regarding the recombination efficiency and timing of transgene expression (i.e. DTA), we demonstrated in **Fig. 1B** that most Scgb1a1⁺ cells are rapidly lost. It is clear that not all Scgb1a1⁺ cells undergo recombination and die due to DTA expression. However, we see this as an advantage, as Scgb1a1⁺ cells are still around and can participate in the regeneration process. The same considerations apply to the Sftpc-CreER mouse model.

Therefore, we are convinced that the mouse models used and the experiments performed are indeed an excellent and very controlled strategy to find and characterize new progenitor populations.

Rather, it appears that multiple cell types, as defined at homeostasis, may be continuously labeled throughout the experimental course. The conditional DTA experiments depend upon knowledge of the timing of DTA expression, thus specific cell types expressing DTA and perhaps their susceptibility to apoptosis.

Author response: Unfortunately, it is not entirely clear to us what the reviewer refers to by "... continuously labeled" and we assume this is also based on the misunderstanding with the CycB1-GFP mouse line. As mentioned above, GFP expression is not a continuous labeling strategy that depends on the presence of tamoxifen, but it labels dividing cells in a dynamic and functional manner. This means, when cells enter G0 or G1 after cell division, they become GFP⁻ again.

Regarding the second comment, we carefully tested the timing of cell depletion due to DTA expression using immunofluorescence and showed examples of these experiments in **Fig. 1B** and **S3D**. Based on these results, we chose our analysis time points. Since we start seeing substantial depletion of Scgb1a1⁺ cells after two days, and we were interested in understanding the early events upon epithelial injury, we focused our analysis on day 2 and day 3. If seen of value, we can also include an immunofluorescence image from day 1 (shown in this response letter as **Author Response Fig. 2** to reviewer #1).

Regarding the susceptibility of cells to apoptosis by DTA, we want to state that the Rosa26-DTA mice have been used in numerous studies to deplete specific cell types in different organs ^{2, 19, 20, 21}. DTA is highly toxic by inhibiting protein synthesis through inactivation of elongation factor 2 (EF-2), which is required for protein synthesis, and its mechanism of action has been extensively characterized ²². Since all cells depend on protein synthesis, DTA is generally toxic to all cells independent of cell type. In addition, there are no off-target bystander effects of DTA on neighboring cells, since the catalytic subunit alone (i.e. DTA) does not bind to cell surfaces and murine cells do not express diphtheria toxin receptor.

Regarding the proposal of goblet cells as progenitors, secretory cells differentiate reversibly during inflammation and metaplasia; thus, it may not be surprising that goblet cells or other secretory cells are in the process of dynamic differentiation during injury, they share club,

basal, and goblet cell characteristics and therefore serve as proliferative cells and may undergo recombination during the experiments since Scgb1a1 may be expressed during differentiation and activate Cre.

Author response: We thank the reviewer for the comment. It has been shown by *in vivo* lineage tracing of Scgb1a1⁺ cells that secretory cells can dedifferentiate to basal cells during lung regeneration²³. Since the study was done in the mouse trachea where goblet and club cells co-exist, and both club and goblet cells express Scgb1a1, it is not possible to discern which cell type(s) have the ability to dedifferentiate. Although it seems not surprising that goblet cells have this property, it has yet to be proven experimentally. The only indication that this might be true for goblet cells was provided by Lange *et al.*¹⁰ (cited and discussed in our manuscript) using bioinformatics analysis.

Therefore, we were quite intrigued by our finding that goblet cells, as well as the identified basal-goblet cells, actively proliferate upon epithelial injury in several of our *in vivo* experiments. The ability to divide *in vivo* is intrinsic to non-terminally differentiated cells and was never shown before for these cell types. Additionally, we isolated these dividing cells based on a functional proliferation marker and characterized their transcriptome by scRNA-seq. Directionality inferred by RNA velocity suggested that goblet cells give rise to basal cells, thereby supporting the results obtained in bleomycin-injured lungs by Lange *et al.* with a completely different injury model used by us. This suggests that goblet to basal dedifferentiation could indeed be a general mechanism of the lung during regeneration (see also the modified text passages in the results (419-421) and the discussion (lines 496-505).

The organoid experiments with adventitial fibroblasts likely include Pdgfra⁺ mesenchymal cells, findings consistent with previously published work. Present experiments are not accompanied by quantitative data; a more careful characterization of the organoid and the evaluation of the purity of the fibroblast after isolation is needed.

Author response: We thank the reviewer for bringing this relevant information to our attention, which was also commented on in a similar way by the other reviewers. Adventitial and alveolar fibroblasts express Pdgfra⁺³ (**Author Response Figure 12a**), so both of these cell types are subpopulations of the previously described Pdgfra⁺ cells¹⁷. Pdgfra⁺ cells were shown to support epithelial organoid formation, but this has not been demonstrated separately for adventitial or alveolar fibroblasts. To avoid confusion with previously described mesenchymal populations, we added the following changes (underlined) to line 190 of the manuscript: “To test this, we performed organoid cultures with epithelial progenitor cells (EPCAM^{high} CD24^{dim}) in co-culture with adventitial fibroblasts (PDGFRA⁺ CD34⁺ SCA-1⁺) or alveolar fibroblasts (CD34⁻ SCA-1⁻ PDGFRA⁺ NPNT⁺).”

Alveolar and bronchioalveolar organoids were identified based on morphology. We can remove this classification, if the reviewer does not find it convincing. When co-cultured with adventitial cells, 1,000 epithelial progenitor cells gave rise to, on average, 29 organoids (**Author Response Figure 12b**) of different sizes, ranging from 0.125 to 1.075 mm in diameter. This information is now included in the manuscript text (line 193).

We did not assess the purity of the isolated fibroblasts after sorting. We rather checked by mass spectrometry, if the use of surface markers such as CD34, SCA-1, PDGFRA, and NPNT enabled the isolation of the adventitial and alveolar fibroblast populations identified by scRNA-seq. Accordingly, we show that CD34⁺ SCA-1⁺ cells expressed adventitial markers such as DCN and COL14A1 (adventitial fibroblasts), while PDGFRA⁺ NPNT⁺ cells express INMT and

COL13A1 (alveolar fibroblasts) (Fig. S2D). We hope that the reviewer will find this analysis sufficient.

Author Response Fig. 12 (same as Author Response Fig. 10 in response to reviewer #2): (a) UMAP embeddings of SRC mesenchymal cells at day 0 showing cell annotation and expression of *Pdgfra*. (b) Number of epithelial organoids per well when 1,000 epithelial progenitors are co-cultured with adventitial or alveolar fibroblasts for 3 weeks.

The authors have also identified the survival of a number of cells expressing DTA; is the DTA protein expressed? What are the proposed mechanisms underlying their resistance, and what biological insights are to be derived from the resistance? While gene expression studies suggest similarities between DTA⁺ epithelial subsets and cells identified in lungs from Covid-19 patients, mechanistic insights regarding their role in Covid-19 pathogenesis are not provided.

Author response: In our study, we show that DTA mRNA is expressed, but, unfortunately, cannot determine DTA protein because there are no functional anti-DTA antibodies for immunofluorescence or other applications available. However, based on the fact that cells expressing DTA are clearly transcriptionally distinct from their counterparts (e.g., upregulation of proinflammatory factors), we are confident that DTA protein is also expressed.

The integration of our scRNA-seq data with scRNA-seq datasets from various human lung diseases (including COVID-19) aimed to verify whether, despite the artificial nature of DTA expression in cells, this model can be used to reveal cell populations and/or mechanisms in human lung diseases. One hypothesis for the similarity between human COVID-19 cells and murine DTA⁺ cells is that the intracellular expression of DTA triggers a non-self response mechanism, as seen in a virus infection. In accordance, some DTA⁺ cells also show high expression of an influenza gene signature (Fig. S4I). Nonetheless, DTA⁺ cells are transcriptionally much closer to epithelial COVID-19 cells due to overexpression of specific cytokines and chemokines. This population of epithelial cells might contribute to triggering the cytokine storm that is often seen in COVID-19 patients. This cytokine storm can lead to development of acute respiratory distress syndrome and death, but the mechanisms that trigger the inflammation are unclear. This hypothesis was now included in the discussion part of the manuscript (lines 527-532).

Understanding why an epithelial cell from a COVID-19 patient expresses more inflammatory factors than an epithelial cell from an influenza patient, and investigations to gain insights into the mechanism of tissue injury in COVID-19 patients and lung regeneration, would indeed be an interesting additional piece of information. However, this would require intensive studies in primary human lung material, which would be a separate study in itself. We hope that the

reviewer agrees with us that a detailed analysis of human COVID-19 lungs is not our focus and is far beyond the scope of this study.

References

1. Liu Q, *et al.* Lung regeneration by multipotent stem cells residing at the bronchioalveolar-duct junction. *Nature Genetics* **51**, 728-738 (2019).
2. Salwig I, *et al.* Bronchioalveolar stem cells are a main source for regeneration of distal lung epithelia in vivo. *EMBO J* **38**, e102099-e102099 (2019).
3. Tsukui T, *et al.* Collagen-producing lung cell atlas identifies multiple subsets with distinct localization and relevance to fibrosis. *Nat Commun* **11**, 1920 (2020).
4. Hurskainen M, *et al.* Single cell transcriptomic analysis of murine lung development on hyperoxia-induced damage. *Nat Commun* **12**, 1565 (2021).
5. Xie T, *et al.* Single-Cell Deconvolution of Fibroblast Heterogeneity in Mouse Pulmonary Fibrosis. *Cell Reports* **22**, 3625-3640 (2018).
6. Rawlins EL, *et al.* The role of Scgb1a1+ Clara cells in the long-term maintenance and repair of lung airway, but not alveolar, epithelium. *Cell Stem Cell* **4**, 525-534 (2009).
7. Yu J, Liu SL. Emerging Role of LY6E in Virus-Host Interactions. *Viruses* **11**, (2019).
8. Zheng D, *et al.* Regeneration of alveolar type I and II cells from Scgb1a1-expressing cells following severe pulmonary damage induced by bleomycin and influenza. *PLoS One* **7**, e48451 (2012).
9. Germain P-L, Lun A, Garcia Meixide C, Macnair W, Robinson MD. Doublet identification in single-cell sequencing data using scDbtFinder. *F1000Research* **10**, 979-979 (2022).
10. Lange M, *et al.* CellRank for directed single-cell fate mapping. *Nat Methods* **19**, 159-170 (2022).
11. Kiselev VY, Yiu A, Hemberg M. scmap: projection of single-cell RNA-seq data across data sets. *Nat Methods* **15**, 359-362 (2018).
12. Klochendler A, *et al.* A Transgenic Mouse Marking Live Replicating Cells Reveals In Vivo Transcriptional Program of Proliferation. *Developmental Cell* **23**, 681-690 (2012).
13. Butler A, Hoffman P, Smibert P, Papalexi E, Satija R. Integrating single-cell transcriptomic data across different conditions, technologies, and species. *Nat Biotechnol* **36**, 411-420 (2018).
14. Melms JC, *et al.* A molecular single-cell lung atlas of lethal COVID-19. *Nature* **595**, 114-119 (2021).
15. Chua RL, *et al.* COVID-19 severity correlates with airway epithelium-immune cell interactions identified by single-cell analysis. *Nat Biotechnol* **38**, 970-979 (2020).
16. Blanco-Melo D, *et al.* Imbalanced Host Response to SARS-CoV-2 Drives Development of COVID-19. *Cell* **181**, 1036-1045 e1039 (2020).
17. Barkauskas CE, *et al.* Type 2 alveolar cells are stem cells in adult lung. *The Journal of clinical investigation* **123**, 3025-3036 (2013).

18. Bertocello I, McQualter J. Isolation and Clonal Assay of Adult Lung Epithelial Stem/Progenitor Cells. *Current Protocols in Stem Cell Biology*, 1-12 (2011).
19. Xing YL, *et al.* High-efficiency pharmacogenetic ablation of oligodendrocyte progenitor cells in the adult mouse CNS. *Cell Rep Methods* **3**, 100414 (2023).
20. Voehringer D, Liang HE, Locksley RM. Homeostasis and effector function of lymphopenia-induced "memory-like" T cells in constitutively T cell-depleted mice. *J Immunol* **180**, 4742-4753 (2008).
21. Li N, *et al.* Ablation of somatostatin cells leads to impaired pancreatic islet function and neonatal death in rodents. *Cell Death Dis* **9**, 682 (2018).
22. Collier RJ. Understanding the mode of action of diphtheria toxin: a perspective on progress during the 20th century. *Toxicon* **39**, 1793-1803 (2001).
23. Tata PR, *et al.* Dedifferentiation of committed epithelial cells into stem cells in vivo. *Nature*, (2013).

REVIEWER COMMENTS

Reviewer #1 (Remarks to the Author):

I appreciate the careful consideration the authors put into addressing the concerns of myself and the other reviewers.

Reviewer #2 (Remarks to the Author):

The authors have diligently addressed the major concerns I raised and provided satisfactory answers to my questions. Nevertheless, there are still a few remaining questions and comments that I believe should be addressed:

1. Regarding the low cell numbers of the newly discovered cell state and whether they come from different biological replicates (i.e., animals). In your response, you mentioned, "Given that club_Krt15+ cells are only observed in the SfSRC model on day 2, for which we used nine mice, we cannot exclude the possibility that they originate from a single mouse, although it seems unlikely." It would be beneficial to clarify in the manuscript that mice were pooled for scRNAseq, and single-cell data was not derived from each mouse individually, as indicated in Supplementary Table 1.
2. The inclusion of histological analysis for SfSRC mice is important and should be added to the manuscript.
3. I appreciate the additional explanation you provided about the mouse model used in the study. However, my concern regarding the missing functional validation and experimental proof remains. Specifically, I am referring to the absence of mechanistic studies regarding the newly discovered cell states and their potential progenitor functions. To address this limitation, I recommend adding a dedicated "Limitations" section to the end of the discussion. This section should highlight the need for further research, particularly in terms of functional data and mechanistic studies related to the identified cell states. Furthermore, it would be beneficial to acknowledge in the "Limitations" section the absence of DTA+/DTA+ like cell protein/tissue validations in the mouse models and COVID-19. Addressing these limitations in a dedicated section will provide a more comprehensive outlook and guide future research directions.

Reviewer #3 (Remarks to the Author):

This is a revised paper in which DTA was expressed in Scgb1a1-CreT mice. Scgb1a1-CreT, together with Sftpc-CreT mice, were also used to induce injury of both AT2 and Scgb1a1 cells. Single cell RNA data was subjected to complex bioinformatics analyses to identify cells and processes during the regeneration and repair process. The authors have added data and addressed some issues raised by the reviewers. Major concerns raised in the review regarding the overstatement of findings regarding these new cell types and the importance and mechanism of survival of DTA-expressing cells, and the relationship to Covid-19 persist are in need of experimental validation to support major conclusions.

Specific Comments:

Abstract: The conclusion that they "resemble" cells in Covid-19 patients is based on

bioinformatics similarity and perhaps is an overstatement without orthogonal validation. It is also well-established that club cells and basal cells are airway progenitors.

Line 71-72: Demonstrated not suggested can contribute to regeneration of the alveoli.

Line 109: What is meant by "functional manner"?

Line 137: Regarding the proliferation of AT2 cells after Scgb1a1-CreT – were alveolar AT2 cells killed during the experiment? Was there necrosis or apoptosis of AT2 cells after exposure to TAM? Were the RNAs carefully corrected for ambient RNA? How are the authors assured of this correction? Sftpc is very highly expressed and often detected in multiple cell types in the lung. How are the authors assured that SoupX fully corrected the data from ambient RNA?

Line 303: The proposed Covid-associated "DTA-like cells" is an overreach and without functional biological validation based on bioinformatics similarity. Is this cell unique in Covid-19 and in present experiments? Are they similar to other cells during lung injury or stress? Expression of DTA is not a useful biomarker for human disease. What is the unique nature of this "unique cell type"? Needed are experiments designed to purify and identify its unique properties experimentally. Do these cells survive and persist in their transcriptome for prolonged periods?

The authors used the expression of inflammatory mediators (chemokines and cytokines) by lung epithelial cells as if this is a new concept. It is, however, well established that epithelial cells in the lung contribute to inflammatory signaling.

Line 401-402: The authors identify "new cell types" on the basis of gene expression patterns and markers based on clustering. Functional and experimental validations are needed to conclude that these are, in fact, novel cell types and not cell states. What unique capabilities do these cells have?

Line 460-461: The sentence is unclear as written; suggest deleting "presenting mainly club cells."

Line 527-530: Regarding non-self response". This is somewhat overly speculative, and without experimental validation, the close link between DTA-like cells and subsets of cells in Covid-19 lung is overly stated.

MethodSoupX s: How are the authors assured that SoupX is sufficiently controlled for ambient RNA?

"Viable" DTA-expressing cells needs more than mito DNA eval to validate. Do they persist? How long? What is the method by which cells can remain viable expressing DTA?

There is a concern that the authors have not clearly identified cell types from stressed cells "state" or variability within cell types without functional validation. Stressed cells are found in infections, e.g., Covid, and express cytokines transiently during injury and death.

REVIEWER COMMENTS

Reviewer #1 (Remarks to the Author):

I appreciate the careful consideration the authors put into addressing the concerns of myself and the other reviewers.

Reviewer #2 (Remarks to the Author):

The authors have diligently addressed the major concerns I raised and provided satisfactory answers to my questions. Nevertheless, there are still a few remaining questions and comments that I believe should be addressed:

Author response: Thank you for again carefully reviewing our manuscript, the positive and encouraging comments, and the constructive feedback. We addressed all the concerns raised and further adjusted the manuscript to improve clarity. Changes are highlighted in red in the revised manuscript and referenced by line numbers in our point-by-point response.

1. Regarding the low cell numbers of the newly discovered cell state and whether they come from different biological replicates (i.e., animals). In your response, you mentioned, "Given that club_Krt15+ cells are only observed in the SfSRC model on day 2, for which we used nine mice, we cannot exclude the possibility that they originate from a single mouse, although it seems unlikely." It would be beneficial to clarify in the manuscript that mice were pooled for scRNAseq, and single-cell data was not derived from each mouse individually, as indicated in Supplementary Table 1.

Author response: Thank you for this suggestion. We have added the following sentence to the methods section to be clearer on this point (line 627): "In all experiments, single-cell suspensions from different mouse lungs were pooled before fluorescence-activated cell sorting was performed". Furthermore, we included in the paragraph "scRNA-seq analysis" of the methods section that pooled lungs were used for scRNA-seq (line 705).

2. The inclusion of histological analysis for SfSRC mice is important and should be added to the manuscript.

Author response: We fully agree and included the histological analysis of SfSRC mice in the Supplementary material as Fig. S6B and the description of the corresponding results in the manuscript text lines 369 - 373.

3. I appreciate the additional explanation you provided about the mouse model used in the study. However, my concern regarding the missing functional validation and experimental proof remains. Specifically, I am referring to the absence of mechanistic studies regarding the newly discovered cell states and their potential progenitor functions. To address this limitation, I recommend adding a dedicated "Limitations" section to the end of the discussion. This section should highlight the need for further research, particularly in terms of functional data and mechanistic studies related to the identified cell states. Furthermore, it would be beneficial to acknowledge in the "Limitations" section the absence of DTA+/DTA+ like cell protein/tissue validations in the mouse models and COVID-19. Addressing these limitations in a dedicated section will provide a more comprehensive outlook and guide future research directions.

Author response: We agree with the reviewer that it would be helpful to point out the missing functional validations in a separate "limitations paragraph" in the discussion. However, while writing such a paragraph, we realized that it became too long, in particular in terms of the maximum number of words allowed, as we had to repeat partially the results for better comprehension. We therefore decided to include the mentioned limitations at the appropriate place in the discussion. First, we added the following text to the discussion of DTA⁺/DTA⁺-like cells, which is just before the last paragraph of the discussion (lines 540-543): "Of note, the DTA⁺ cell population in the mouse lungs was defined by the presence of DTA mRNA, and validation of DTA protein expression remains to be performed. Likewise, the presence of the DTA⁺-like cells in the tissue of

COVID-19 lungs must still be verified.” Second, we added the following sentence to the last paragraph of the discussion (lines 545-547; the same “limitation” is also mentioned in lines 497-498 of the discussion):” *We also identified novel cell types and/or cell states, although we would like to point out that further studies are needed to determine their exact functional and mechanistic role.”* In this way, the two statements are close to each other towards the end of the discussion, so that they mimic, to some extent, a “ limitation paragraph”.

Reviewer #3 (Remarks to the Author):

This is a revised paper in which DTA was expressed in Scgb1a1-CreT mice. Scgb1a1-CreT, together with Sftpc-CreT mice, were also used to induce injury of both AT2 and Scgb1a1 cells. Single cell RNA data was subjected to complex bioinformatics analyses to identify cells and processes during the regeneration and repair process. The authors have added data and addressed some issues raised by the reviewers. Major concerns raised in the review regarding the overstatement of findings regarding these new cell types and the importance and mechanism of survival of DTA-expressing cells, and the relationship to Covid-19 persist are in need of experimental validation to support major conclusions.

Author response: Thank you for again taking the time to review our manuscript. We addressed the concerns raised and performed additional experiments and analyses. Changes are highlighted in red in the revised manuscript and referenced by line numbers in our point-by-point response.

Regarding the criticism of overstatement, we believe that we have presented our results clearly without overstatement. We have described hypotheses as such and not as established evidence. Furthermore, at the suggestion of reviewer #2, we have now even explicitly pointed out that the newly identified cell types/cell stages and the DTA⁺/DTA⁺-like cells have yet to be functionally and mechanistically investigated in subsequent studies. This new text can be found at the end of the discussion (lines 540 – 543).

Specific Comments:

Abstract: The conclusion that they “resemble” cells in Covid-19 patients is based on bioinformatics similarity and perhaps is an overstatement without orthogonal validation. It is also well-established that club cells and basal cells are airway progenitors.

Author response: To be more precise, we now wrote that DTA-expressing cells “transcriptionally resemble” a population present in COVID-19 patients.

We do not understand the reviewer’s second comment since we did not claim to have identified basal and club cells as lung progenitors in either the abstract or the manuscript text.

Line 71-72: Demonstrated not suggested can contribute to regeneration of the alveoli.

Author response: We have replaced the word “suggested” with “demonstrated”.

Line 109: What is meant by “functional manner”?

Author response: “Functional manner” in this sentence is to emphasize that the identification of dividing cells is not based on bioinformatic methods, but that CycB1-GFP mice allow the identification of cells that divide at the time they perform this function.

Line 137: Regarding the proliferation of AT2 cells after Scgb1a1-CreT – were alveolar AT2 cells killed during the experiment? Was there necrosis or apoptosis of AT2 cells after exposure to TAM? Were the RNAs carefully corrected for ambient RNA? How are the authors assured of this correction? Sftpc is very highly expressed and often detected in multiple cell types in the lung. How are the authors assured that SoupX fully corrected the data from ambient RNA?

Author response: The first part of this comment refers to the SRC model, in which we induced DTA expression and subsequent apoptosis of Scgb1a1⁺ cells after tamoxifen administration. To determine whether AT2 cells were also killed in this experiment, we performed immunofluorescence for cleaved (active) caspase-3 in the lung tissue of these mice. As expected, at day 2 after tamoxifen, caspase-3 positivity is evident in bronchial cells as they undergo apoptosis by DTA, whereas AT2 cells do not show caspase-3 staining and therefore remain viable (**Author response Fig. 13**). In contrast, as expected, we observed caspase-3-positive apoptotic AT2 cells and consequently fewer AT2 cells in the SfSRC model at day 2 after tamoxifen, in which AT2 cells are directly targeted by DTA (**Author response Fig. 13**).

Author response Fig. 13: Immunofluorescence analysis of SRC and SfSRC mouse lungs with antibodies detecting active caspase-3 (red) and Lamp3 (yellow, AT2 cells), and DAPI (blue, nuclei) on day 0 and day 2 after tamoxifen administration. Arrows: apoptotic airway cells. Arrow heads: apoptotic AT2 cells. Scale bar: 100µm. Representative images of at least three analyzed mice per condition are shown.

With respect to the second part of the comment, we can assure the reviewer that we carefully corrected for ambient RNA using SoupX¹, a well-established tool for removal of free mRNA contamination from droplet based scRNA-seq data². For each dataset, we checked for possible contamination with several highly expressed marker genes from different cell types (including *Sftpc*) and adjusted the SoupX parameters if necessary. The list of genes and the datasets for which we did not use the default SoupX parameters are described in the “Processing of scRNA-seq data” section of the Methods (lines 721-725). To demonstrate to the reviewer that the data are fully corrected for *Sftpc*, we show in **Author response Fig. 14** the *Sftpc* counts before and after correction with SoupX for the dataset of Fig. 1. Finally, we would like to point out that clustering and cell annotation are fairly robust towards background noise because they are based on gene expression signatures rather than individual genes. For example, AT2 cells are annotated not only by *Sftpc* expression, but also by other AT2-specific canonical markers such as *Napsa* and *Lamp3*, which are among the top differentially expressed genes in the identified dividing AT2 population (**Fig. 1G**).

Author response Fig. 14: (A, B) Expression of *Sftpc* before (left) and after correction with SoupX (right) (B) of the dataset used in Fig. 1 (A).

Line 303: The proposed Covid-associated "DTA-like cells" is an overreach and without functional biological validation based on bioinformatics similarity. Is this cell unique in Covid-19 and in present experiments? Are they similar to other cells during lung injury or stress? Expression of DTA is not a useful biomarker for human disease. What is the unique nature of this "unique cell type"? Needed are experiments designed to purify and identify its unique properties experimentally. Do these cells survive and persist in their transcriptome for prolonged periods? The authors used the expression of inflammatory mediators (chemokines and cytokines) by lung epithelial cells as if this is a new concept. It is, however, well established that epithelial cells in the lung contribute to inflammatory signaling.

Author response: These are questions and statements combined in one commentary that address different topics, which we respond to separately in the following.

Regarding "The proposed Covid-associated "DTA-like cells" is an overreach and without functional biological validation based on bioinformatics similarity.": We respectfully disagree with the reviewer and would like to emphasize that our description of the "DTA⁺-like cells" is in no way overstated but we clearly communicate that their identification is based solely on transcriptional similarity to DTA⁺ cells.

Regarding "Is this cell unique in Covid-19 and in present experiments? Are they similar to other cells during lung injury or stress?": Integration of scRNA-seq data from SRC lung cells at day 2 after tamoxifen with data sets from other mouse injury models or human lung diseases showed that DTA⁺-like cells were almost exclusive to and displayed a more robust transcriptional signature (more differentially expressed genes (DEG) and uniquely expressed genes) in COVID-19 patients. For example, we could not identify DTA⁺-like cells in influenza-infected human lung cells³ (Author response Fig. 15A) or in LPS-injured mouse lung cells⁴ (Author response Fig. 15B). We found scarce DTA⁺-like cells (n = 30) in bleomycin-injured mouse lungs⁵, but these expressed a less distinct gene signature (lower number of genes and less specific genes) as compared to the DTA⁺-like cells found in COVID-19 lungs (Author response Fig. 15C). Accordingly, DTA⁺ cells exhibit a stronger expression of the COVID-19 gene signature in comparison to other lung disease signatures (Fig. S4I).

Regarding "Expression of DTA is not a useful biomarker for human disease.": We do not claim that DTA is a useful biomarker for human disease. We rather describe that DTA⁺ cells share a distinct gene expression profile with DTA⁺-like cells in COVID-19 lungs.

Regarding "What is the unique nature of this "unique cell type"? Needed are experiments designed to purify and identify its unique properties experimentally. Do these cells survive and persist in their transcriptome for prolonged periods?": We assume that these questions relate to the DTA⁺-like cells. As stated in our first response letter, a detailed functional analysis of DTA⁺-like cells in COVID-19 lungs would require intensive studies on primary human lung material (including the development of cell isolation strategies and the acquisition of fresh human lung material), which would be a separate study in itself. Ultimately, these experiments are of interest but are well beyond the reasonable scope of this report. Nonetheless, we

acknowledge the need to further analyze these DTA-like cells using COVID-19 patient material, and included this point in the discussion (lines 540-543).

Regarding “The authors used the expression of inflammatory mediators (chemokines and cytokines) by lung epithelial cells as if this is a new concept. It is, however, well established that epithelial cells in the lung contribute to inflammatory signaling.”: We would like to strongly disagree with this statement and clarify that we do not claim to have discovered that lung epithelial cells can secrete pro-inflammatory factors. In contrast, we even begin the corresponding paragraph with the sentence “*Inflammatory pathways, such as the ones described above, can be activated in epithelial cells by microbial components and lead to the initiation of an immune response through chemokine secretion*” and cite a wonderful review by Hewitt and Lloyd on the ability of epithelial cells to influence the immune response in the lung (lines 312-314).

We would like to emphasize that, in our study, we distinguish between two different types of pro-inflammatory epithelial responses. On the one hand, we describe the upregulation of cell type-specific pro-inflammatory factors in AT2, club, and ciliated cells after Scgb1a1⁺ cell depletion (**Fig. 3D, F and G, Fig. S3F**). On the other hand, we show that injured cells express distinct pro-inflammatory transcripts and share a common expression profile independently of their cell of origin (**Fig. 4D and E, Fig. 5A and B, Fig. S5B**). Notably, this transcriptional profile was also identified in rare epithelial cells from COVID-19 patients (**Fig. 4H and I**).

Author response Fig. 15: Integration of the SRC scRNA-seq dataset (day 0 and day2) with scRNA-seq data from influenza-infected airway epithelial human cells ³ (A), LPS-injured mouse lung cells ⁴ (B), and bleomycin-injured mouse lung cells ⁵ (C). The UMAPs on the left side show the SRC cell types together with the different datasets used for integration. The three UMAPs on the right panels depict the clusters calculated upon integration of the SRC cells with each injury model. (C) The arrows in the UMAPs to the right point to the DTA⁺-like cells (cluster 9). (D) Heatmap showing the expression of the DEG of cluster 9 representing DTA⁺-like cells in the bleomycin-injured dataset. These are only 21 genes, which are also not very specific to this cell type, as mentioned in the text above.

Line 401-402: The authors identify "new cell types" on the basis of gene expression patterns and markers based on clustering. Functional and experimental validations are needed to conclude that these are, in fact, novel cell types and not cell states. What unique capabilities do these cells have?

Author response: The use of single-cell techniques has enabled the identification of new cell types, cell sub-types, transitional cells, as well as physiological and disease-associated cell states. Currently, there is an ongoing debate as to when a subpopulation of a canonical cell type should be recognized as a novel cell type rather than a more specialized subtype or a transient state induced by a particular cellular or

experimental condition, and the lung is not an exception⁶. For example, Hillock cells found in the mouse trachea by Montoro et al. and Plasschaert et al.^{7,8} were described as transitional cells, while human Hillock-like cells were considered a new cell type by the integrated Human Lung Cell Atlas (HLCA)⁹.

Independently of how basal_Krt13⁺ and club_Krt15⁺ cells are called, we showed that these cells have a distinct transcriptional profile and cluster separately from basal and club cells, respectively (**Fig. 6D and F**). Gene set enrichment analysis (GSEA) of club_Krt15⁺ markers suggests that these cells might exert an immunomodulatory role (**Author response Fig. 16A**), which is consistent with the similarity to Hillock cells. Basal_Krt13⁺ specifically express *Lgals7* (**Author response Fig. 16B**), a lectin expressed by keratinocytes and found mainly in squamous epithelia¹⁰. Accordingly, basal_Krt13⁺ show an enrichment in genes implicated in epithelium keratinization (**Author response Fig. 16C**). Nonetheless, club_Krt15⁺ and basal_Krt13⁺, as well as the other new cell populations, need to be functionally characterized in future studies, and we mention this in lines 497-498 and again in lines 545-547 of the discussion section.

Author response Fig. 16: (A) Gene set enrichment analysis of club_Krt15⁺ marker genes (Metascape)¹¹. (B) Expression of *Lgals7* in epithelial lung cells from the SfsRC model (all timepoints). (C) Protein-protein interactions of basal_Krt13⁺ marker genes calculated with MCODE algorithm (Metascape)¹¹.

Line 460-461: The sentence is unclear as written; suggest deleting "presenting mainly club cells."

Author response: We deleted this part of the sentence, as suggested by the reviewer.

Line 527-530: Regarding non-self response". This is somewhat overly speculative, and without experimental validation, the close link between DTA-like cells and subsets of cells in Covid-19 lung is overly stated.

Author response: This paragraph was added in response to reviewers #1 and #3 during the first revision. It is indeed speculative and a hypothesis that was not tested in this study, therefore we started the sentence

with "One hypothesis is ...". Nevertheless, since the reviewer considers this to be an exaggeration, we have deleted this text passage.

MethodSoupX s: How are the authors assured that SoupX is sufficiently controlled for ambient RNA?

Author response: This has already been answered in the question above on the same topic.

"Viable" DTA-expressing cells needs more than mito DNA eval to validate. Do they persist? How long? What is the method by which cells can remain viable expressing DTA? There is a concern that the authors have not clearly identified cell types from stressed cells "state" or variability within cell types without functional validation. Stressed cells are found in infections, e.g., Covid, and express cytokines transiently during injury and death.

Author response: Besides the demonstration that DTA⁺ cells do not have a higher percentage of mitochondria genes or less genes or lower counts per cell (**Fig. S1H and S4C**), and therefore represent viable cells at the time of analysis, we also want to point out that we gate on viable cells (DAPI negative) during cell sorting to exclude dead cells (DAPI positive) (**Fig. S3A and S4A**). Accordingly, although we do not know the exact lifespan of DTA⁺ cells, we are confident that these cells were viable during sorting and scRNA-seq analysis, and survived long enough to divide and/or differentiate (into DTA⁺ Foxj1⁺ cells).

Regarding the question about the method by which the cells could survive despite DTA expression, we hypothesize that there is a dose-dependent response to DTA, meaning that the cells would be able to tolerate lower amounts of DTA. It is also possible that certain epithelial cells are intrinsically resistant to DTA. Although it would be interesting to understand the mechanism that allows the presence of viable DTA⁺ cells in the lung, this would not change our main findings about these cells: (i) they persist despite DTA expression, (ii) they can divide and differentiate, and (iii) they have a distinct transcriptional profile compared with other epithelial cells.

Finally, we would like to emphasize that we carefully assessed the cell of origin for each DTA⁺ population by comparing their transcriptional profile to the expression of the top markers in the normal populations (**Fig. 4D**) and to the overall transcriptional similarity to the normal populations (**Fig. S4F**).

References

1. Young MD, Behjati S. SoupX removes ambient RNA contamination from droplet-based single-cell RNA sequencing data. *Gigascience* **9**, 1-10 (2020).
2. Heumos L, *et al.* Best practices for single-cell analysis across modalities. *Nat Rev Genet* **24**, 550-572 (2023).
3. Medaglia C, *et al.* An anti-influenza combined therapy assessed by single cell RNA-sequencing. *Commun Biol* **5**, 1075 (2022).
4. Riemondy KA, *et al.* Single cell RNA sequencing identifies TGFbeta as a key regenerative cue following LPS-induced lung injury. *JCI Insight* **5**, (2019).
5. Strunz M, *et al.* Alveolar regeneration through a Krt8+ transitional stem cell state that persists in human lung fibrosis. *Nat Commun* **11**, 3559 (2020).
6. Dudchenko O, Ordovas-Montanes J, Bingle CD. Respiratory epithelial cell types, states and fates in the era of single-cell RNA-sequencing. *Biochem J* **480**, 921-939 (2023).
7. Montoro DT, *et al.* A revised airway epithelial hierarchy includes CFTR-expressing ionocytes. *Nature* **560**, 319-324 (2018).

8. Plasschaert LW, *et al.* A single-cell atlas of the airway epithelium reveals the CFTR-rich pulmonary ionocyte. *Nature* **560**, 377-381 (2018).
9. Sikkema L, *et al.* An integrated cell atlas of the lung in health and disease. *Nat Med* **29**, 1563-1577 (2023).
10. Saussez S, Kiss R. Galectin-7. *Cell Mol Life Sci* **63**, 686-697 (2006).
11. Zhou Y, *et al.* Metascape provides a biologist-oriented resource for the analysis of systems-level datasets. *Nat Commun* **10**, 1523 (2019).

REVIEWERS' COMMENTS

Reviewer #2 (Remarks to the Author):

The authors have satisfactorily answered my questions and addressed all of my concerns.

Reviewer #3 (Remarks to the Author):

The authors have addressed many of the technical and editorial issues raised in previous reviews. The authors infer lineages and progenitors computationally. Since the injured cells divide and differentiate, they likely transition through multiple cell states and intermediates. What they become after regeneration and after homeostasis is less well clarified. Precise lineage tracing and orthogonal validation, perhaps by directly isolating and studying the proposed novel cell types identified by the authors, would be useful in validating the conclusions in the present work. The authors have added text regarding the limitations of the discussion, as requested. The authors provide a carefully prepared and extensive data set that perhaps would be useful to the field in supporting further investigation of the diverse cell types or states predicted from their bioinformatic experiments.

REVIEWERS' COMMENTS

Reviewer #2 (Remarks to the Author):

The authors have satisfactorily answered my questions and addressed all of my concerns.

Reviewer #3 (Remarks to the Author):

The authors have addressed many of the technical and editorial issues raised in previous reviews. The authors infer lineages and progenitors computationally. Since the injured cells divide and differentiate, they likely transition through multiple cell states and intermediates. What they become after regeneration and after homeostasis is less well clarified. Precise lineage tracing and orthogonal validation, perhaps by directly isolating and studying the proposed novel cell types identified by the authors, would be useful in validating the conclusions in the present work. The authors have added text regarding the limitations of the discussion, as requested. The authors provide a carefully prepared and extensive data set that perhaps would be useful to the field in supporting further investigation of the diverse cell types or states predicted from their bioinformatic experiments.

Author response: Thank you again for carefully reviewing our manuscript. We agree that lineage tracing studies, as well as *in vitro* studies, of the newly identified cells and progenitors will indeed consolidate our findings and improve the understanding of their function and relation to other cell types, and we are pleased that the reviewer now accepts the discussion we have added in this regard. We are also delighted that the reviewer now considers the data presented to be "carefully prepared", "extensive", and "useful to the field", and we are confident that our results can serve as a catalyst for the development of new studies.